# Minimax Optimal Reinforcement Learning with Quasi-Optimism

**Harin Lee**
Seoul National University
harinboy@snu.ac.kr

**Min-hwan Oh**
Seoul National University
minoh@snu.ac.kr

## Abstract

In our quest for a reinforcement learning (RL) algorithm that is both practical and provably optimal, we introduce EQO (Exploration via Quasi-Optimism). Unlike existing minimax optimal approaches, EQO avoids reliance on empirical variances and employs a simple bonus term proportional to the inverse of the state-action visit count. Central to EQO is the concept of *quasi-optimism*, where estimated values need not be fully optimistic, allowing for a simpler yet effective exploration strategy. The algorithm achieves the sharpest known regret bound for tabular RL under the mildest assumptions, proving that fast convergence can be attained with a practical and computationally efficient approach. Empirical evaluations demonstrate that EQO consistently outperforms existing algorithms in both regret performance and computational efficiency, providing the best of both theoretical soundness and practical effectiveness.

## 1 Introduction

Reinforcement learning (RL) has seen substantial progress in its theoretical foundations, with numerous algorithms achieving minimax optimality (Azar et al., 2017; Zanette & Brunskill, 2019; Dann et al., 2019; Zhang et al., 2020; Li et al., 2021; Zhang et al., 2021a; 2024). These algorithms are often lauded for providing strong regret bounds, theoretically ensuring their provable optimality in worst-case scenarios. However, despite these guarantees, an important question remains: *Can we truly claim that we can solve tabular reinforcement learning problems practically well?* By "solving reinforcement learning practically well," we expect these theoretically sound and optimal algorithms to deliver both favorable theoretical and empirical performance.[1]

Although provably efficient RL algorithms offer regret bounds that are nearly optimal (up to logarithmic or constant factors), they are often designed to handle worst-case scenarios. This focus on worst-case outcomes leads to overly conservative behavior, as these algorithms must construct bonus terms to guarantee optimism under uncertainty. Consequently, they may suffer from practical inefficiencies in more typical scenarios where worst-case conditions may rarely arise.

Empirical evaluations of these minimax optimal algorithms frequently reveal their shortcomings in practice (see Section 5). In fact, many minimax optimal algorithms often underperform compared to algorithms with sub-optimal theoretical guarantees, such as UCRL2 (Jaksch et al., 2010). This suggests that the pursuit of provable optimality may come at the expense of practical performance. This prompts the question: *Is this seeming tradeoff between provable optimality and practicality inherent? Or, can we design an algorithm that attains both provable optimality and superior practical performance simultaneously?*

To address this, we argue that a fundamental shift is needed in the design of provable RL algorithms. The prevailing reliance on empirical variances to construct worst-case confidence bounds—a tech-

---

[1]If one questions the practical relevance of tabular RL methods given the advancements in function approximation (Jin et al., 2020; He et al., 2023; Agarwal et al., 2023), it is important to recognize that many real-world environments remain inherently tabular, lacking feature representations for generalization. In such cases, efficient and practical tabular RL algorithms are not only relevant but essential. Furthermore, even when function approximation is feasible, tabular methods can still play a crucial role. For instance, state-space dependence may persist even under linear function approximation (Lee & Oh, 2024; Hwang & Oh, 2023).

nique employed by all minimax optimal algorithms (Azar et al., 2017; Zanette & Brunskill, 2019; Dann et al., 2019; Zhang et al., 2021a) (see Table 1)—may no longer be the only viable or practical approach. Instead, we propose new methodologies that, while practical, can still be proven efficient to overcome this significant limitation.

In this work, we introduce a tabular reinforcement learning algorithm, EQO, which stands for *Exploration via Quasi-Optimism*. Unlike existing minimax-optimal algorithms, EQO does not rely on empirical variances. While it employs a bonus term for exploration, EQO stands out for its simplicity and practicality. The bonus term is proportional to the inverse of the state-action visit count, avoiding the dependence on empirical variances that previous approaches rely on.

On the theoretical side, we demonstrate that our proposed algorithm EQO, despite its algorithmic simplicity, achieves the sharpest known regret bound for tabular reinforcement learning. More importantly, this crucial milestone is attained under the mildest assumptions (see Section 4.1). Thus, our results establish that the *fastest convergence to optimality* (the sharpest regret) can be achieved by a *simple and practical* algorithm in the *broadest* (the weakest assumptions) problem settings.

To complement—not as a tradeoff—the theoretical merit, we show that EQO empirically outperforms existing provably efficient algorithms, including previous minimax optimal algorithms. The practical superiority is demonstrated both in terms of regret performance in numerical experiments and computational efficiency. Overall, EQO achieves both minimax optimal regret bounds and superior empirical performance, offering a promising new approach that balances theoretical soundness with practical efficiency.

Our main contributions are summarized as follows:

- We propose a reinforcement learning algorithm, EQO (Algorithm 1), which introduces a distinct exploration strategy. While previous minimax optimal algorithms rely on empirical-variance-based bonus terms, EQO employs a simpler bonus term of the form $c/N(s, a)$, where $c$ is a constant and $N(s, a)$ is the visit count of the state-action pair $(s, a)$. This straightforward yet effective approach reduces computational complexity while maintaining robust exploration, making EQO both practical and theoretically sound. Additionally, this simplicity allows for convenient control of the algorithm through a single parameter, making it particularly advantageous in practice where parameter tuning is essential.

- Our algorithm achieves the tightest regret bound in the literature for tabular reinforcement learning. Even compared to the state-of-the-art bounds by Zhang et al. (2021a), EQO provides sharper logarithmic factors, establishing it as the algorithm with the most efficient regret bound to date. Our novel analysis introduces the concept of *quasi-optimism* (see Section 4.4.2), where estimated values need not be *fully* optimistic.[2] This relaxation simplifies the bonus term and simultaneously controls the amount of underestimation, ultimately leading to a sharper regret bound.

- A key strength of our approach is that it operates under weaker assumptions than the previous assumptions in the existing literature, making it applicable to a broader range of problems (see Section 4.1). While prior work assumes bounded returns for every episode, we relax this condition to require only the value function (i.e., the expected return) to be bounded. This relaxation broadens the applicability of our algorithm.

- We show that EQO enjoys tight sample complexity bounds in terms of mistake-style PAC guarantees and best-policy identification tasks (see Section 2.1 for detailed definitions). This further highlights our proposed algorithm's robust performance.

- We perform numerical experiments that demonstrate that EQO empirically outperforms existing provably efficient algorithms, including prior minimax-optimal approaches. The superiority of EQO is evident both in its regret performance and computational efficiency, showcasing its ability to attain theoretical guarantees with practical performance.

---

[2]By *fully optimistic*, we refer to the conventional UCB-type estimates that lie above the optimal value with high probability. See the distinction with our new *quasi-optimism* in Section 4.4.2.

Table 1: Comparison of minimax optimal algorithms for tabular reinforcement learning under the time-homogeneous setting. Constant and logarithmic factors are omitted. The **Empirical Variance** column indicates whether the algorithm requires empirical variance. The **Boundedness** column shows the quantity on which the boundedness assumptions are imposed, where bound on "Reward" is 1 and bounds on "Return" and "Value" are $H$.

| Paper | Regret Bound | Empirical Variance | Boundedness[†] |
|---|---|---|---|
| Azar et al. (2017) | $H\sqrt{SAK} + H^2S^2A + \sqrt{H^3K}$ | Required | Reward |
| Zanette & Brunskill (2019) | $H\sqrt{SAK} + H^2S^{3/2}A(\sqrt{H}+\sqrt{S})$ | Required | Return |
| Dann et al. (2019) | $H\sqrt{SAK} + H^2S^2A$ | Required | Reward |
| Zhang et al. (2021a) | $H\sqrt{SAK} + HS^2A$ | Required | Return |
| **This work** | $H\sqrt{SAK} + HS^2A$ [*] | Not required | Value |

[*] Our regret bound is the sharpest with more improved logarithmic factors than that of Zhang et al. (2021a).

[†] Bounded reward is the strongest assumption; bounded return is weaker than bounded reward; our bounded value assumption is the weakest among the three conditions (see the discussion in Section 4.1).

## 1.1 RELATED WORK

There has been substantial progress in RL theory over the past decade, with numerous algorithms advancing our understanding of regret minimization and sample complexity (Jaksch et al., 2010; Osband & Roy, 2014; Azar et al., 2017; Dann et al., 2017; Agrawal & Jia, 2017; Ouyang et al., 2017; Jin et al., 2018; Osband et al., 2019; Russo, 2019; Zhang et al., 2020; 2021b) , and some recent lines of work focusing on gap-dependent bounds (Simchowitz & Jamieson, 2019; Dann et al., 2021; Wagenmaker et al., 2022; Tirinzoni et al., 2022; Wagenmaker & Jamieson, 2022). In the remainder of this section, we focus on comparisons with more closely related methods and techniques.

**Regret-Minimizing Algorithms for Tabular RL.** In episodic finite-horizon reinforcement learning, the known regret lower bound is $\Omega(H\sqrt{SAK})$ (Jaksch et al., 2010; Domingues et al., 2021). The first result to achieve a matching upper bound is by Azar et al. (2017). Their algorithm, `UCBVI-BF`, adopts the *optimism in the face of uncertainty* (OFU) principle by adding an optimistic bonus during the estimation. For the sharper regret bound, they use a Bernstein-Freedman type concentration inequality and design a bonus term that utilizes the empirical variance of the estimated values at the next time step. Zanette & Brunskill (2019) show that by estimating both upper and lower bounds of the value function, their algorithm automatically adapts to the hardness of the problem without requiring prior knowledge. Zhang et al. (2021a) improve the previous analysis and reduce the non-leading term to $\widetilde{\mathcal{O}}(HS^2A)$, achieving the regret bound that is independent of the lengths of the episodes when the maximum return per episode is rescaled to 1.

**Time-Inhomogeneous Setting.** There has also been an increasing number of works that focus on time-inhomogeneous MDPs, which have different transition probabilities and rewards at each time step.[3] One active area of study is to design an efficient model-free algorithm, which has a space complexity of $o(HS^2A)$ (Strehl et al., 2006). Jin et al. (2018) propose a variant of Q-learning with a bonus similar to `UCBVI-BF` and show that this model-free algorithm achieves $\mathcal{O}(\sqrt{K})$ regret. However, their regret bound is worse than the lower bound by a factor of $\sqrt{H}$. This additional factor is removed by Zhang et al. (2020), achieving the minimax regret bound for the time-inhomogeneous setting with a model-free algorithm for the first time. Li et al. (2021) further improve the non-leading term and achieve a regret bound of $\widetilde{\mathcal{O}}(H^{3/2}\sqrt{SAK}+H^6SA)$. For model-based algorithms, the non-leading terms are further optimized. Ménard et al. (2021b) combine the Q-learning approach with momentum, achieving a non-leading term of $\widetilde{\mathcal{O}}(H^4SA)$. Recent work by Zhang et al. (2024) further reduce it to $\widetilde{\mathcal{O}}(H^2SA)$, resulting in the bound of $\widetilde{\mathcal{O}}(\min\{H^{3/2}\sqrt{SAK}, HK\})$ on the whole range

---

[3]The regret lower bound for the time-inhomogeneous case is $\Omega(H^{3/2}\sqrt{SAK})$ (Domingues et al., 2021), being worse than the time-homogeneous case by a factor of $\sqrt{H}$. Due to this sub-optimality, the time-inhomogeneous setting is often viewed as a special case of the time-homogeneous setting with $HS$ states.

of $K$. Deviating from the algorithmic framework shared by these works, Tiapkin et al. (2022) propose a posterior-sampling algorithm and achieve the minimax bound without computing empirical variances.

**PAC Bounds.** Dann & Brunskill (2015) present a PAC upper bound of $\widetilde{\mathcal{O}}(\frac{H^2 S^2 A}{\varepsilon^2})$ and a lower bound of $\Omega(\frac{H^2 S A}{\varepsilon^2})$, where their focus is on what is later named as the mistake-style PAC bound. Dann et al. (2017) generalize the concept to uniform-PAC, which also implies a high-probability cumulative regret bound as well. Dann et al. (2019) propose a further generalized framework named *Mistake-IPOC*, which encompasses uniform-PAC, best-policy identification, and an anytime cumulative regret bound. Notably, their algorithm achieves the minimax PAC bounds for the first time. Other PAC tasks include best-policy identification (BPI) (Fiechter, 1994; Domingues et al., 2021; Kaufmann et al., 2021), where the goal is to return a policy whose sub-optimality is small with high probability, and reward-free exploration (Kaufmann et al., 2021; Ménard et al., 2021a), where the goal is similar to BPI, but the agent does not receive reward feedback while exploring.

$\mathcal{O}(1/N)$**-bonus Exploration.** To the best of our knowledge, our algorithm is the first to use an exploration bonus of the form $c/N$ for the reinforcement learning setting and attain regret guarantees. In the multi-armed bandit setting, Simchi-Levi et al. (2023; 2024) utilize a similar bonus term, but the underlying motivations and derivations differ significantly. Their focus is on controlling the tail probability of regret—that is, minimizing the probability of observing large regret. Their bonus term arises from Hoeffding's inequality with a specific failure probability. In contrast, our work is aimed at developing a practical algorithm for tabular reinforcement learning. Our bonus term stems from a distinct context—decoupling the variance factors and visit counts that naturally arise in reinforcement learning settings when applying Freedman's inequality (see Section 4.4). Importantly, Simchi-Levi et al. (2023; 2024) do not appear to leverage Freedman's inequality, either directly or indirectly, in their derivations. The distinction becomes evident when comparing the failure probabilities of the inequalities.

## 2 PRELIMINARIES

### 2.1 PROBLEM SETTING

We consider a finite-horizon time-homogeneous Markov decision process (MDP) $\mathcal{M} = (\mathcal{S}, \mathcal{A}, P, r, H)$, where $\mathcal{S}$ is the state space, $\mathcal{A}$ is the action space, $P : \mathcal{S} \times \mathcal{A} \to \Delta\mathcal{S}$ is the state transition distribution, $r : \mathcal{S} \times \mathcal{A} \to \mathbb{R}$ is the reward function, and $H \in \mathbb{N}$ is the time horizon of an episode. We focus on tabular MDPs, where the cardinalities of the state and action spaces are finite and denoted by $S := |\mathcal{S}|$ and $A := |\mathcal{A}|$. The agent and the environment interact for a sequence of episodes. The interaction of the $k$-th episode begins with the environment providing an initial state, $s_1^k \in \mathcal{S}$. For time steps $h = 1, \ldots, H$, the agent chooses an action $a_h^k \in \mathcal{A}$, then receives a random reward $R_h^k \in \mathbb{R}$ and the next state $s_{h+1}^k \in \mathcal{S}$ from the environment. The mean of the random reward is $r(s_h^k, a_h^k)$ and the next state is independently sampled from $P(\cdot \mid s_h^k, a_h^k)$. These probability distributions are unknown to the agent. The goal of the agent is to maximize the total rewards.

A policy is a sequence of $H$ functions $\pi = \{\pi_h\}_{h=1}^H$ with $\pi_h : \mathcal{S} \to \mathcal{A}$ for all $h$. An agent following a policy $\pi$ chooses action $a = \pi_h(s)$ at time step $h$ when the current state is $s$. We define the value function of policy $\pi$ at time step $h$ as $V_h^\pi(s) := \mathbb{E}_{\pi(\cdot|s_h=s)}\big[\sum_{j=h}^H r(s_j, a_j)\big]$, where $\mathbb{E}_{\pi(\cdot|s_h=s)}$ denotes the expectation over $(s_h = s, a_h, \ldots, s_H, a_H, s_{H+1})$ with $a_j = \pi_j(s_j)$ and $s_{j+1} \sim P(\cdot|s_j, a_j)$ for $j = h, \ldots, H$. Similarly, we define the action-value function as $Q_h(s, a) := \mathbb{E}_{\pi(\cdot|s_h=s,a_h=a)}\big[\sum_{j=h}^H r(s_j, a_j)\big]$. We set $V_{H+1}^\pi(s) = 0$ for all $\pi$ and $s \in \mathcal{S}$. $\pi^*$ is the optimal policy, which chooses the action that maximizes the expected return at every time step, and it holds that $V_h^{\pi^*}(s) = \sup_\pi V_h^\pi(s)$ for all $h = 1, \ldots, H$ and $s \in \mathcal{S}$. We denote $V_h^{\pi^*}$ by $V_h^*$, and call it the optimal value function. The regret of a policy $\pi$ is defined as $V_1^*(s_1) - V_1^\pi(s_1)$. The agent's goal is to find policies that minimize cumulative regret for a given MDP. The cumulative regret over $K$ episodes is defined as:

$$\text{Regret}(K) := \sum_{k=1}^K (V_1^*(s_1^k) - V_1^{\pi^k}(s_1^k)).$$

---

**Algorithm 1:** EQO (Exploration via Quasi-Optimism)

   **Input:** $\{c_k\}_{k=1}^{\infty}$

1  **for** $k = 1, 2, \ldots, K$ **do**

2     **foreach** $(s, a, s') \in \mathcal{S} \times \mathcal{A} \times \mathcal{S}$ **do**

3         $N^k(s, a) \leftarrow \sum_{i=1}^{k-1} \sum_{h=1}^{H} \mathbb{1}\{(s_h^i, a_h^i) = (s, a)\};$

4         $\hat{r}^k(s, a) \leftarrow \frac{1}{N^k(s,a)} \sum_{i=1}^{k-1} \sum_{h=1}^{H} R_h^i \mathbb{1}\{(s_h^i, a_h^i) = (s, a)\};$

5         $\hat{P}^k(s'|s, a) \leftarrow \frac{1}{N^k(s,a)} \sum_{i=1}^{k-1} \sum_{h=1}^{H} \mathbb{1}\{(s_h^i, a_h^i, s_{h+1}^i) = (s, a, s')\};$

6         $V_{H+1}^k(s) \leftarrow 0;$

7     **for** $h = H, H-1, \ldots, 1$ **do**

8         **foreach** $(s, a) \in \mathcal{S} \times \mathcal{A}$ **do**

9             $b^k(s, a) \leftarrow c_k / N^k(s, a);$

10            $Q_h^k(s, a) \leftarrow \begin{cases} \min\left\{\hat{r}^k(s,a) + b^k(s,a) + \hat{P}^k V_{h+1}^k(s, a), H\right\} & \text{if } N^k(s,a) > 0 \\ H & \text{if } N^k(s,a) = 0 \end{cases};$

11         $V_h^k(s) \leftarrow \max_{a \in \mathcal{A}} Q_h^k(s, a)$ for all $s \in \mathcal{S};$

12         $\pi_h^k(s) \leftarrow \text{argmax}_{a \in \mathcal{A}} Q_h^k(s, a)$ for all $s \in \mathcal{S};$

13     Execute $\pi^k$ and obtain $\tau^k = (s_1^k, a_1^k, R_1^k, \ldots, s_H^k, a_H^k, R_H^k, s_{H+1}^k);$

---

In addition to regret, another important measure of performance is the PAC (probably approximately correct) bound (Kakade, 2003), also referred to as sample complexity. This measure focuses on obtaining a policy (after sufficient exploration) whose regret is no more than $\varepsilon$ with probability at least $1 - \delta$, for given values of $\varepsilon > 0$ and $\delta \in (0, 1]$. A policy $\pi$ is said to be *$\varepsilon$-optimal* if its regret satisfies $V_1^*(s_1) - V_1^\pi(s_1) \leq \varepsilon$. We evaluate two different tasks using PAC bounds: (i) the mistake-style PAC, which aims to minimize the number of episodes in which the agent executes a policy that is not $\varepsilon$-optimal, and (ii) best-policy identification, whose objective is to return an $\varepsilon$-optimal policy within the fewest possible episodes.

### 2.2 NOTATIONS

$\mathbb{N} = \{1, 2, \ldots\}$ is the set of natural numbers. For $N \in \mathbb{N}$, we define $[N] := \{1, \ldots, N\}$. $\mathbb{1}\{E\}$ is the indicator function that takes the value 1 when $E$ is true and 0 otherwise. For any function $V : \mathcal{S} \to \mathbb{R}$ and a state-action pair $(s, a) \in \mathcal{S} \times \mathcal{A}$, we denote the mean of $V$ under the probability distribution $P(\cdot|s, a)$ by $PV(s, a) := \sum_{s' \in \mathcal{S}} P(s'|s, a)V(s')$. For any other function $\hat{P} : \mathcal{S} \times \mathcal{A} \to \mathbb{R}^S$, we define $\hat{P}V(s, a) := \sum_{s' \in \mathcal{S}} \hat{P}(s'|s, a)V(s')$ in the same manner. We denote the variance of $V$ under $P(\cdot|s, a)$ by $\text{Var}(V)(s, a) := \sum_{s' \in \mathcal{S}} P(s'|s, a)(V(s') - PV(s, a))^2$. A tuple $\tau = (s_1, a_1, R_1, \ldots, s_H, a_H, R_H, s_{H+1})$ generated by a single episode of interaction is called a trajectory. Let $\tau^k$ be the trajectory of the $k$-th episode. For $h \in [H]$, we also define the *partial trajectory* as $\tau_h^k := (s_1^k, a_1^k, R_1^k, \ldots, s_h^k, a_h^k)$. For all $h \in [H]$ and $k \in \mathbb{N}$, let $\mathcal{F}_h^k = \sigma(\{\tau^i\}_{i=1}^{k-1} \cup \{\tau_h^k\})$ be the $\sigma$-algebra generated by the agent-environment interactions up to the action $a_h^k$ taken at the $h$-th time step of the $k$-th episode. For convenience, we define $\mathcal{F}_{H+1}^k$ as $\sigma(\{\tau^i\}_{i=1}^k)$.

## 3 ALGORITHM: EQO

We introduce our algorithm, *Exploration via Quasi-Optimism* (EQO), which presents a distinct approach to bonus construction compared to prior optimism-based methods. While the framework of our algorithm shares some structural similarities with UCBVI (Azar et al., 2017), which has been widely adopted by several subsequent works (Zanette & Brunskill, 2019; Dann et al., 2019; Zhang et al., 2021a), EQO diverges significantly in its exploration strategy. The key novelty lies in its bonus term, which does not rely on empirical variances, unlike the previous methods. Instead, EQO takes a sequence of real numbers $\{c_k\}_{k=1}^{\infty}$ as input, and the bonus for a state-action pair $(s, a) \in \mathcal{S} \times \mathcal{A}$ in episode $k$ is simply $c_k / N^k(s, a)$, where $N^k(s, a)$ is the visit count of $(s, a)$ up to the previous episode. This simplicity stands in contrast to the empirical-variance-based bonuses used in prior al-

gorithms, demonstrating that empirical variance (and UCB approaches based on estimated variance) is not necessary for achieving efficient exploration in our approach.

A notable advantage of our algorithm is its simplicity in practice. While many existing RL algorithms often involve multiple parameters with complex dependencies, our approach consolidates these into a single parameter, $c_k$, making tuning much more straightforward.[4]

# 4 THEORETICAL GUARANTEES

## 4.1 ASSUMPTIONS

Before presenting our theoretical guarantees, we provide the regularity assumptions necessary for the analysis. We emphasize that our assumptions are weaker than those in the previous RL literature.

**Assumption 1** (Boundedness). $0 \leq V_h^*(s) \leq H$ *holds for all* $s \in \mathcal{S}$ *and* $h \in [H]$, *and* $0 \leq R_h^k \leq H$ *holds for all* $h \in [H]$ *and* $k \in \mathbb{N}$.

**Assumption 2** (Adaptive random reward). $\mathbb{E}[R_h^k|\mathcal{F}_h^k] = r(s_h^k, a_h^k)$ *holds for all* $h \in [H]$ *and* $k \in \mathbb{N}$.

Assumption 1 regularizes the scaling of the problem instances. The most widely used regularity assumption is that the random rewards lie within the interval $[0, 1]$ for all time steps (Jaksch et al., 2010; Azar et al., 2017; Dann et al., 2019). A slightly generalized version assumes that the return, defined as the total reward of an episode, is bounded as $0 \leq \sum_{h=1}^{H} R_h \leq H$, and that each random reward is non-negative (Jiang & Agarwal, 2018; Zanette & Brunskill, 2019; Zhang et al., 2021a). Such an assumption allows non-uniform reward schemes. For instance, the agent may receive a reward of $H$ at exactly one time step and no rewards at the other time steps. We further relax this boundedness assumption by constraining only the optimal values $V_h^*(s)$ to be within the interval $[0, H]$, along with the conventional boundedness on the random rewards within $[0, H]$. Since the value function is the expected return, *our bounded value condition is weaker than the bounded return assumption* (and hence, also weaker than the widely used uniform boundedness of rewards) used in the previous literature.

Assumption 2 allows martingale-style random rewards. Standard MDPs assume a fixed reward distribution for each state-action pair, where rewards are sampled independently of history and the next state. Some recent studies introduce a joint probability distribution on the next state and the reward, defined as $p : \mathcal{S} \times \mathcal{A} \to \Delta(\mathcal{S} \times \mathbb{R})$, so that $(s_{h+1}, R_h) \overset{i.i.d.}{\sim} p(s_h, a_h)$ (Krishnamurthy et al., 2016; Sutton, 2018). We further weaken this assumption by requiring only that the mean of the random reward equals $r(s, a)$, allowing specific distributions to adapt to the history. Note that in Assumption 2, $s_{h+1}^k$ is not part of $\mathcal{F}_h^k$. This fact allows dependence between $s_{h+1}^k$ and $R_h^k$, making our assumption more general.

## 4.2 REGRET BOUND

We now present the regret upper bounds enjoyed by our algorithm EQO (Algorithm 1).

**Theorem 1** (Regret bound of EQO). *Fix* $\delta \in (0, 1]$. *Suppose the number of episodes, denoted by* $K$, *is known to the agent. Let* $c := \max\{7H\ell_1, 1.4H\sqrt{K\ell_1/(SA\ell_{2,K})}\}$, *where* $\ell_1 = \log \frac{24HSA}{\delta}$ *and* $\ell_{2,K} = \log(1 + KH/(SA))$. *If Algorithm 1 is run with* $c_k = c$ *for all* $k \in [K]$, *then with probability at least* $1 - \delta$, *the cumulative regret of* $K$ *episodes is bounded as follows:*

$$Regret(K) \leq 38H\sqrt{SAK\ell_1\ell_{2,K}} + 256HS^2A\ell_{1,K}'(1 + \ell_{2,K}),$$

*where* $\ell_{1,K}' = \log(50HSA(\log(eKH))^2/\delta)$.

When the number of episodes $K$ is specified, Theorem 1 states that the input of Algorithm 1, $\{c_k\}_{k=1}^{K}$, can be set as a constant independent of $k$, making the algorithm even simpler. In case where $K$ is unknown, it is possible to attain a regret bound that holds for all $K \in \mathbb{N}$ by updating $c_k$ in a doubling-trick styled fashion. Theorem 2 states the anytime regret bound result enjoyed by Algorithm 1. Note that resetting the algorithm is not necessary, unlike the actual doubling trick.

---

[4] The theoretical results in Theorems 1, 3, and 4 justify setting $c_k$ as a $k$-independent constant, offering both theoretical and practical convenience.

**Theorem 2** (Anytime regret bound of EQO). *Fix $\delta \in (0, 1]$. For any episode $k \in \mathbb{N}$, take $c_k = \max\{7H\ell_{1,k}, 1.4H\sqrt{k_2\ell_{1,k}/(SA\ell_{2,k_2})}\}$, where $k_2 = 2^{\lfloor \log_2 k \rfloor}$, $\ell_{1,k} = \log \frac{24HSA(1+\lfloor \log_2 k \rfloor)^2}{\delta}$, and $\ell_{2,k_2} = \log(1 + k_2H/(SA))$. If Algorithm 1 is run with $c_k$ as defined above, then with probability at least $1 - \delta$, the cumulative regret of $K$ episodes for any $K \in \mathbb{N}$ is bounded as follows:*

$$Regret(K) \leq 75H\sqrt{SAK\ell_{1,K}\ell_{2,K}} + 256HS^2A\ell'_{1,K}(1 + \ell_{2,K}),$$

*where $\ell'_{1,K} = \log(50HSA(\log(eKH))^2/\delta)$.*

**Discussions of Theorems 1 and 2.** We discuss the regret bounds of both Theorems 1 and 2. The first terms of the regret bounds are in $\widetilde{\mathcal{O}}(H\sqrt{SAK})$, which matches the lower bound up to logarithmic factors. In fact, the logarithmic factors of Theorems 1 and 2 are $\mathcal{O}\left(\sqrt{\log \frac{HSA}{\delta} \log(KH)}\right)$ and $\mathcal{O}\left(\sqrt{\log \frac{HSA(\log K)}{\delta} \log(KH)}\right)$ respectively, which are even tighter than the state-of-the-art guarantee in Zhang et al. (2021a). The second terms of the regret bounds are $\widetilde{\mathcal{O}}(HS^2A)$, which implies that our algorithm matches the lower bound for $K \geq S^3A$. This bound matches the previously best non-leading term in the time-homogeneous setting by Zhang et al. (2021a) even in the logarithmic factors. Therefore, our regret bounds are *the tightest compared to all the previous results* in the time-homogeneous setting up to constant factors. Furthermore, to the best of our knowledge, our result is *the first to prove that the minimax regret bound is achievable under the weakest boundedness assumption on the value function*.

## 4.3 PAC BOUNDS

We demonstrate that by setting the parameters $c_k$ appropriately, Algorithm 1 achieves PAC bounds.

**Theorem 3** (Mistake-style PAC bound). *Let $\varepsilon \in (0, H]$ and $\delta \in (0, 1]$. Suppose Algorithm 1 is run with $c_k = \frac{56H^2\ell_1}{\varepsilon}$ for all $k \in \mathbb{N}$, where $\ell_1 = \log \frac{24HSA}{\delta}$. Then, with probability at least $1 - \delta$, the number of episodes in which the algorithm executes policies that are not $\varepsilon$-optimal is at most $K_0$, where $K_0 = \widetilde{\mathcal{O}}((\frac{H^2SA}{\varepsilon^2} + \frac{HS^2A}{\varepsilon})\log \frac{1}{\delta})$.*

In Appendix D, we present $(\varepsilon, \delta)$-EQO (Algorithm 2), which runs Algorithm 1 with parameters specified as in Theorem 3, then performs an additional procedure to certify $\varepsilon$-optimal policies. With this extension, our algorithm is capable of solving the BPI task with the same bound as in the mistake-style PAC bounds.

**Theorem 4** (Best-policy identification). *Let $\varepsilon \in (0, H]$ and $\delta \in (0, 1]$. Algorithm 2 provides an $\varepsilon$-optimal policy within $K_0 + 1$ episodes, where $K_0$ takes the same value as in Theorem 3.*

For $\varepsilon < H/S$, the bound $K_0$ is $\widetilde{\mathcal{O}}(H^2SA(\log \frac{1}{\delta})/\varepsilon^2)$, which matches the lower bounds for both tasks (Domingues et al., 2021). For both tasks, our results exhibit the tightest non-leading term compared to the previous results. For detailed discussions and proofs of Theorems 3 and 4, refer to Appendix D.

## 4.4 SKETCH OF REGRET ANALYSIS

In this subsection, we provide a sketch of proofs of Theorem 1 and Theorem 2. For simplicity, we denote all logarithmic factors by $\ell$ in this subsection. The full statements of the proposition and lemmas with specific logarithmic terms and their detailed proofs are presented in Appendix C.

We propose Proposition 1 that demonstrates the effect of the bonus terms on the cumulative regret. In this proposition, we introduce an auxiliary sequence of positive real numbers $\{\lambda_k\}_{k=1}^{\infty}$ and set $c_k = 7H\ell/\lambda_k$.

**Proposition 1.** *Let $\{\lambda_k\}_{k=1}^{\infty}$ be a sequence of non-increasing positive real numbers with $\lambda_1 \leq 1$. Suppose Algorithm 1 is run with $c_k = 7H\ell/\lambda_k$ for all $k \in \mathbb{N}$. Then, with probability at least $1 - \delta$, the cumulative regret of $K$ episodes for any $K \in \mathbb{N}$ is bounded as follows:*

$$\sum_{k=1}^{K} Regret(K) \leq 4H\sum_{k=1}^{K}\lambda_k + \frac{88}{\lambda_K}HSA\ell^2 + 168HS^2A\ell^2.$$

Proposition 1 demonstrates that the exploration-exploitation trade-off can be balanced by the parameter $\lambda_k$. The term $\sum_{k=1}^{K} \lambda_k$ represents the regret incurred due to the estimation error, while the term proportional to $1/\lambda_K$ represents the regret incurred from exploration. For example, if the values of $\{\lambda_k\}_{k=1}^{K}$ are large, the algorithm runs with smaller bonuses. This reduces the regret caused by excessive exploration, but the algorithm may exploit sub-optimal policies due to a lack of information, contributing to a larger $\sum_{k=1}^{K} \lambda_k$ term. Both Theorem 1 and Theorem 2 follow from Proposition 1 by setting appropriate values for $\lambda_k$. For Theorem 1, we set $\lambda_k = \min\{1, 5\sqrt{SA\ell^2/K}\}$ for all $k \in [K]$. For Theorem 2, we set $\lambda_k = \min\{1, 5\sqrt{SA\ell^2/k_2}\}$, where $k_2 = 2^{\lfloor \log_2 k \rfloor}$. The proof of Proposition 1 is sketched throughout the following subsections.

### 4.4.1 HIGH-PROBABILITY EVENT

We denote the high-probability event under which Proposition 1 holds by $\mathcal{E}$. $\mathcal{E}$ is defined as an intersection of six high-probability events, including concentration events of transition model estimation and reward model estimation. Refer to Appendix B for the specific events that constitute $\mathcal{E}$ and the proofs that the events happen with high probability.

Although our algorithm does not use empirical variances, all the concentration results in the analysis are based on Freedman's inequality (Freedman, 1975). The following lemma is a variant of the inequality that we use frequently throughout the analysis. While the current presentation focuses on i.i.d. sequences, it is also applicable to martingales, as shown in Lemma 36 of Appendix F.

**Lemma 1.** *Let $C > 0$ be a constant and $\{X_t\}_{t=1}^{\infty}$ be i.i.d. copies of a random variable $X$ with $X \leq C$. Then, for any $\lambda \in (0, 1]$ and $\delta \in (0, 1]$, the following inequality holds for all $n \in \mathbb{N}$ with probability at least $1 - \delta$:*

$$\frac{1}{n}\sum_{t=1}^{n} X_t \leq \frac{3\lambda}{4C}\operatorname{Var}(X) + \frac{C}{\lambda n}\log\frac{1}{\delta}.$$

One advantage of this form is that the variance term and the $1/n$ term are isolated, whereas the previous Bernstein-type bound includes a term of the form $\sqrt{\operatorname{Var}/n}$. While the sum of the variances achieves a tight bound within the expectation, the $1/n$ terms must be summed according to actual visit counts. This discrepancy necessitates the use of multiple concentration inequalities, alternating between the expected and sampled trajectories when bounding the sum of $\sqrt{\operatorname{Var}/n}$ terms. However, Lemma 1 allows us to address the two factors independently. Refer to Appendix F for more details about Freedman's inequality and its derivatives we utilize.

### 4.4.2 QUASI-OPTIMISM

Optimism-based analysis begins by showing $V_h^k(s) \geq V_h^*(s)$ for all $s, h$, where the use of empirical variances plays a crucial role (Azar et al., 2017; Jin et al., 2018; Zanette & Brunskill, 2019; Dann et al., 2019; Zhang et al., 2021a; 2024). However, our bonus term does not contain any empirical variances. In fact, our bonus term does not guarantee optimism or even probabilistic optimism. Instead, it guarantees what we name *quasi-optimism*, meaning that the estimated values are almost optimistic. Specifically, the estimated values need to be increased by a constant amount to ensure they exceed the optimal values. In other words, the estimation may be less than the optimal value, but only by a bounded amount. We formally present our result in Lemma 2.

**Lemma 2** (Quasi-optimism). *Under $\mathcal{E}$, it holds that for all $s \in \mathcal{S}$, $h \in [H+1]$, $k \in \mathbb{N}$,*

$$V_h^k(s) + \frac{3}{2}\lambda_k H \geq V_h^*(s).$$

We outline the main ideas behind quasi-optimism. Fix $h \in [H]$, $s \in \mathcal{S}$, and $k \in \mathbb{N}$. For ease of presentation, we assume that the reward function is known and that $V_h^k(s) < H$. For $a^* := \pi_h^*(s)$, we have $V_h^*(s) = r(s, a^*) + PV_{h+1}^*(s, a^*)$ by the Bellman equation and $V_h^k(s) \geq Q_h^k(s, a^*) = r(s, a^*) + b^k(s, a^*) + \hat{P}^k V_h^k(s, a^*)$ by the definitions of $V_h^k$ and $Q_h^k$. Therefore, we obtain that

$$V_h^*(s) - V_h^k(s) \leq PV_{h+1}^*(s, a^*) - b^k(s, a^*) - \hat{P}^k V_h^k(s, a^*)$$

$$= -b^k(s, a^*) + \underbrace{(P - \hat{P}^k)V_{h+1}^*(s, a^*)}_{I_1} + \underbrace{\hat{P}^k(V_{h+1}^* - V_{h+1}^k)(s, a^*)}_{I_2}. \tag{1}$$

In the previous method of guaranteeing full optimism, one assumes $I_2 \leq 0$ using mathematical induction and then faces the challenging task of fully bounding $I_1$ by $b^k(s, a^*)$. In the proof of Lemma 2, we set a slightly relaxed induction hypothesis. As a result, $I_2$ may be greater than zero, while $b^k(s, a^*)$ no longer needs to fully bound $I_1$. The key to quasi-optimism is to *allow underestimation of $I_1$, while controlling the resulting increase in $I_2$*. We explain each concept in detail.

Applying Lemma 1 to $V_{h+1}^*(s')$ with $s' \sim P(\cdot|s, a^*)$, we obtain the following inequality (Lemma 5):

$$\left| (\hat{P}^k - P)V_{h+1}^*(s, a^*) \right| \leq \frac{\lambda_k}{4H} \text{Var}(V_{h+1}^*)(s, a^*) + \frac{3H\ell}{\lambda_k N^k(s, a^*)}.$$

We set $b^k(s, a^*) = \frac{3H\ell}{\lambda_k N^k(s, a^*)}$ to compensate for the $1/N$ term, but leave the variance term. Then, we obtain a recurrence relation of

$$V_h^*(s) - V_h^k(s) \leq \frac{\lambda_k}{4H} \text{Var}(V_{h+1}^*)(s, a^*) + \hat{P}^k(V_{h+1}^* - V_{h+1}^k)(s, a^*). \tag{2}$$

We use backward induction on $h$ to obtain a closed-form bound for $V_h^*(s) - V_h^k(s)$, specifically, $V_h^*(s) - V_h^k(s) \leq W_h(s)$ for some functions $\{W_h\}_{h=1}^H$. To infer what $W$ should look like, we consider the case where the recurrence term is based on $P$ instead of $\hat{P}^k$.

If we had $V_h^*(s) - V_h^k(s) \leq \frac{\lambda_k}{4H} \text{Var}(V_{h+1}^*)(s, a^*) + P(V_{h+1}^* - V_{h+1}^k)(s, a^*)$, where $\hat{P}^k$ in Eq. (2) is replaced with $P$, by iteratively expanding the $V_{h+1}^* - V_{h+1}^k$ part, we observe that the expected sum of the variance terms along a trajectory serves as an upper bound for $V_h^*(s) - V_h^k(s)$, that is, $V_h^*(s) - V_h^k(s) \leq \frac{\lambda_k}{4H} \mathbb{E}_{\pi^*(\cdot|s_h=s)}[\sum_{j=h}^H \text{Var}(V_{j+1}^*)(s_j, a_j)]$. This sum has a non-trivial bound of $H^2$, and this fact has been frequently exploited to achieve better $H$-dependency in the regret bound since Azar et al. (2017). However, it has not been used for showing optimism, and furthermore, this observation relies on the boundedness of returns (see, for example, Eq. (26) of Azar et al. (2017)). We derive a novel way of bounding the sum of variances without requiring such a condition, which is applicable to showing quasi-optimism. We first present the following difference-type bound for the variance (Lemma 27):

$$\text{Var}(V_{h+1}^*)(s, a^*) \leq 2HV_h^*(s) - (V_h^*)^2(s) - P(2HV_{h+1}^* - (V_{h+1}^*)^2)(s, a^*).$$

Using this inequality and mathematical induction, one can show that the expected sum of variances is bounded by $2HV_h^*(s) - (V_h^*)^2(s)$, which is at most $H^2$. Then, the recurrence relation with $P$ instead of $\hat{P}^k$ implies $V_h^*(s) - V_H^k(s) \leq \frac{\lambda_k}{4H}(2HV_h^*(s) - (V_h^*)^2(s))$.

Now, we deal with the original recurrence relation Eq. (2), where a technical approach is required to handle the dependence on $\hat{P}^k$. Recall that we aim to find functions $\{W_h\}_{h=1}^H$ that satisfy $V_h^*(s) - V_h^k(s) \leq W_h(s)$ under Eq. (2). Assuming an induction hypothesis $V_{h+1}^*(s) - V_{h+1}^k(s) \leq W_{h+1}(s)$ for all $s \in \mathcal{S}$, we bound $\hat{P}^k(V_{h+1}^* - V_{h+1}^k)(s, a^*)$ as follows:

$$\hat{P}^k(V_{h+1}^* - V_{h+1}^k)(s, a^*) \leq \hat{P}^k W_{h+1}(s, a^*) = (\hat{P}^k - P)W_{h+1}(s, a^*) + PW_{h+1}(s, a^*).$$

We see that $W_h(s)$ must bound not only the sum of the variance terms but also an additional error term $(\hat{P}^k - P)W_{h+1}(s, a^*)$. The demonstration above suggests setting $W_{h+1}(s) = \frac{\lambda_k}{H}(c_1 HV_{h+1}^*(s) - c_2(V_{h+1}^*)^2(s))$ for some constants $c_1$ and $c_2$. Then, since $W_{h+1}$ is a function of $V_{h+1}^*$, applying Freedman's inequality to the error term results in a $\text{Var}(V_{h+1}^*)(s, a^*)$-related term and a $1/N^k(s, a^*)$ term. The $1/N$ term is compensated by increasing $b^k(s, a^*)$ and the variance term is merged into the variance term that is already present in the recurrence relation. Then, only $PW_{h+1}(s, a^*)$ remains, and we apply the method of bounding the sum of variances explained earlier. Through some technical calculations, we show that the induction argument becomes valid with $c_1 = 2$ and $c_2 = 1/2$. Then, we have $V_h^*(s) - V_h^k(s) \leq W_h(s) \leq \frac{3}{2}\lambda_k H$ for all $s, h$, leading to Lemma 2. The full demonstration of the induction step is presented in Appendix C.1.

### 4.4.3 BOUNDING THE CUMULATIVE REGRET

We bound $V_1^k(s_1^k) - V_1^{\pi^k}(s_1^k)$, the amount of overestimation with respect to the true value function. Similarly to the previous section, we use Freedman's inequality and bound the amount of overestimation at a single time step by the sum of a variance term and a term proportional to $1/N^k(s, a)$. We denote the latter by $\beta^k(s, a) := \frac{1}{N^k(s, a)}(\frac{11H\ell}{\lambda_k} + 21HS\ell)$. As in the previous section, the expected sum of the variance terms is bounded by $\lambda_k H$. Therefore, the amount of overestimation is

bounded by the sum of $\lambda_k H$ and the expected sum of $\beta^k$. We define $U_h^k(s)$ as the sum of $\beta^k$ along a trajectory that follows $\pi^k$ starting from state $s$ at time step $h$ with appropriate clipping. Specifically, let $U_{H+1}^k(s) := 0$ for all $s \in \mathcal{S}$, then define $U_h^k(s)$ for $h = H, H-1, \dots, 1$ iteratively as follows:

$$U_h^k(s) := \min\{\beta^k(s, \pi_h^k(s)) + PU_{h+1}^k(s, \pi_h^k(s)), H\}.$$

The next lemma states that the amount of overestimation is bounded by the sum of $\lambda_k H$ and $U_h^k$.

**Lemma 3.** *Under* $\mathcal{E}$, $V_h^k(s) - V_h^{\pi^k}(s) \leq \frac{5}{2}\lambda_k H + 2U_h^k(s)$ *holds for all* $s \in \mathcal{S}$, $h \in [H+1]$, *and* $k \in \mathbb{N}$.

By $\sum_{n=1}^N 1/n \leq 1 + \log N$, the sum of $1/N^k(s,a)$ is well-controlled when the sum is taken over the sampled trajectories. Using concentration results between the expected and sampled trajectories, we derive the following bound for the sum of $U_1^k$:

**Lemma 4.** *Under* $\mathcal{E}$, *it holds that* $\sum_{k=1}^K U_1^k(s_1^k) \leq \frac{44}{\lambda_K} HSA\ell^2 + 84HS^2A\ell^2$ *for all* $K \in \mathbb{N}$.

The detailed proofs of Lemma 3 and Lemma 4 can be found in Appendix C.2 and Appendix C.3, respectively. Proposition 1 is proved by combining Lemmas 2 to 4.

## 5 EXPERIMENTS

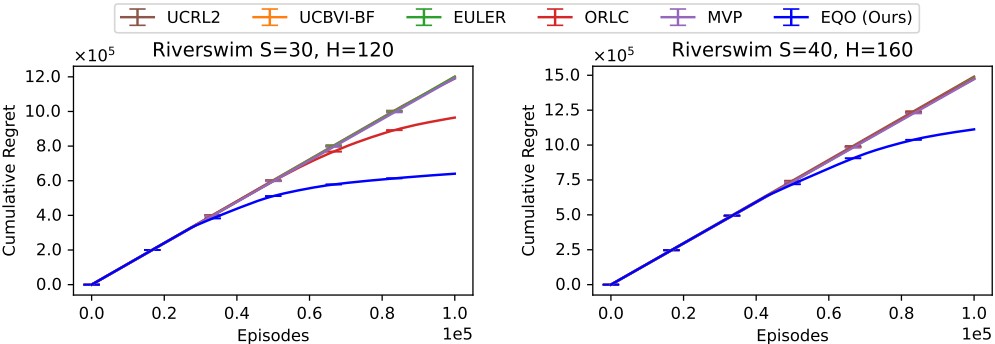

Figure 1: Cumulative regret under RiverSwim MDP with varying $S$ and $H$.

We perform numerical experiments to compare the empirical performance of algorithms for tabular reinforcement learning. We consider the standard MDP named RiverSwim (Strehl & Littman, 2008; Osband et al., 2013), which is known to be a challenging environment that requires strategic exploration. We compare our algorithm EQO with previous algorithms, UCRL2 (Jaksch et al., 2010), UCBVI-BF (Azar et al., 2017), EULER (Zanette & Brunskill, 2019), ORLC (Dann et al., 2019), and MVP (Zhang et al., 2021a). We run the algorithms on the RiverSwim MDP with various configurations of $S$ and $H$. The results for $S = 30$ and $S = 40$ are presented in Figure 1, where we observe the superior performance of EQO. Additionally, Table 4 in Appendix G shows that our algorithm also takes less execution time. We provide experiment details in Appendix G.

## 6 CONCLUSION

Our algorithm EQO simultaneously achieves the minimax regret bound and demonstrates practical applicability. Our work introduces the concept of *quasi-optimism*, which relaxes the conventional optimism principle and plays a pivotal role in achieving both theoretical advancements and practical improvements. This fresh perspective offers new insights into obtaining minimax regret bounds, and we anticipate that the underlying idea will be transferable to a wide range of problem settings beyond tabular reinforcement learning, such as model-free estimation or general function approximation.

## Reproducibility Statement

We provide the complete proofs of the theoretical results presented in Section 4 throughout the appendix, and the whole set of employed assumptions is clearly stated in Section 4.1. We also guarantee the reproducibility of the numerical experiments in Section 5 and Appendices G and H.2 by providing the source code with specific seeds as supplementary material.

## Acknowledgements

This work was supported by the National Research Foundation of Korea (NRF) grant funded by the Korea government (MSIT) (No. RS-2022-NR071853 and RS-2023-00222663) and by AI-Bio Research Grant through Seoul National University.

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

Table 2: Table of notations

| Notations for analysis | |
|---|---|
| $K$ | Number of episodes |
| $\tau^k$ | Trajectory of $k$-th episode, $(s_1^k, a_1^k, R_1^k, \ldots, s_H^k, a_H^k, R_H^k, s_{H+1}^k)$ |
| $\tau_h^k$ | Partial trajectory of $k$-th episode up to $h$-th action selection, $(s_1^k, a_1^k, R_1^k, \ldots, s_h^k, a_h^k)$ |
| $\mathcal{F}_h^k$ | $\sigma$-algebra $\sigma(\{\tau^i\}_{i=1}^{k-1} \cup \tau_h^k)$ |
| $N^k(s,a)$ | Visit count of $(s,a) \in \mathcal{S} \times \mathcal{A}$ up to $k-1$-th episode |
| $n_h^k(s,a)$ | Visit count of $(s,a) \in \mathcal{S} \times \mathcal{A}$ up to $h$-th time step of $k$-th episode |
| $\eta^k$ | $\min\{h \in [H] : n_h^k(s_h^k, a_h^k) > 2N^k(s_h^k, a_h^k), H+1\}$ |
| $\Delta_h(V)(s,a)$ | $V_h(s) - PV_{h+1}(s,a)$ |
| $\iota_k$ | 1 when $\{\lambda_k\}_{k=1}^{\infty}$ is constant, $1 + \lfloor \log_2 k \rfloor$ when $\{\lambda_k\}_{k=1}^{\infty}$ changes at powers of two |
| $\ell_1$ | $\log \frac{24HSA}{\delta}$ |
| $\ell_{1,k}$ | $\log \frac{24HSA\iota_k^2}{\delta}$ |
| $\ell'_{1,k}$ | $\log \frac{50HSA(1+\log kH)^2}{\delta}$ |
| $\ell_{2,k}$ | $\log(1 + \frac{kH}{SA})$ |
| $\ell_{3,k}$ | $\log \frac{12SA(1+\log kH)}{\delta}$ |
| $\ell_{3,k}(s,a)$ | $\log \frac{12SA(1+\log N^k(s,a))}{\delta}$ |
| $\beta^k(s,a)$ | $\frac{1}{N^k(s,a)}\left(\frac{11H\ell_{1,k}}{\lambda_k} + 21HS\ell_{3,k}(s,a)\right)$ |
| $\beta_1^k(s,a)$ | $\frac{1}{N^k(s,a)}\left(\frac{3H\ell_{1,k}}{\lambda_k} + 21HS\ell_{3,k}(s,a)\right)$ |
| $U_h^k(s)$ | $\min\{\beta^k(s, \pi_h^k(s)) + PU_{h+1}^k(s, \pi_h^k(s)), H\}$ if $h \in [H]$, 0 if $h = H+1$ |
| Notations exclusive for analysis of PAC bounds | |
| $\ell_{4,\varepsilon}$ | $\log(1 + 270(\frac{H^3\ell_1}{\varepsilon^2} + \frac{H^2 S(2\ell_1 + \ell_{5,\varepsilon})}{\varepsilon}))$ |
| $\ell_{5,\varepsilon}$ | $1 + \log\log(He/\varepsilon)$ |
| $\hat{\beta}^k(s,a)$ | $\frac{1}{N^k(s,a)}\left(\frac{88H^2\ell_1}{\varepsilon} + 30HS\ell_{3,k}(s,a)\right)$ |
| $\bar{\beta}^k(s,a)$ | $\frac{1}{N^k(s,a)}\left(\frac{88H^2\ell_1}{\varepsilon} + 73HS\ell_{3,k}(s,a)\right)$ |
| $\hat{U}_h^k(s)$ | $\min\{\hat{\beta}^k(s, \pi_h^k(s)) + \hat{P}^k\hat{U}_{h+1}^k(s, \pi_h^k(s)), H\}$ if $h \in [H]$, 0 if $h = H+1$ |
| $\bar{U}_h^k(s)$ | $\min\{\bar{\beta}^k(s, \pi_h^k(s)) + P\bar{U}_{h+1}^k(s, \pi_h^k(s)), H\}$ if $h \in [H]$, 0 if $h = H+1$ |
| $\hat{\mathcal{T}}_K$ | Set of $k \in [K]$ that satisfies $\hat{U}_1^k(s_1^k) > \varepsilon/8$ |
| $\hat{T}_K$ | Size of $\hat{\mathcal{T}}_K$ |
| $\bar{\mathcal{T}}_K$ | Set of $k \in [K]$ that satisfies $\bar{U}_1^k(s_1^k) > \varepsilon/16$ |
| $\bar{T}_K$ | Size of $\bar{\mathcal{T}}_K$ |
| $\bar{N}^k(s,a)$ | Visit count of $(s,a) \in \mathcal{S} \times \mathcal{A}$ for episodes in $\bar{\mathcal{T}}_{k-1}$ |
| $\bar{n}_h^k(s,a)$ | Visit count of $(s,a) \in \mathcal{S} \times \mathcal{A}$ for episodes in $\bar{\mathcal{T}}_k$, up to $h$-th time step of $k$-th episode |
| $\bar{\eta}^k$ | $\min\{h \in [H] : \bar{n}_h^k(s_h^k, a_h^k) > 2\bar{N}^k(s_h^k, a_h^k), H+1\}$ |

# Appendix

## A DEFINITIONS AND NOTATIONS

In this section, we define additional concepts and notations for the analysis. We also provide Table 2 for notations defined in this paper. Conventional notations such as $S$, $A$, $H$, $V_h^\pi$, or $\mathbb{N}$ are omitted. Notations that are used exclusively for the analysis of the PAC bounds are introduced in Appendix D.2.

For the well-definedness of some statements in the analysis, we define $1/N^k(s,a)$, $\hat{P}^k(s'|s,a)$, and $\hat{r}^k(s,a)$ to be $+\infty$ when $N^k(s,a) = 0$ throughout this paper.

For any sequence of $H+1$ functions $V = \{V_h\}_{h=1}^{H+1}$ with $V_h : \mathcal{S} \to \mathbb{R}$, we define $\Delta_h(V)(s,a) := V_h(s) - PV_{h+1}(s,a)$. It is similar to the Bellman error, but lacks the reward term. Therefore, for any policy $\pi$, we have $\Delta_h(V_h^\pi)(s, \pi_h(s)) = V_h^\pi(s) - PV_{h+1}^\pi(s, \pi_h(s)) = r(s, \pi_h(s))$ for all $h \in [H]$ and $s \in \mathcal{S}$.

We define $n_h^k(s,a)$ to be the number of times the state-action pair $(s,a) \in \mathcal{S} \times \mathcal{A}$ is visited up to the $h$-th time step of the $k$-th episode, inclusively. To handle some exceptional cases more conveniently, we define $\eta^k$ to be the first time step $h$ such that $n_h^k(s_h^k, a_h^k) > 2N^k(s_h^k, a_h^k)$ occurs in the $k$-th episode for $k \in \mathbb{N}$. In other words, $\eta^k$ is the first time step where the number of times a state-action pair $(s,a) \in \mathcal{S} \times \mathcal{A}$ is visited during the $k$-th episode exceeds $N^k(s,a)$. We define $\eta^k$ to be $H+1$ if there is no such time step. $\eta^k$ is a stopping time with respect to $\{\mathcal{F}_h^k\}_{h=1}^{H+1}$, that is, we have $\{\eta^k = h\} \in \mathcal{F}_h^k$ for all $h \in [H+1]$.

The input $\{c_k\}_{k=1}^\infty$ of Algorithm 1 depends on a sequence of non-increasing positive numbers, $\{\lambda_k\}_{k=1}^\infty$. We mainly consider two cases where $\lambda_k$ is fixed for all $k \in \mathbb{N}$ and $\lambda_k$ changes only at powers of 2, i.e., $\lambda_k \neq \lambda_{k-1}$ only when $k = 2^m$ for some positive integer $m$. We let $\iota_k$ denote the (maximum possible) number of distinct values in $\lambda_1, \ldots, \lambda_k$. Specifically, in the first case where $\lambda_k$ is fixed, we set $\iota_k := 1$ for all $k \in \mathbb{N}$. In the second case where $\lambda_k$ rarely changes, we set $\iota_k := 1 + \lfloor \log_2 k \rfloor$.

Several different logarithmic terms appear in the analysis. For simplicity, we define $\ell_{1,k} := \log \frac{24HSA\iota_k^2}{\delta}$, $\ell_{2,k} := \log(1 + \frac{kH}{SA})$, and $\ell_{3,k} := \log \frac{12SA(1+\log kH)}{\delta}$. We overload the definition of $\ell_{3,k}$ to be a function on $\mathcal{S} \times \mathcal{A}$ with $\ell_{3,k}(s,a) := \log \frac{12SA(1+\log N^k(s,a))}{\delta}$. Additionally, we define $\ell_{1,K}' := \log \frac{50HSA(1+\log KH)^2}{\delta}$, which serves as an upper bound for $\max\{\ell_{1,K}, \ell_{3,K}\}$.

We rigorously define $\beta^k$ and $U_h^k$ introduced in Section 4.4.3.

$$\beta^k(s,a) := \frac{1}{N^k(s,a)} \left( \frac{11H\ell_{1,k}}{\lambda_k} + 21HS\ell_{3,k}(s,a) \right) .$$

$U_h^k(s)$ is the clipped expectation of the sum of $\beta^k$ under $\pi^k$, defined as follows:

$$U_{H+1}^k(s) := 0$$
$$U_h^k(s) := \min \left\{ \beta^k(s, \pi_h^k(s)) + PU_{h+1}^k(s, \pi_h^k(s)), H \right\} \text{ for } h \in [H] .$$

# B HIGH-PROBABILITY EVENTS

In this section, we define the events required for the analysis and show that they hold with high probability. Throughout this section, we assume $\delta \in (0,1]$ and set $\delta' := \delta/6$, and let $\{\lambda_k\}_{k=1}^\infty$ be a fixed sequence of positive real numbers with $\lambda_k \leq 1$ for all $k$.

**Lemma 5.** *With probability at least $1 - \delta'$,*

$$\left| (\hat{P}^k - P)V_{h+1}^*(s,a) \right| \leq \frac{\lambda_k}{4H} \text{Var}(V_{h+1}^*)(s,a) + \frac{3H\ell_{1,k}}{\lambda_k N^k(s,a)}$$

*holds for all $(s,a) \in \mathcal{S} \times \mathcal{A}$, $h \in [H]$, and $k \in \mathbb{N}$.*

*Proof.* Fix $(s,a) \in \mathcal{S} \times \mathcal{A}$, $h \in [H]$, and $\lambda' \in (0,1]$. Suppose $\{s_t\}_{t=1}^\infty$ is a sequence of i.i.d. samples drawn from $P(\cdot|s,a)$. Let $X_t = V_{h+1}^*(s_t) - PV_{h+1}^*(s,a)$. Then, $|X_t| \leq H$ holds almost surely, $\mathbb{E}[X_t] = 0$, and $\mathbb{E}[X_t^2] = \text{Var}(V_{h+1}^*)(s,a)$. Applying Lemma 36 on $\{X_t\}_{t=1}^\infty$ with $\lambda = \lambda'/3$, the following inequality holds for all $n \in \mathbb{N}$ with probability at least $1 - \delta'$:

$$\sum_{t=1}^n X_t \leq \frac{\lambda'n}{4H} \text{Var}(V_{h+1}^*)(s,a) + \frac{3H}{\lambda'} \log \frac{1}{\delta'} .$$

Dividing both sides by $n$ yields

$$\frac{1}{n} \sum_{t=1}^n X_t = (\hat{P}_n - P)V_{h+1}^*(s,a) \leq \frac{\lambda'}{4H} \text{Var}(V_{h+1}^*)(s,a) + \frac{3H}{\lambda'n} \log \frac{1}{\delta'} ,$$

where $\hat{P}_n(s'|s,a) := \sum_{t=1}^n \mathbb{1}\{s_t = s'\}/n$ is the empirical mean of $P(s'|s,a)$ based on $n$ samples. Repeating the same process for $-X_t$, then taking the union bound over the two results, as well as over all $(s,a) \in \mathcal{S} \times \mathcal{A}$ and $h \in [H]$ yields that

$$\left|(\hat{P}_n - P)V_{h+1}^*(s,a)\right| \le \frac{\lambda'}{4H}\operatorname{Var}(V_{h+1}^*)(s,a) + \frac{3H}{\lambda'n}\log\frac{2HSA}{\delta'}$$

holds for all $n \in \mathbb{N}$, $(s,a) \in \mathcal{S} \times \mathcal{A}$, and $h \in [H]$ with probability at least $1 - \delta'$. Now, let $\lambda_1', \lambda_2', \dots$ be the subsequence of $\{\lambda_k\}_{k=1}^\infty$ obtained by removing repetitions. In other words, we have $\lambda'_{\iota_k} = \lambda_k$ for all $k \in \mathbb{N}$. We take the union bound over $\{\lambda_i'\}_i$ by assigning probability $\delta'/(2i^2)$ for $\lambda_i'$. By $\sum_{i=1}^\infty \delta'/(2i^2) \le \delta'$, we have that with probability at least $1 - \delta'$,

$$\left|(\hat{P}_n - P)V_{h+1}^*(s,a)\right| \le \frac{\lambda_i'}{4H}\operatorname{Var}(V_{h+1}^*)(s,a) + \frac{3H}{\lambda_i'n}\log\frac{4HSAi^2}{\delta'} \tag{3}$$

holds for all $n \in \mathbb{N}$, $(s,a) \in \mathcal{S} \times \mathcal{A}$, $h \in [H]$, and $i \in \mathbb{N}$. For any $k \in \mathbb{N}$, by taking $n = N^k(s,a)$ and $i = \iota_k$, inequality (3) implies

$$\left|(\hat{P}^k - P)V_{h+1}^*(s,a)\right| \le \frac{\lambda_k}{4H}\operatorname{Var}(V_{h+1}^*)(s,a) + \frac{3H}{\lambda_k N^k(s,a)}\log\frac{4HSA\iota_k^2}{\delta'}.$$

Replacing $\delta'$ with $\delta/6$, the logarithmic term becomes $\log(24HSA\iota_k^2/\delta) = \ell_{1,k}$, completing the proof. $\square$

**Lemma 6.** *With probability at least $1 - \delta'$,*

$$(P - \hat{P}^k)(V_{h+1}^*)^2(s,a) \le \frac{1}{2}\operatorname{Var}(V_{h+1}^*)(s,a) + \frac{6H^2\ell_{1,k}}{N^k(s,a)}$$

*holds for all $(s,a) \in \mathcal{S} \times \mathcal{A}$, $h \in [H]$, and $k \in \mathbb{N}$.*

*Proof.* Fix $(s,a) \in \mathcal{S} \times \mathcal{A}$ and $h \in [H]$. Let $\{s_t\}_{t=1}^\infty$ be a sequence of i.i.d. samples from $P(\cdot|s,a)$ and define $X_t = P(V_{h+1}^*)^2(s,a) - (V_{h+1}^*)^2(s_t)$, similarly to the proof of Lemma 5. Then, $|X_t| \le H^2$ holds almost surely, $\mathbb{E}[X_t] = 0$, and

$$\mathbb{E}[X_t^2] = \operatorname{Var}((V_{h+1}^*)^2)(s,a) \le 4H^2\operatorname{Var}(V_{h+1}^*)(s,a)$$

holds for all $t \in \mathbb{N}$, where we use Lemma 35 for the inequality. Applying Lemma 36 with $\lambda = 1/6$, the following inequality holds for all $n \in \mathbb{N}$ with probability at least $1 - \delta'$:

$$\sum_{t=1}^n X_t \le \frac{n}{2}\operatorname{Var}(V_{h+1}^*)(s,a) + 6H^2\log\frac{1}{\delta'}.$$

Plugging in $X_t = P(V_{h+1}^*)^2(s,a) - (V_{h+1}^*)^2(s_t)$ and dividing both sides by $n$ yields

$$(P - \hat{P}_n)(V_{h+1}^*)^2(s,a) \le \frac{1}{2}\operatorname{Var}(V_{h+1}^*)(s,a) + \frac{6H^2}{n}\log\frac{1}{\delta'},$$

where $\hat{P}_n(s'|s,a) := \sum_{t=1}^n \mathbb{1}\{s_t = s'\}/n$. Taking the union bound over $(s,a) \in \mathcal{S} \times \mathcal{A}$ and $h \in [H]$, we obtain that

$$(P - \hat{P}_n)(V_{h+1}^*)^2(s,a) \le \frac{1}{2}\operatorname{Var}(V_{h+1}^*)(s,a) + \frac{6H^2}{n}\log\frac{HSA}{\delta'}$$

holds for all $n \in \mathbb{N}$, $(s,a) \in \mathcal{S} \times \mathcal{A}$, and $h \in [H]$ with probability at least $1 - \delta'$. Replacing $\delta'$ with $\delta/6$, the logarithmic term becomes $\log(6HSA/\delta)$, which is less than $\ell_{1,k} = \log(24HSA\iota_k^2/\delta)$ for any $k \in \mathbb{N}$. The proof is completed by taking $n = N^k(s,a)$ for each $k \in \mathbb{N}$. $\square$

**Lemma 7.** *The following inequality holds with probability at least $1 - \delta'$ for any $(s,a) \in \mathcal{S} \times \mathcal{A}$, $s' \in \mathcal{S}$, and $k \in \mathbb{N}$:*

$$\left|\hat{P}^k(s'|s,a) - P(s'|s,a)\right| \le 2\sqrt{\frac{2P(s'|s,a)\ell_{3,k}(s,a)}{N^k(s,a)}} + \frac{2\ell_{3,k}(s,a)}{3N^k(s,a)}$$

*Proof.* Fix $(s, a) \in \mathcal{S} \times \mathcal{A}$ and $s' \in \mathcal{S}$. We write $p := P(s'|s, a)$ for simplicity. Suppose $\{s_t\}_{t=1}^{\infty}$ is a sequence of i.i.d. samples drawn from $P(\cdot|s, a)$. Let $X_t = \mathbb{1}\{s_t = s'\} - p$. Note that $\mathbb{E}[X_t] = 0$ and $\mathbb{E}[X_t^2] = p(1 - p)$. Applying Lemma 37 with $c = p(1 - p)$, the following inequality holds for all $n \in \mathbb{N}$ with probability at least $1 - \delta'$:

$$\sum_{t=1}^{n} X_t \leq 2\sqrt{p(1 - p)n \log \frac{2(1 + \log n)^2}{\delta'}} + \frac{1}{3} \log \frac{2(1 + \log n)^2}{\delta'} \,.$$

We apply the same bound on $\sum_{t=1}^{n} -X_t$, then take the union bound. Further, we bound $p(1 - p) \leq p$ and obtain that

$$\left| \sum_{t=1}^{n} X_t \right| \leq 2\sqrt{pn \log \frac{4(1 + \log n)^2}{\delta'}} + \frac{1}{3} \log \frac{4(1 + \log n)^2}{\delta'} \tag{4}$$

holds for all $n \in \mathbb{N}$ with probability at least $1 - \delta'$. Let $\hat{P}_n(s'|s, a) := \sum_{t=1}^{n} \mathbb{1}\{s' = s_t\}/n$. Dividing both sides of inequality (4) by $n$, we obtain that

$$\left| \hat{P}_n(s' \mid s, a) - P(s' \mid s, a) \right| \leq 2\sqrt{\frac{P(s' \mid s, a)}{n} \log \frac{4(1 + \log n)^2}{\delta'}} + \frac{1}{3n} \log \frac{4(1 + \log n)^2}{\delta'} \,.$$

By taking the union bound over $(s, a, s') \in \mathcal{S} \times \mathcal{A} \times \mathcal{S}$, the logarithmic terms become $\log(4S^2 A(1 + \log n)^2/\delta')$, which is bounded by $\log(4S^2 A^2(1 + \log n)^2/\delta'^2) = 2\log(2SA(1 + \log n)/\delta')$. Therefore, we obtain that

$$\left| \hat{P}_n(s' \mid s, a) - P(s' \mid s, a) \right| \leq 2\sqrt{\frac{2P(s' \mid s, a)}{n} \log \frac{2SA(1 + \log n)}{\delta'}} + \frac{2}{3n} \log \frac{2SA(1 + \log n)}{\delta'} \tag{5}$$

holds for all $n \in \mathbb{N}$, $(s, a, s') \in \mathcal{S} \times \mathcal{A} \times \mathcal{S}$ with probability at least $1 - \delta'$. Finally, by taking $n = N^k(s, a)$ for any $k \in \mathbb{N}$, inequality (5) implies

$$\left| \hat{P}^k(s' \mid s, a) - P(s' \mid s, a) \right|$$

$$\leq 2\sqrt{\frac{2P(s' \mid s, a)}{N^k(s, a)} \log \frac{2SA(1 + \log N^k(s, a))}{\delta'}} + \frac{2}{3N^k(s, a)} \log \frac{2SA(1 + \log N^k(s, a))}{\delta'}$$

$$= 2\sqrt{\frac{2P(s' \mid s, a)\ell_{3,k}(s, a)}{N^k(s, a)}} + \frac{2\ell_{3,k}(s, a)}{3N^k(s, a)} \,,$$

where we use that $\log(2SA(1 + \log N^k(s, a))/\delta') = \log(12SA(1 + \log N^k(s, a))/\delta) = \ell_{3,k}(s, a)$. $\square$

**Lemma 8.** *With probability at least $1 - \delta'$, the following inequality holds for all $(s, a) \in \mathcal{S} \times \mathcal{A}$ and $k \in \mathbb{N}$:*

$$\left| \hat{r}^k(s, a) - r(s, a) \right| \leq \lambda_k r(s, a) + \frac{H\ell_{1,k}}{\lambda_k N^k(s, a)} \,,$$

*Proof.* Fix $(s, a) \in \mathcal{S} \times \mathcal{A}$ and $\lambda' \in (0, 1]$. Let $\{R_t\}_{t=1}^{\infty}$ be a sequence of rewards obtained by choosing $(s, a)$. Let $X_t = R_t - r(s, a)$. By Assumptions 1 and 2, $\{X_t\}_{t=1}^{\infty}$ is a martingale difference sequence with $|X_t| \leq H$ almost surely for all $t$. For simplicity, let $\mathbb{E}_t$ be the conditional expectation conditioned on $\{X_i\}_{i=1}^{t}$. Then, by Lemma 36 with $\lambda = \lambda'$, it holds with probability $1 - \delta'$ that

$$\sum_{t=1}^{n} X_t \leq \frac{\lambda'}{H} \sum_{t=1}^{n} \mathbb{E}_{t-1}\left[X_t^2\right] + \frac{H}{\lambda'} \log \frac{1}{\delta'}$$

for all $n \in \mathbb{N}$, where we bound $3/4$ with $1$ for simplicity. We proceed by using that for a random variable with $0 \leq X \leq H$, it holds that $\mathrm{Var}(X) \leq \mathbb{E}[X^2] \leq H\mathbb{E}[X]$, which implies that $\mathbb{E}_{t-1}[X_t^2] = \mathrm{Var}(R_t) \leq H\mathbb{E}_{t-1}[R_t] = Hr(s, a)$. Therefore, we obtain that

$$\sum_{t=1}^{n} X_t \leq \lambda' n r(s, a) + \frac{H}{\lambda'} \log \frac{1}{\delta'} \,.$$

Dividing both sides by $n$, we derive that with probability at least $1 - \delta'$,

$$\frac{1}{n} \sum_{t=1}^{n} X_t = \hat{r}_n(s,a) - r(s,a) \leq \lambda' r(s,a) + \frac{H}{\lambda' n} \log \frac{1}{\delta'}$$

holds for all $n \in \mathbb{N}$, where $\hat{r}_n := \sum_{t=1}^{n} R_t / n$ is the empirical mean of $n$ random rewards. Repeat the process for $-X_t$ instead of $X_t$, then take the union bound over the two events and over all $(s,a) \in \mathcal{S} \times \mathcal{A}$ and $\lambda_k$, as in the final steps of the proof of Lemma 5. Then, we obtain that

$$\left| \hat{r}^k(s,a) - r(s,a) \right| \leq \lambda_k r(s,a) + \frac{H}{\lambda_k N^k(s,a)} \log \frac{4 S A \iota_k^2}{\delta'}$$

holds for all $(s,a) \in \mathcal{S} \times \mathcal{A}$ and $k \in \mathbb{N}$. The proof is completed by upper bounding the logarithmic term $\log(4 S A \iota_k^2 / \delta')$ by $\ell_{1,k} = \log(24 H S A \iota_k^2 / \delta)$. $\qquad\square$

**Lemma 9.** *Let $\eta^k$ and $U_h^k(\cdot)$ be defined as in Appendix A. With probability at least $1 - \delta'$, the following inequality holds for all $K \in \mathbb{N}$:*

$$\sum_{k=1}^{K} \sum_{h=1}^{\eta^k - 1} \left( P U_{h+1}^k(s_h^k, a_h^k) - U_{h+1}^k(s_{h+1}^k) \right) \leq \frac{1}{4H} \sum_{k=1}^{K} \sum_{h=1}^{\eta^k - 1} \mathrm{Var}(U_{h+1}^k)(s_h^k, a_h^k) + 3H \log \frac{6}{\delta} .$$

*Proof.* Let $X_h^k = \mathbb{1}\{h < \eta^k\}(P U_{h+1}^k(s_h^k, a_h^k) - U_{h+1}^k(s_{h+1}^k))$. We have $\mathbb{1}\{h < \eta^k\} \in \mathcal{F}_h^k$ and $(P U_{h+1}^k(s_h^k, a_h^k) - U_{h+1}^k(s_{h+1}^k)) \in \mathcal{F}_{h+1}^k$, and hence $X_h^k \in \mathcal{F}_{h+1}^k$. Furthermore, we have $\mathbb{E}[X_h^k | \mathcal{F}_h^k] = 0$ since $s_{h+1}^k \sim P(\cdot | s_h^k, a_h^k)$ is independent of $\mathcal{F}_h^k$. Therefore, $\{X_h^k\}_{k,h}$ is a martingale difference sequence with respect to $\{\mathcal{F}_h^k\}_{k,h}$. We have $|X_h^k| \leq H$ almost surely and $\mathbb{E}[(X_h^k)^2 | \mathcal{F}_h^k] = \mathbb{1}\{h < \eta^k\} \mathrm{Var}(U_{h+1}^k)(s_h^k, a_h^k)$. Using Lemma 36 with $\lambda = 1/3$, we obtain that

$$\sum_{k=1}^{K} \sum_{h=1}^{H} X_h^k \leq \frac{1}{4H} \sum_{k=1}^{K} \sum_{h=1}^{H} \mathbb{1}\{h < \eta^k\} \mathrm{Var}(U_{h+1}^k)(s_h^k, a_h^k) + 3H \log \frac{1}{\delta'}$$

holds for all $K \in \mathbb{N}$ with probability at least $1 - \delta'$, which is equivalent to the desired result. $\qquad\square$

**Lemma 10.** *With probability at least $1 - \delta'$, the following inequality holds for all $K \in \mathbb{N}$:*

$$\sum_{k=1}^{K} \sum_{h=1}^{\eta^k - 1} \left( P(U_{h+1}^k)^2(s_h^k, a_h^k) - (U_{h+1}^k)^2(s_{h+1}^k) \right) \leq \frac{1}{2} \sum_{k=1}^{K} \sum_{h=1}^{\eta^k - 1} \mathrm{Var}(U_{h+1}^k)(s_h^k, a_h^k) + 6H^2 \log \frac{6}{\delta}$$

*Proof.* Let $X_h^k = \mathbb{1}\{h < \eta^k\}(P(U_{h+1}^k)^2(s_h^k, a_h^k) - (U_{h+1}^k)^2(s_{h+1}^k))$. For the same reason as in the proof of Lemma 9, $\{X_h^k\}_{k,h}$ is a martingale difference sequence with respect to $\{\mathcal{F}_h^k\}_{k,h}$. We have $|X_h^k| \leq H^2$ almost surely and

$$\mathbb{E}[(X_h^k)^2 | \mathcal{F}_h^k] = \mathbb{1}\{h < \eta^k\} \mathrm{Var}((U_{h+1}^k)^2)(s_h^k, a_h^k) \leq \mathbb{1}\{h < \eta^k\} 4H^2 \mathrm{Var}(U_{h+1}^k)(s_h^k, a_h^k) ,$$

where we use Lemma 35 for the inequality. Using Lemma 36 with $\lambda = 1/6$, we obtain that

$$\sum_{k=1}^{K} \sum_{h=1}^{H} X_h^k \leq \frac{1}{2} \sum_{k=1}^{K} \sum_{h=1}^{H} \mathbb{1}\{h < \eta^k\} \mathrm{Var}(U_{h+1}^k)(s_h^k, a_h^k) + 6H^2 \log \frac{1}{\delta'}$$

holds for all $K \in \mathbb{N}$ with probability at least $1 - \delta'$, which is equivalent to the desired result. $\qquad\square$

Now, we define the event $\mathcal{E}$, under which Theorems 1 and 2 hold.

**Lemma 11.** *Let $\mathcal{E}$ be the intersection of the events of Lemmas 5, 6, 7, 8, 9, and 10. Then, $\mathcal{E}$ occurs with probability at least $1 - \delta$.*

*Proof.* By each of the lemmas and the union bound, $\mathcal{E}$ happens with probability at least $1 - 6\delta' = 1 - \delta$. $\qquad\square$

## C   PROOFS OF THEOREMS 1 AND 2

In this section, we provide the full proofs of Theorems 1 and 2. We begin by restating Proposition 1 with specific logarithmic terms. The proof of the proposition is identical to the one presented in Section 4.4. Lemmas used to prove this proposition are also proved in this section.

**Proposition 2** (Restatement of Proposition 1). *Let $\{\lambda_k\}_{k=1}^{\infty}$ be a sequence of non-increasing positive real numbers with $\lambda_1 \leq 1$. Suppose Algorithm 1 is run with $c_k = 7H\ell_{1,k}/\lambda_k$. Then, under the event of $\mathcal{E}$, the cumulative regret of $K$ episodes is bounded as follows for any $K \in \mathbb{N}$:*

$$Regret(K) \leq 4H\sum_{k=1}^{K}\lambda_k + \frac{88H}{\lambda_K}SA\ell_{1,K}\ell_{2,K} + 168HS^2A\ell_{2,K}\ell_{3,K} + 6HSA\ell_{1,K}\,.$$

*Proof.* The inequality holds by Lemmas 2, 3, and 14, where the last lemma is a restatement of Lemma 4 with specific logarithmic factors. □

Theorems 1 and 2 are proved by assigning appropriate values for $\lambda_k$ in Proposition 2.

*Proof of Theorem 1.* Take $\lambda_k = \min\{1, 5\sqrt{SA\ell_1\ell_{2,K}/K}\}$ for all $k \in [K]$. We apply Proposition 2. First, we bound the sum of $\lambda_k$ for $k \in [K]$ as follows:

$$4H\sum_{k=1}^{K}\lambda_k \leq 4HK \cdot 5\sqrt{\frac{SA\ell_1\ell_{2,K}}{K}}$$
$$= 20H\sqrt{SAK\ell_1\ell_{2,K}}\,.$$

We also have that

$$\frac{88HSA\ell_1\ell_{2,K}}{\lambda_K} = 88HSA\ell_1\ell_{2,K}\max\left\{1, \frac{1}{5}\sqrt{\frac{K}{SA\ell_1\ell_{2,K}}}\right\}$$
$$\leq 88HSA\ell_1\ell_{2,K}\left(1 + \frac{1}{5}\sqrt{\frac{K}{SA\ell_1\ell_{2,K}}}\right)$$
$$= 88HSA\ell_1\ell_{2,K} + 18H\sqrt{SAK\ell_1\ell_{2,K}}\,. \tag{6}$$

By Proposition 2, the cumulative regret of $K$ episodes is bounded as follows:

$$\text{Regret}(K) \leq 38H\sqrt{SAK\ell_1\ell_{2,K}} + 168HS^2A\ell_{2,K}\ell_{3,K} + 88HSA\ell_1\ell_{2,K} + 6HSA\ell_1\,.$$

We further bound the last three terms into a simpler form. Recall that $\ell'_{1,K} = \log\frac{50HSA(1+\log KH)^2}{\delta}$ and that both $\ell_1 \leq \ell'_{1,K}$ and $\ell_{3,K} \leq \ell'_{1,K}$ hold. Therefore, we bound the terms as follows:

$$168HS^2A\ell_{2,K}\ell_{3,K} + 88HSA\ell_1\ell_{2,K} + 6HSA\ell_1$$
$$\leq 168HS^2A\ell'_{1,K}\ell_{2,K} + 88HSA\ell'_{1,K}\ell_{2,K} + 6HSA\ell'_{1,K}$$
$$\leq 168HS^2A\ell'_{1,K}\ell_{2,K} + 88HSA\ell'_{1,K}(1 + \ell_{2,K})$$
$$\leq 256HS^2A\ell'_{1,K}(1 + \ell_{2,K})\,. \tag{7}$$

□

*Proof of Theorem 2.* Fix $k \in \mathbb{N}$ momentarily. Let $m$ be the greatest integer such that $2^m \leq k$. We first show that $\{\lambda_k\}_{k=1}^{\infty}$ is non-increasing, so that Proposition 2 is applicable. We take $\lambda_k = \min\{1, 5\sqrt{SA\ell_{1,2^m}\ell_{2,2^m}/2^m}\}$. Taking $C = \log(24HSA/\delta)$ and defining $f(m)$ as in Lemma 33, we have $\lambda_k = \sqrt{f(m)}$ and the conclusion of the lemma implies that $\{\lambda_k\}_{k=1}^{\infty}$ is non-increasing.

We bound the sum of $\lambda_k$ for $k \in [K]$. Note that $\ell_{1,2^m} \le \ell_{1,k}$, $\ell_{2,2^m} \le \ell_{2,k}$, and $2^m \ge k/2$ hold, hence we have $\lambda_k \le 5\sqrt{2SA\ell_{1,k}\ell_{2,k}/k}$. Therefore, we derive that

$$
\begin{aligned}
4H\sum_{k=1}^{K} \lambda_k &\le 4H\sum_{k=1}^{K} 5\sqrt{\frac{2SA\ell_{1,k}\ell_{2,k}}{k}} \\
&\le 20H\sqrt{2SA\ell_{1,K}\ell_{2,K}}\sum_{k=1}^{K}\sqrt{\frac{1}{k}} \\
&\le 40H\sqrt{2SAK\ell_{1,K}\ell_{2,K}} \\
&\le 57H\sqrt{SAK\ell_{1,K}\ell_{2,K}}\,,
\end{aligned}
$$

where we use that $\sum_{k=1}^{K} k^{-1/2} \le 2\sqrt{K}$ for the penultimate inequality. By the same steps as in inequality (6) of the proof of Theorem 1, we have that

$$
\frac{88HSA\ell_{1,K}\ell_{2,K}}{\lambda_K} \le 88HSA\ell_{1,K}\ell_{2,K} + 18H\sqrt{SAK\ell_{1,K}\ell_{2,K}}\,.
$$

By Proposition 2, the cumulative regret of $K$ episodes is bounded as follows for all $K \in \mathbb{N}$:

$$
\mathrm{Regret}(K) \le 75H\sqrt{SAK\ell_{1,K}\ell_{2,K}} + 168HS^2\ell_{2,K}\ell_{3,K} + 88HSA\ell_{1,K}\ell_{2,K} + 6HSA\ell_{1,K}\,.
$$

Using inequality (7) in the proof of Theorem 1, the sum of the last three terms is upper bounded by $256HS^2A\ell'_{1,K}(1+\ell_{2,K})$, completing the proof. $\qquad\square$

## C.1 PROOF OF LEMMA 2

In this subsection, we prove Lemma 2, which states that our algorithm exhibits quasi-optimism.

*Proof of Lemma 2.* Elementary calculus implies that for $x \in [0, H]$, the bound $0 \le 2x - \frac{1}{2H}x^2 \le \frac{3}{2}H$ holds. Therefore, it is sufficient to prove the following stronger inequality, which we prove by backward induction on $h$:

$$
V_h^*(s) - V_h^k(s) \le \lambda_k\left(2V_h^*(s) - \frac{1}{2H}(V_h^*)^2(s)\right)\,.
$$

The inequality trivially holds for $h = H + 1$ as both sides are 0 in this case. We suppose the inequality holds for $h+1$ and show that it holds for $h$. Since the right-hand side is greater than or equal to 0, the inequality trivially holds when $V_h^k(s) = H$. Suppose $V_h^k(s) < H$. Denoting $a := \pi_h^k(s)$ and $a^* := \pi_h^*(s)$, we have that

$$
V_h^k(s) = Q_h^k(s,a) \ge Q_h^k(s,a^*) = \hat{r}^k(s,a^*) + b^k(s,a^*) + \hat{P}^k V_{h+1}^k(s,a^*)\,,
$$

where the first inequality holds by the choice of $a = \mathrm{argmax}_{a' \in \mathcal{A}} Q_h^k(s,a')$ of the algorithm, and the last equality holds since $Q_h^k(s,a^*) \le V_h^k(s) < H$. We bound $V_h^*(s) - V_h^k(s)$ as follows:

$$
\begin{aligned}
V_h^*(s) - V_h^k(s) &\le \left(r(s,a^*) + PV_{h+1}^*(s,a^*)\right) - \left(\hat{r}^k(s,a^*) + b^k(s,a^*) + \hat{P}^k V_{h+1}^k(s,a^*)\right) \\
&= -b^k(s,a^*) + \underbrace{r(s,a^*) - \hat{r}^k(s,a^*)}_{I_1} + \underbrace{PV_{h+1}^*(s,a^*) - \hat{P}^k V_{h+1}^k(s,a^*)}_{I_2}\,. \quad (8)
\end{aligned}
$$

$I_1$ is bounded by Lemma 8 as follows:

$$
I_1 \le \lambda_k r(s,a^*) + \frac{H\ell_{1,k}}{\lambda_k N^k(s,a^*)}\,. \tag{9}
$$

We bound $I_2$ as follows:

$$
\begin{aligned}
I_2 &= (P - \hat{P}^k)V_{h+1}^*(s, a^*) + \hat{P}^k(V_{h+1}^* - V_{h+1}^k)(s, a^*) \\
&\leq (P - \hat{P}^k)V_{h+1}^*(s, a^*) + \lambda_k \hat{P}^k \left(2V_{h+1}^* - \frac{1}{2H}(V_{h+1}^*)^2\right)(s, a^*) \\
&= (P - \hat{P}^k)V_{h+1}^*(s, a^*) + \lambda_k(\hat{P}^k - P)\left(2V_{h+1}^* - \frac{1}{2H}(V_{h+1}^*)^2\right)(s, a^*) \\
&\quad + \lambda_k P \left(2V_{h+1}^* - \frac{1}{2H}(V_{h+1}^*)^2\right)(s, a^*) \\
&= (1 - 2\lambda_k)(P - \hat{P}^k)V_{h+1}^*(s, a^*) + \frac{\lambda_k}{2H}(P - \hat{P}^k)(V_{h+1}^*)^2(s, a^*) \\
&\quad + \lambda_k P \left(2V_{h+1}^* - \frac{1}{2H}(V_{h+1}^*)^2\right)(s, a^*),
\end{aligned}
\tag{10}
$$

where the first equality adds and subtracts $\hat{P}^k V_{h+1}^*(s, a^*)$, the next inequality is due to the induction hypothesis, and the following equality adds and subtracts $P(2V_{h+1}^* - (V_{h+1}^*)^2/(2H))$. We bound the first two terms using the concentration events. Since $0 \leq \lambda_k \leq 1$, we have $|1 - 2\lambda_k| \leq 1$. Using Lemma 5, we have $|(P - \hat{P}^k)V_{h+1}^*(s, a^*)| \leq \frac{\lambda_k}{4H}\operatorname{Var}(V_{h+1}^*)(s, a^*) + \frac{3H\ell_{1,k}}{\lambda_k N^k(s,a^*)}$. By Lemma 6, we have $(P - \hat{P}^k)(V_{h+1}^*)^2(s, a^*) \leq \operatorname{Var}(V_{h+1}^*)(s, a^*)/2 + \frac{6H^2\ell_{1,k}}{N^k(s,a^*)}$. Plugging in these bounds into inequality (10), we obtain that

$$
\begin{aligned}
I_2 &\leq \frac{\lambda_k}{4H}\operatorname{Var}(V_{h+1}^*)(s, a^*) + \frac{3H\ell_{1,k}}{\lambda_k N^k(s,a^*)} + \frac{\lambda_k}{4H}\operatorname{Var}(V_{h+1}^*)(s, a^*) + \frac{3H\lambda_k\ell_{1,k}}{N^k(s,a^*)} \\
&\quad + \lambda_k P \left(2V_{h+1}^* - \frac{1}{2H}(V_{h+1}^*)^2\right)(s, a^*) \\
&\leq \frac{\lambda_k}{2H}\operatorname{Var}(V_{h+1}^*)(s, a^*) + \frac{6H\ell_{1,k}}{\lambda_k N^k(s,a^*)} + \lambda_k P \left(2V_{h+1}^* - \frac{1}{2H}(V_{h+1}^*)^2\right)(s, a^*) \\
&= \frac{\lambda_k}{2H}\left(\operatorname{Var}(V_{h+1}^*)(s, a^*) - P(V_{h+1}^*)^2(s, a^*)\right) + 2\lambda_k P V_{h+1}^*(s, a^*) + \frac{6H\ell_{1,k}}{\lambda_k N^k(s,a^*)},
\end{aligned}
$$

where the second inequality applies $\lambda_k \leq 1/\lambda_k$ from $\lambda_k \leq 1$. By Lemma 27, we have $\operatorname{Var}(V_{h+1}^*)(s, a^*) - P(V_{h+1}^*)^2(s, a^*) \leq -(V_h^*)^2(s) + 2H \max\{\Delta_h(V^*)(s, a^*), 0\}$, where in this case we have $\Delta_h(V^*)(s, a^*) = r(s, a^*)$. Therefore, we obtain that

$$
I_2 \leq -\frac{\lambda_k}{2H}(V_h^*)^2(s) + \lambda_k r(s, a^*) + 2\lambda_k P V_{h+1}^*(s, a^*) + \frac{6H\ell_{1,k}}{\lambda_k N^k(s,a^*)}.
\tag{11}
$$

Combining inequalities (8), (9), and (11) together, we complete the induction step as follows:

$$
\begin{aligned}
V_h^*(s) - V_h^k(s) &\leq -b^k(s, a^*) + \lambda_k r(s, a^*) + \frac{H\ell_{1,k}}{\lambda_k N^k(s,a^*)} \\
&\quad - \frac{\lambda_k}{2H}(V_h^*)^2(s) + \lambda_k r(s, a^*) + 2\lambda_k P V_{h+1}^*(s, a^*) + \frac{6H\ell_{1,k}}{\lambda_k N^k(s,a^*)} \\
&= -b^k(s, a^*) + \frac{7H\ell_{1,k}}{\lambda_k N^k(s,a^*)} + 2\lambda_k(r(s, a^*) + P V_{h+1}^*(s, a^*)) - \frac{\lambda_k}{2H}(V_h^*)^2(s) \\
&= \lambda_k \left(2V_h^*(s) - \frac{1}{2H}(V_h^*)^2(s)\right),
\end{aligned}
$$

where the last inequality uses that $b^k(s, a^*) = 7H\ell_{1,k}/(\lambda_k N^k(s, a^*))$ and $r(s, a^*) + P V_{h+1}^*(s, a^*) = V_h^*(s)$. $\qquad\square$

## C.2 PROOF OF LEMMA 3

To prove this lemma, we need the following two technical lemmas.

**Lemma 12.** *For any $s \in \mathcal{S}$, $h \in [H+1]$, and $k \in \mathbb{N}$, define $\widetilde{V}_h^k(s) := V_h^k(s) - V_h^*(s)$. Under the event $\mathcal{E}$, the following inequality holds for all $(s,a) \in \mathcal{S} \times \mathcal{A}$, $h \in [H]$, and $k \in \mathbb{N}$:*

$$\left| (\hat{P}^k - P) V_{h+1}^k(s,a) \right| \le \frac{\lambda_k}{4H} \operatorname{Var}(V_{h+1}^*)(s,a) + \frac{1}{10H} \operatorname{Var}(\widetilde{V}_{h+1}^k)(s,a) + \beta_1^k(s,a) \,,$$

*where $\beta_1^k(s,a) := \frac{1}{N^k(s,a)} (3H\ell_{1,k}/\lambda_k + 21HS\ell_{3,k}(s,a))$.*

*Proof.* We add and subtract $(\hat{P}^k - P)V_{h+1}^*(s,a)$ and then use the triangle inequality to obtain

$$\left| (\hat{P}^k - P) V_{h+1}^k(s,a) \right| \le \underbrace{\left| (\hat{P}^k - P)\left(V_{h+1}^k - V_{h+1}^*\right)(s,a) \right|}_{I_1} + \underbrace{\left| (\hat{P}^k - P) V_{h+1}^*(s,a) \right|}_{I_2} \,.$$

By Lemma 5, $I_2$ is bounded by $\frac{\lambda_k}{4H} \operatorname{Var}(V_{h+1}^*)(s,a) + 3H\ell_{1,k}/(\lambda_k N^k(s,a))$. To bound $I_1$, we apply Lemma 29 with $\rho = 10$ and obtain

$$I_1 \le \frac{1}{10H} \operatorname{Var}(\widetilde{V}_{h+1}^k)(s,a) + \frac{21HS\ell_{3,k}(s,a)}{N^k(s,a)} \,.$$

Putting these bounds together, we conclude that

$$\left| (\hat{P}^k - P) V_{h+1}^k(s,a) \right|$$
$$\le \frac{\lambda_k}{4H} \operatorname{Var}(V_{h+1}^*)(s,a) + \frac{1}{10H} \operatorname{Var}(\widetilde{V}_{h+1}^k)(s,a) + \frac{1}{N^k(s,a)} \left( \frac{3H\ell_{1,k}}{\lambda_k} + 21HS\ell_{3,k}(s,a) \right)$$
$$= \frac{\lambda_k}{4H} \operatorname{Var}(V_{h+1}^*)(s,a) + \frac{1}{10H} \operatorname{Var}(\widetilde{V}_{h+1}^k)(s,a) + \beta_1^k(s,a) \,.$$

$\square$

**Lemma 13.** *Under the event $\mathcal{E}$, the following inequality holds for all $s \in \mathcal{S}$, $h \in [H]$, and $k \in \mathbb{N}$:*

$$\Delta_h(V^k - V^{\pi^k})(s,a)$$
$$\le \Delta_h \left( \lambda_k \left( 3V^* - \frac{1}{2H}(V^*)^2 \right) - \frac{1}{5H} \left( \widetilde{V}^k + \frac{3}{2}\lambda_k H \right)^2 \right)(s,a) + 2\beta^k(s,a) \,, \qquad (12)$$

*where $a := \pi_h^k(s)$ and $\beta^k(s,a) := \frac{1}{N^k(s,a)} (11H\ell_{1,k}/\lambda_k + 21HS\ell_{3,k}(s,a))$.*

*Proof.* We begin as follows:

$$\Delta_h(V^k - V^{\pi^k})(s,a) = \left(V_h^k(s) - PV_{h+1}^k(s,a)\right) - \left(V_h^{\pi^k}(s) - PV_{h+1}^{\pi^k}(s,a)\right)$$
$$\le \left(\hat{r}^k(s,a) + b^k(s,a) + (\hat{P}^k - P)V_{h+1}^k(s,a)\right) - r(s,a)$$
$$= b^k(s,a) + (\hat{r}^k(s,a) - r(s,a)) + (\hat{P}^k - P)V_{h+1}^k(s,a) \,.$$

By Lemma 8, we have that $\hat{r}^k(s,a) - r(s,a) \le \lambda_k r(s,a) + \frac{H\ell_{1,k}}{\lambda_k N^k(s,a)}$. By Lemma 12, it holds that $(\hat{P}^k - P)V_{h+1}^k(s,a) \le \frac{\lambda_k}{4H} \operatorname{Var}(V_{h+1}^*)(s,a) + \frac{1}{10H} \operatorname{Var}(\widetilde{V}_{h+1}^k)(s,a) + \beta_1^k(s,a)$. Define $I_1 := \max\{\Delta_h(V^k - V^{\pi^k})(s,a), 0\}$. Combining the bounds and using that $\beta^k(s,a) = b^k(s,a) + \beta_1^k(s,a) + H\ell_{1,k}/(\lambda_k N^k(s,a))$ holds by definition, we obtain

$$I_1 \le \lambda_k r(s,a) + \frac{\lambda_k}{4H} \operatorname{Var}(V_{h+1}^*)(s,a) + \frac{1}{10H} \operatorname{Var}(\widetilde{V}_{h+1}^k)(s,a) + \beta^k(s,a) \,. \qquad (13)$$

Applying Lemma 27 to $\operatorname{Var}(V_{h+1}^*)(s,a)$, we have that $\operatorname{Var}(V_{h+1}^*)(s,a) \le -\Delta_h((V^*)^2)(s,a) + 2H \max\{\Delta_h(V^*)(s,a), 0\}$. Since Lemma 28 states that $\Delta_h(V^*)(s,a) \ge 0$, we infer that

$$\operatorname{Var}(V_{h+1}^*)(s,a) \le -\Delta_h((V^*)^2)(s,a) + 2H\Delta_h(V^*)(s,a) \,. \qquad (14)$$

By Lemma 2, we have $\widetilde{V}_h^k + \frac{3}{2}\lambda_k H \geq 0$ for all $h \in [H+1]$. Applying Lemma 27 to $\mathrm{Var}(\widetilde{V}_{h+1}^k)(s,a) = \mathrm{Var}(\widetilde{V}_{h+1}^k + \frac{3}{2}\lambda_k H)(s,a)$, we obtain that

$$\mathrm{Var}\left(\widetilde{V}_{h+1}^k + \frac{3}{2}\lambda_k H\right)(s,a) \leq -\Delta_h\left(\left(\widetilde{V}^k + \frac{3}{2}\lambda_k H\right)^2\right)(s,a)$$

$$+ (2H + 3\lambda_k H)\max\left\{\Delta_h\left(\widetilde{V}^k + \frac{3}{2}\lambda_k H\right)(s,a), 0\right\}.$$

We bound $\Delta_h(\widetilde{V}^k + \frac{3}{2}\lambda_k H)(s,a)$ as follows:

$$\Delta_h\left(\widetilde{V}^k + \frac{3}{2}\lambda_k H\right)(s,a) = \Delta_h(\widetilde{V}^k)(s,a)$$

$$= \Delta_h(V^k)(s,a) - \Delta_h(V^*)(s,a)$$

$$\leq \Delta_h(V^k)(s,a) - r(s,a)$$

$$= \Delta_h(V^k)(s,a) - \Delta_h(V^{\pi^k})(s,a)$$

$$= \Delta_h(V^k - V^{\pi^k})(s,a),$$

where the inequality is due to Lemma 28. Therefore, by the definition of $I_1$, we obtain that $\max\{\Delta_h(\widetilde{V}^k + \frac{3}{2}\lambda_k H)(s,a), 0\} \leq I_1$ and conclude that

$$\mathrm{Var}\left(\widetilde{V}_{h+1}^k + \frac{3}{2}\lambda_k H\right)(s,a) \leq -\Delta_h\left(\left(\widetilde{V}^k + \frac{3}{2}\lambda_k H\right)^2\right)(s,a) + (2H + 3\lambda_k H)I_1$$

$$\leq -\Delta_h\left(\left(\widetilde{V}^k + \frac{3}{2}\lambda_k H\right)^2\right)(s,a) + 5HI_1, \qquad (15)$$

where we use $\lambda_k \leq 1$ for the last inequality. Plugging inequalities (14) and (15) into inequality (13), then applying $r(s,a) \leq \Delta_h(V^*)(s,a^*)$ by Lemma 28, we obtain that

$$I_1 \leq \Delta_h\left(\lambda_k\left(\frac{3}{2}V^* - \frac{1}{4H}(V^*)^2\right) - \frac{1}{10H}\left(\widetilde{V}^k + \frac{3}{2}\lambda_k\right)^2\right)(s,a) + \beta^k(s,a) + \frac{1}{2}I_1.$$

Solving the inequality with respect to $I_1$ implies inequality (12). $\qquad\square$

Now, we are ready to prove Lemma 3.

*Proof of Lemma 3.* For notational simplicity, we define the following quantity:

$$D_h(s) := \lambda_k\left(3V_h^*(s) - \frac{1}{2H}(V_h^*)^2(s)\right) + \frac{1}{5H}\left(\left(\frac{3}{2}\lambda_k H\right)^2 - \left(\widetilde{V}_h^k(s) + \frac{3}{2}\lambda_k H\right)^2\right).$$

For $x \in [0, H]$, the bound $0 \leq 3x - x^2/(2H) \leq \frac{5}{2}H$ holds. Similarly, for $c \in [0, \frac{3}{2}H]$ and $y \in [-H, H]$, we have $c^2 - (y+c)^2 = -y^2 - 2cy$ and $-4H^2 \leq -y^2 - 2cy \leq 0$. Therefore, by setting $x = V_h^*(s)$, $y = \widetilde{V}_h^k(s)$, and $c = \frac{3}{2}\lambda_k H$, we obtain that $-\frac{4}{5}H \leq D_h(s) \leq \frac{5}{2}\lambda_k H$ for all $h \in [H]$ and $s \in \mathcal{S}$.

To prove the lemma, we prove the following stronger inequality by backward induction on $h$:

$$V_h^k(s) - V_h^{\pi^k}(s) \leq D_h(s) + 2U_h^k(s).$$

Since $D_{H+1}(s) = 0$ for all $s \in \mathcal{S}$, the inequality trivially holds for $h = H+1$ as $0 \leq 0$. Suppose that the inequality holds for $h + 1$. By Lemma 13, which can be rewritten as $\Delta_h(V^k - V^{\pi^k}) \leq \Delta_h(D)(s,a) + 2\beta^k(s,a)$, we have that

$$V_h^k(s) - V_h^{\pi^k}(s) = \Delta_h(V^k - V^{\pi^k})(s,a) + P(V_{h+1}^k - V_{h+1}^{\pi^k})(s,a)$$

$$\leq \Delta_h(D)(s,a) + 2\beta^k(s,a) + P(V_{h+1}^k - V_{h+1}^{\pi^k})(s,a). \qquad (16)$$

By the induction hypothesis, we have that

$$P(V_{h+1}^k - V_{h+1}^{\pi^k})(s,a) \le P(D_{h+1} + 2U_{h+1}^k)(s,a). \tag{17}$$

Combining inequalities (16) and (17) yields

$$V_h^k(s) - V_h^{\pi^k}(s) \le D_h(s) + 2(\beta^k(s,a) + PU_{h+1}^k(s,a)). \tag{18}$$

Finally, by that $-4H/5 \le D_h(s)$ and $V_h^k(s) - V_h^{\pi^k}(s) \le H$ always hold, the following inequality always holds:

$$V_h^k(s) - V_h^{\pi^k}(s) \le H \le D_h(s) + 2H. \tag{19}$$

By inequalities (18) and (19), we conclude that

$$V_h^k(s) - V_h^{\pi^k}(s) \le D_h(s) + 2\min\{\beta^k(s,a) + PU_{h+1}^k(s,a), H\}$$
$$= D_h(s) + 2U_h^k(s),$$

completing the induction argument. $\qquad\square$

### C.3   PROOF OF LEMMA 4

We restate Lemma 4 with specific logarithmic factors.

**Lemma 14** (Restatement of Lemma 4). *Under $\mathcal{E}$, it holds that*

$$\sum_{k=1}^K U_1^k(s_1^k) \le \frac{44HSA\ell_{1,K}}{\lambda_K} + 84HS^2A\ell_{2,K}\ell_{3,K} + 3HSA\ell_{1,K}$$

*for all $K \in \mathbb{N}$.*

We prove this lemma in two steps: using the concentration results to bound $\sum_k U^k$ by $\sum_{k,h} \beta^k(s_h^k, a_h^k)$ and using the logarithmic bound for the harmonic series, $\sum_{n=1}^N 1/n \le 1 + \log N$.

**Lemma 15.** *Let $\eta^k$ be defined as in Appendix A. Under the event $\mathcal{E}$, it holds that*

$$\sum_{k=1}^K U_1^k(s_1^k) \le 2\sum_{k=1}^K \sum_{h=1}^{\eta^k-1} \beta^k(s_h^k, a_h^k) + 3HSA\ell_{1,K}.$$

*for all $K \in \mathbb{N}$.*

*Proof.* Decompose $U_1^k(s_1^k)$ as follows:

$$U_1^k(s_1^k) \le \beta^k(s_1^k, a_1^k) + PU_2^k(s_1^k, a_1^k)$$
$$= \beta^k(s_1^k, a_1^k) + PU_2^k(s_1^k, a_1^k) - U_2^k(s_2^k) + U_2^k(s_2^k)$$
$$\vdots$$
$$\le \sum_{h=1}^{\eta^k-1} \left(\beta^k(s_h^k, a_h^k) + PU_{h+1}^k(s_h^k, a_h^k) - U_{h+1}^k(s_{h+1}^k)\right) + U_{\eta^k}^k(s_{\eta^k}^k)$$
$$\le \sum_{h=1}^{\eta^k-1} \left(\beta^k(s_h^k, a_h^k) + PU_{h+1}^k(s_h^k, a_h^k) - U_{h+1}^k(s_{h+1}^k)\right) + H\mathbb{1}\{\eta^k \ne H+1\},$$

where the last inequality uses that $U_{H+1}^k(s) = 0$ and $U_h^k(s) \le H$ for all $s \in \mathcal{S}$ and $h \in [H]$. We take the sum of $U_1^k(s_1^k)$ for $k = 1, 2, \ldots, K$. Let $I_1 := \sum_{k=1}^K \sum_{h=1}^{\eta^k-1} (PU_{h+1}^k(s_h^k, a_h^k) - U_{h+1}^k(s_{h+1}^k))$, so that

$$\sum_{k=1}^K U_1^k(s_1^k) \le \sum_{k=1}^K \sum_{h=1}^{\eta^k-1} \beta^k(s_h^k, a_h^k) + H\sum_{k=1}^K \mathbb{1}\{\eta^k \ne H+1\} + I_1. \tag{20}$$

By Lemma 9, we obtain that

$$I_1 \le \frac{1}{4H} \sum_{k=1}^{K} \sum_{h=1}^{\eta^k-1} \mathrm{Var}(U_{h+1}^k)(s_h^k, a_h^k) + 3H \log \frac{6}{\delta}$$

$$=: \frac{1}{4H} I_2 + 3H \log \frac{6}{\delta} \,, \tag{21}$$

where we define $I_2 := \sum_{k=1}^{K} \sum_{h=1}^{\eta^k-1} \mathrm{Var}(U_{h+1}^k)(s_h^k, a_h^k)$. By Lemma 27, we have that

$$\mathrm{Var}(U_{h+1}^k)(s_h^k, a_h^k) \le -\Delta_h((U^k)^2)(s_h^k, a_h^k) + 2H \max\{\Delta_h(U^k)(s_h^k, a_h^k), 0\}$$
$$\le -\Delta_h((U^k)^2)(s_h^k, a_h^k) + 2H\beta^k(s_h^k, a_h^k)$$
$$= -(U_h^k)^2(s_h^k) + P(U_{h+1}^k)^2(s_h^k, a_h^k) + 2H\beta^k(s_h^k, a_h^k) \,,$$

where the second inequality uses that

$$\Delta_h(U^k)(s, a) = U_h^k(s) - PU_{h+1}^k(s, a) \le (\beta^k(s, a) + PU_{h+1}^k(s, a)) - PU_{h+1}^k(s, a) = \beta^k(s, a) \,.$$

Therefore, the sum of the variances of $U_{h+1}^k(s_h^k, a_h^k)$ for the $k$-th episode is bounded as follows:

$$\sum_{h=1}^{\eta^k-1} \mathrm{Var}(U_{h+1}^k)(s_h^k, a_h^k) \le -\sum_{h=1}^{\eta^k-1} (U_h^k)^2(s_h^k) + \sum_{h=1}^{\eta^k-1} P(U_{h+1}^k)^2(s_h^k, a_h^k) + \sum_{h=1}^{\eta^k-1} 2H\beta^k(s_h^k, a_h^k)$$

$$= \sum_{h=1}^{\eta^k-1} 2H\beta^k(s_h^k, a_h^k) - (U_1^k)^2(s_1^k) + (U_{\eta^k}^k)^2(s_{\eta^k}^k)$$

$$+ \sum_{h=1}^{\eta^k-1} \left( P(U_{h+1}^k)^2(s_h^k, a_h^k) - (U_{h+1}^k)^2(s_{h+1}^k) \right)$$

$$\le \sum_{h=1}^{\eta^k-1} 2H\beta^k(s_h^k, a_h^k) + H^2 \mathbb{1}\{\eta^k \ne H+1\}$$

$$+ \sum_{h=1}^{\eta^k-1} (P(U_{h+1}^k)^2(s_h^k, a_h^k) - (U_{h+1}^k)^2(s_{h+1}^k)) \,,$$

where we again use that $U_{\eta^k}^k(s_{\eta^k}^k) \le H\mathbb{1}\{\eta^k \ne H+1\}$ for the last inequality. Therefore, by taking the sum over $k \in [K]$, $I_2$ is bounded as follows:

$$I_2 \le \sum_{k=1}^{K} \sum_{h=1}^{\eta^k-1} 2H\beta^k(s_h^k, a_h^k) + H^2 \sum_{k=1}^{K} \mathbb{1}\{\eta^k \ne H+1\}$$

$$+ \sum_{k=1}^{K} \sum_{h=1}^{\eta^k-1} (P(U_{h+1}^k)^2(s_h^k, a_h^k) - (U_{h+1}^k)^2(s_{h+1}^k)) \,.$$

The last double sum is bounded by Lemma 10 as follows:

$$\sum_{k=1}^{K} \sum_{h=1}^{\eta^k-1} (P(U_{h+1}^k)^2(s_h^k, a_h^k) - (U_{h+1}^k)^2(s_{h+1}^k))$$

$$\le \frac{1}{2} \sum_{k=1}^{K} \sum_{h=1}^{\eta^k-1} \mathrm{Var}(U_{h+1}^k)(s_h^k, a_h^k) + 6H^2 \log \frac{6}{\delta}$$

$$= \frac{1}{2} I_2 + 6H^2 \log \frac{6}{\delta} \,.$$

Therefore, we deduce that

$$I_2 \le \sum_{k=1}^{K} \sum_{h=1}^{\eta^k-1} 2H\beta^k(s_h^k, a_h^k) + H^2 \sum_{k=1}^{K} \mathbb{1}\{\eta^k \ne H+1\} + \frac{1}{2} I_2 + 6H^2 \log \frac{6}{\delta} \,.$$

Solving the inequality with respect to $I_2$, we obtain that

$$I_2 \leq \sum_{k=1}^{K} \sum_{h=1}^{\eta^k-1} 4H\beta^k(s_h^k, a_h^k) + 2H^2 \sum_{k=1}^{K} \mathbb{1}\{\eta^k \neq H+1\} + 12H^2 \log \frac{6}{\delta}. \tag{22}$$

Plugging the bound of inequality (22) into inequality (21), we obtain that

$$I_1 \leq \sum_{k=1}^{K} \sum_{h=1}^{\eta^k-1} \beta^k(s_h^k, a_h^k) + \frac{H}{2} \sum_{k=1}^{K} \mathbb{1}\{\eta^k \neq H+1\} + 6H \log \frac{6}{\delta}. \tag{23}$$

By combining inequalities (20) and (23), we conclude that

$$\sum_{k=1}^{K} U_1^k(s_1^k) \leq 2 \sum_{k=1}^{K} \sum_{h=1}^{\eta^k-1} \beta^k(s_h^k, a_h^k) + \frac{3H}{2} \sum_{k=1}^{K} \mathbb{1}\{\eta^k \neq H+1\} + 6H \log \frac{6}{\delta}.$$

Finally, we bound the last two terms using Lemma 30 as follows:

$$\frac{3H}{2} \sum_{k=1}^{K} \mathbb{1}\{\eta^k \neq H+1\} + 6H \log \frac{6}{\delta} \leq \frac{3H}{2} SA \log_2 2H + 6H \log \frac{6}{\delta}$$

$$\leq 3HSA \log 2H + 3HSA \log \frac{6}{\delta}$$

$$= 3HSA \log \frac{12H}{\delta}$$

$$\leq 3HSA\ell_{1,K},$$

where the first inequality is due to Lemma 30 and the second inequality applies $\log_2 2H \leq 2\log 2H$ on the first term and $A \geq 2$ on the second term. The proof is complete. $\square$

*Proof of Lemma 14.* By Lemma 15, we have

$$\sum_{k=1}^{K} U_1^k(s_1^k) \leq 2 \sum_{k=1}^{K} \sum_{h=1}^{\eta^k-1} \beta^k(s_h^k, a_h^k) + 3HSA\ell_{1,K}.$$

Let $\gamma_k = 11H\ell_{1,k}/\lambda_k + 21HS\ell_{3,k}$. Then, it holds that $\beta(s, a) \leq \gamma_k/N^k(s, a)$. We apply Lemma 31 and obtain that

$$\sum_{k=1}^{K} \sum_{h=1}^{\eta^k-1} \beta^k(s_h^k, a_h^k) \leq \sum_{k=1}^{K} \sum_{h=1}^{\eta^k-1} \frac{\gamma_k}{N^k(s, a)}$$

$$\leq 2\gamma_K SA \log \left(1 + \frac{KH}{SA}\right)$$

$$= \frac{22HSA\ell_{1,K}\ell_{2,K}}{\lambda_K} + 42HS^2 A\ell_{2,K}\ell_{3,K}.$$

Combining the two inequalities completes the proof. $\square$

# D  PAC BOUNDS

In this section, we provide the analysis of PAC bounds. We summarize previous achievements and our results on PAC bounds for episodic finite-horizon MDPs in Table 3. We note that although Jin et al. (2018) propose a conversion that enables a regret-minimizing algorithm to solve BPI tasks, the conversion is sub-optimal in terms of $1/\delta$-dependence; it results in $1/\delta^2$-dependence when $\log \frac{1}{\delta}$ is possible. Refer to Appendix E in Ménard et al. (2021a) for a detailed discussion.

## D.1  ALGORITHM

We introduce $(\varepsilon, \delta)$-EQO, an algorithm for the PAC tasks, and describe it in Algorithm 2. The interaction between the agent and the environment is the same as EQO, where the parameters are set based on $\varepsilon$ and $\delta$. Then, it executes additional procedures to verify whether the policy $\pi^k$ is $\varepsilon$-optimal, which is necessary for BPI tasks.

Table 3: Comparison of PAC bounds of different algorithms for tabular reinforcement learning. '-' denotes that the bound is not available.

| Paper | Best-Policy Identification | Mistake-style PAC |
|---|---|---|
| Dann & Brunskill (2015) | - | $\frac{H^2 S^2 A}{\varepsilon^2} \log \frac{1}{\delta}$ |
| Dann et al. (2017) | - | $(\frac{H^4 SA}{\varepsilon^2} + \frac{H^4 S^3 A^2}{\varepsilon}) \log \frac{1}{\delta}$ |
| Dann et al. (2019) | $(\frac{H^2 SA}{\varepsilon^2} + \frac{H^3 S^2 A}{\varepsilon}) \log \frac{1}{\delta}$ | $(\frac{H^2 SA}{\varepsilon^2} + \frac{H^3 S^2 A}{\varepsilon}) \log \frac{1}{\delta}$ |
| Ménard et al. (2021a) | $\frac{H^2 SA}{\varepsilon^2} \log \frac{1}{\delta} + \frac{H^2 SA}{\varepsilon}(S + \log \frac{1}{\delta})$ | - |
| Zhang et al. (2021a) | $(\frac{H^2 SA}{\delta^2 \varepsilon^2} + \frac{H S^2 A}{\delta \varepsilon}) \log \frac{1}{\delta}$ | - |
| **This work** | $(\frac{H^2 SA}{\varepsilon^2} + \frac{H S^2 A}{\varepsilon}) \log \frac{1}{\delta}$ | $(\frac{H^2 SA}{\varepsilon^2} + \frac{H S^2 A}{\varepsilon}) \log \frac{1}{\delta}$ |

---

**Algorithm 2:** $(\varepsilon, \delta)$-EQO

**Input** : $\varepsilon \in (0, H], \delta \in (0, 1]$
**Output:** $\Pi$, Set of $\varepsilon$-optimal policies

1   $\hat{\beta}(n) := \frac{1}{n} \left( \frac{88H^2}{\varepsilon} \log \frac{24HSA}{\delta} + 30HS \log \frac{12SA \log(en)}{\delta} \right)$;

2   $\Pi \leftarrow \emptyset$;

3   **for** $k = 1, 2, \ldots$ **do**

4     Compute $\pi^k$ using Algorithm 1 with $c_k = \frac{56H^2}{\varepsilon} \log \frac{24HSA}{\delta}$;

5     $\hat{U}_{H+1}^k(s) \leftarrow 0$ for all $s \in \mathcal{S}$;

6     **for** $h = H, H-1, \ldots, 1$ **do**

7       **foreach** $s \in \mathcal{S}$ **do**

8         $a \leftarrow \pi_h^k(s)$;

9         $\hat{\beta}^k(s,a) \leftarrow \hat{\beta}(N^k(s,a))$;

10         $\hat{U}_h^k(s) \leftarrow \begin{cases} \min\left\{ \hat{\beta}^k(s,a) + \hat{P}^k \hat{U}_{h+1}^k(s,a), H \right\} & \text{if } N^k(s,a) > 0 \\ H & \text{if } N^k(s,a) = 0 \end{cases}$;

11     **if** $\hat{U}_1^k(s_1) \leq \frac{\varepsilon}{8}$ **then**

12       Add $\pi^k$ to $\Pi$;

13       // If current task is BPI, return $\pi^k$;

14     Execute policy $\pi^k$ and observe trajectory $(s_1^k, a_1^k, s_2^k, \ldots, s_H^k, a_H^k, s_{H+1}^k)$;

---

### D.2   ADDITIONAL DEFINITIONS FOR PAC BOUNDS

In this section, we define additional concepts that are required to analyze the PAC bounds.

We define two more logarithmic terms, $\ell_{4,\varepsilon} = \log(1 + 270(\frac{H^3 \ell_1}{\varepsilon^2} + \frac{H^2 S(2\ell_1 + \ell_{5,\varepsilon})}{\varepsilon}))$ and $\ell_{5,\varepsilon} = 1 + \log \log(He/\varepsilon)$. We also define analogous concepts for $\beta^k, U_h^k, N^k, n_h^k$, and $\eta^k$. We define $\hat{\beta}$ and $\bar{\beta}$, which are functions that map $\mathbb{N}$ to $\mathbb{R}$ as follows:

$$\hat{\beta}(n) := \frac{1}{n} \left( \frac{88H^2 \ell_1}{\varepsilon} + 30HS\ell_{3,n} \right)$$

$$\bar{\beta}(n) := \frac{1}{n} \left( \frac{88H^2 \ell_1}{\varepsilon} + 73HS\ell_{3,n} \right).$$

For $k \in \mathbb{N}$, $\hat{\beta}^k$ and $\overline{\beta}^k$ are functions from $\mathcal{S} \times \mathcal{A}$ to $\mathbb{R}$ defined using $\hat{\beta}$ and $\overline{\beta}$:

$$\hat{\beta}^k(s, a) := \hat{\beta}(N^k(s, a))$$
$$= \frac{1}{N^k(s, a)} \left( \frac{88 H^2 \ell_1}{\varepsilon} + 30 H S \ell_{3,k}(s, a) \right)$$
$$\overline{\beta}^k(s, a) := \overline{\beta}(N^k(s, a))$$
$$= \frac{1}{N^k(s, a)} \left( \frac{88 H^2 \ell_1}{\varepsilon} + 73 H S \ell_{3,k}(s, a) \right).$$

$\hat{U}_h^k(s)$ and $\overline{U}_h^k$ are defined in a similar manner to $U_h^k$, but using $\hat{\beta}^k$ and $\overline{\beta}^k$ instead of $\beta^k$, respectively. Additionally, the definition of $\hat{U}_h^k(s)$ uses $\hat{P}^k$ instead of $P$. They are formally defined by the following iterative relationships:

$$\hat{U}_{H+1}^k(s) := \overline{U}_{H+1}^k(s) := 0$$
$$\hat{U}_h^k(s) := \min\{\hat{\beta}^k(s, \pi_h^k(s)) + \hat{P}^k \hat{U}_{h+1}^k(s, \pi_h^k(s)), H\} \text{ for } h \in [H]$$
$$\overline{U}_h^k(s) := \min\{\overline{\beta}^k(s, \pi_h^k(s)) + P\overline{U}_{h+1}^k(s, \pi_h^k(s)), H\} \text{ for } h \in [H].$$

Algorithm 2 adds $\pi^k$ to $\Pi$ if $\hat{U}_1^k(s_1^k) \le \varepsilon/8$. We denote the set of episodes that do not meet this condition among the first $K$ by $\hat{\mathcal{T}}_K$, and its size by $\hat{T}_K$. In the analysis, we are also interested in the episodes with $\overline{U}_h^k(s_1^k) > \varepsilon/16$. For $K \in \mathbb{N}$, we define $\overline{\mathcal{T}}_K := \{k \in [K] : \overline{U}_h^k(s_1^k) > \varepsilon/16\}$ to be the set of episodes that satisfy $\overline{U}_h^k(s_1^k) > \varepsilon/16$ among the first $K$ episodes. Analogously, $\overline{T}_K$ is the size of $\overline{\mathcal{T}}_K$.

We define $\overline{n}_h^k$ and $\overline{N}^k(s, a)$, which are the counterparts of $n_h^k$ and $N^k(s, a)$, but only count the episodes in $\overline{\mathcal{T}}_K$. Specifically, we define them as follows:

$$\overline{n}_h^k(s, a) := \sum_{i \in \overline{\mathcal{T}}_k} \sum_{j=1}^{H} \mathbb{1}\{(s_j^i, a_j^i) = (s, a), (i < k \text{ or } j \le h)\}$$
$$\overline{N}^k(s, a) := \overline{n}_H^{k-1}(s, a)$$
$$= \sum_{i \in \overline{\mathcal{T}}_{k-1}} \sum_{h=1}^{H} \mathbb{1}\{(s_h^i, a_h^i) = (s, a)\}.$$

Finally, we define $\overline{\eta}^k$, which is the counterpart of $\eta^k$ defined by using $\overline{n}_h^k$ and $\overline{N}^k$ instead. Specifically, $\overline{\eta}^k := \min\{h \in [H] : \overline{n}_h^k(s_h^k, a_h^k) > 2\overline{N}^k(s_h^k, a_h^k)\}$, where $\overline{\eta}^k = H + 1$ if there is no such $h \in [H]$.

### D.3 High-probability Events for PAC Bounds

To prove Theorems 3 and 4, the events of Lemmas 9 and 10 have to be replaced by the following events. To summarize, $U_h^k$ is replaced with $\overline{U}_h^k$, $\eta^k$ is replaced with $\overline{\eta}^k$, and only the episodes in $\overline{\mathcal{T}}_K$ contribute to the sum instead of all $k \in [K]$. Recall that $\delta' = \delta/6$.

**Lemma 16.** *Fix $\varepsilon \in (0, H]$. With probability at least $1 - \delta'$, the following inequality holds for all $K \in \mathbb{N}$:*

$$\sum_{k \in \overline{\mathcal{T}}_K} \sum_{h=1}^{\overline{\eta}^k - 1} \left( P\overline{U}_{h+1}^k(s_h^k, a_h^k) - \overline{U}_{h+1}^k(s_{h+1}^k) \right) \le \frac{1}{4H} \sum_{k \in \overline{\mathcal{T}}_K} \sum_{h=1}^{\overline{\eta}^k - 1} \mathrm{Var}(\overline{U}_{h+1}^k)(s_h^k, a_h^k) + 3H \log \frac{6}{\delta}.$$

*Proof.* Let $I_h^k := \mathbb{1}\{\overline{U}_1^k(s_1^k) > \varepsilon/16, h < \overline{\eta}^k\}$ and $X_h^k = I_h^k(P\overline{U}_{h+1}^k(s_h^k, a_h^k) - \overline{U}_{h+1}^k(s_{h+1}^k))$. Since $I_h^k \in \mathcal{F}_h^k$, $\{X_h^k\}_{k,h}$ is a martingale difference sequence with respect to $\{\mathcal{F}_h^k\}_{k,h}$ as in the proof of Lemma 9. We have $|X_h^k| \le H$ almost surely and $\mathbb{E}[(X_h^k)^2 | \mathcal{F}_h^k] = I_h^k \mathrm{Var}(\overline{U}_{h+1}^k)(s_h^k, a_h^k)$.

Using Lemma 36 with $\lambda = 1/3$, we obtain that

$$\sum_{k=1}^{K}\sum_{h=1}^{H} X_h^k \leq \frac{1}{4H}\sum_{k=1}^{K}\sum_{h=1}^{H} I_h^k \operatorname{Var}(\overline{U}_{h+1}^k)(s_h^k, a_h^k) + 3H\log\frac{1}{\delta'}$$

holds for all $K \in \mathbb{N}$ with probability at least $1 - \delta'$, which is equivalent to the desired result. $\quad\square$

**Lemma 17.** *Fix $\varepsilon \in (0, H]$. Then, with probability at least $1 - \delta$, the following inequality holds for all $K \in \mathbb{N}$:*

$$\sum_{k\in\overline{\mathcal{T}}_K}\sum_{h=1}^{\overline{\eta}^k-1}\left(P(\overline{U}_{h+1}^k)^2(s_h^k, a_h^k) - (\overline{U}_{h+1}^k)^2(s_{h+1}^k)\right) \leq \frac{1}{2}\sum_{k\in\overline{\mathcal{T}}_K}\sum_{h=1}^{\overline{\eta}^k-1}\operatorname{Var}(\overline{U}_{h+1}^k)(s_h^k, a_h^k) + 6H^2\log\frac{6}{\delta}$$

*Proof.* Let $I_h^k = \mathbb{1}\{\overline{U}_1^k(s_1^k) > \varepsilon/16, h < \overline{\eta}^k\}$ and $X_h^k = I_h^k(P(\overline{U}_{h+1}^k)^2(s_h^k, a_h^k) - (\overline{U}_{h+1}^k)^2(s_{h+1}^k))$. As in the proof of Lemma 16, $\{X_h^k\}_{k,h}$ is a martingale difference sequence with respect to $\{\mathcal{F}_h^k\}_{k,h}$. We have $|X_h^k| \leq H^2$ almost surely and

$$\mathbb{E}[(X_h^k)^2|\mathcal{F}_h^k] = I_h^k \operatorname{Var}((\overline{U}_{h+1}^k)^2)(s_h^k, a_h^k) \leq 4H^2 I_h^k \operatorname{Var}(\overline{U}_{h+1}^k)(s_h^k, a_h^k),$$

where we use Lemma 35 for the last inequality. Applying Lemma 36 with $\lambda = 1/6$, we obtain that

$$\sum_{k=1}^{K}\sum_{h=1}^{H} X_h^k \leq \frac{1}{2}\sum_{k=1}^{K}\sum_{h=1}^{H} I_h^k \operatorname{Var}(\overline{U}_{h+1}^k)(s_h^k, a_h^k) + 6H^2\log\frac{1}{\delta'}$$

holds for all $K \in \mathbb{N}$ with probability at least $1 - \delta'$, which is equivalent to the desired result. $\quad\square$

Now, we define the event under which the bound of Theorems 3 and 4 holds.

**Lemma 18.** *Let $\overline{\mathcal{E}}$ be the intersection of the events of Lemmas 5, 6, 7, 8, 16, and 17. Then, $\overline{\mathcal{E}}$ happens with probability at least $1 - \delta$.*

*Proof.* This lemma is proved by taking the union bound over the listed lemmas. $\quad\square$

## D.4 PROOFS OF THEOREMS 3 AND 4

In this section, we prove Theorems 3 and 4. The following proposition presents the theoretical guarantees enjoyed by Algorithm 2, and it directly implies both theorems.

**Proposition 3.** *Fix $\varepsilon \in (0, H]$ and $\delta \in (0, 1]$. Let $\Pi$ be the output of Algorithm 2. Under $\overline{\mathcal{E}}$, the following two propositions hold:*

*1. All policies in $\Pi$ are $\varepsilon$-optimal.*

*2. The number of episodes whose policies are not included in $\Pi$ is at most $K_0$,*

*where $K_0$ is defined as follows:*

$$K_0 := \left\lfloor \frac{12000H^2SA\ell_1\ell_{4,\varepsilon}}{\varepsilon^2} + \frac{5000HS^2A(2\ell_1 + \ell_{5,\varepsilon})\ell_{4,\varepsilon}}{\varepsilon} \right\rfloor.$$

Assuming that Proposition 3 is true, Theorems 3 and 4 are proved as follows:

*Proof of Theorem 3.* Proposition 3 states that under $\overline{\mathcal{E}}$, all policies of $\Pi$ are $\varepsilon$-optimal, hence all the policies that are not $\varepsilon$-optimal are not in $\Pi$. Proposition 3 also states that the number of episodes whose policies are not included in $\Pi$ is at most $K_0$, therefore the number of episodes whose policies are not $\varepsilon$-optimal is at most $K_0$. By Lemma 18, the probability of $\overline{\mathcal{E}}$ is at least $1 - \delta$, completing the proof. $\quad\square$

*Proof of Theorem 4.* Since the number of episodes whose policies are not included in $\Pi$ is at most $K_0$ under $\bar{\mathcal{E}}$ by Proposition 3, there exists at least one episode among the first $K_0 + 1$ whose policy is added to $\Pi$. As all policies in $\Pi$ are $\varepsilon$-optimal, the algorithm may return the first such policy. The probability of this event is guaranteed by Lemma 18. $\qquad\square$

Now, we prove Proposition 3.

The following two lemmas show the relationships between $U_h^k$, $\hat{U}_h^k$, and $\overline{U}_h^k$.

**Lemma 19.** *Under $\bar{\mathcal{E}}$, it holds that for all $s \in \mathcal{S}$, $h \in [H]$, and $k \in \mathbb{N}$,*

$$U_h^k(s) \leq 2\hat{U}_h^k(s)\,.$$

**Lemma 20.** *Under $\bar{\mathcal{E}}$, it holds that for all $s \in \mathcal{S}$, $h \in [H]$, and $k \in \mathbb{N}$,*

$$\hat{U}_h^k(s) \leq 2\overline{U}_h^k(s)\,.$$

The proofs of these lemmas are deferred to Appendices D.5 and D.6 respectively.

We first show that under $\bar{\mathcal{E}}$, the policies in $\Pi$ are $\varepsilon$-optimal. Note that by setting $\lambda_k = \frac{\varepsilon}{8H}$, Algorithm 2 runs Algorithm 1 with $c_k = 7\ell_1/\lambda_k$. Also, the proofs of Lemmas 2 and 3 do not rely on Lemmas 9 and 10. Therefore, the conclusions of Lemmas 2 and 3 hold with $\lambda_k = \frac{\varepsilon}{8H}$ under $\bar{\mathcal{E}}$. This fact leads to the following lemma:

**Lemma 21.** *Suppose that Algorithm 2 is run and the event $\bar{\mathcal{E}}$ holds. If $\hat{U}_1^k(s_1^k) \leq \varepsilon/8$, then policy $\pi^k$ is $\varepsilon$-optimal. Consequently, all the policies in $\Pi$ are $\varepsilon$-optimal.*

*Proof.* By Lemmas 2 and 3, the instantaneous regret in episode $k$ is at most $4\lambda_k + 2U_1^k(s_1^k) = \varepsilon/2 + 2U_1^k(s_1^k)$. By Lemma 19, this quantity is less than or equal to $\varepsilon/2 + 4\hat{U}_1^k(s_1^k)$. Therefore, if $\hat{U}_1^k(s_1^k) \leq \varepsilon/8$, then the instantaneous regret in episode $k$ is at most $\varepsilon/2 + \varepsilon/2 = \varepsilon$. $\qquad\square$

Now, we prove the second part of the proposition that states that the number of episodes whose policies are not added to $\Pi$ is finite. Restating our goal using the notations defined in Appendix D.2, we want to show that $\hat{T}_K \leq K_0$ for all $K \in \mathbb{N}$. To do so, we show $\hat{T}_K \leq \overline{T}_K$ and $\overline{T}_K \leq K_0$. To show $\overline{T}_K \leq K_0$, we provide upper and lower bounds of $\sum_{k \in \overline{T}_K} \overline{U}_1^k(s_1^k)$. While the lower bound is straightforward to obtain, the upper bound is more technical. We state the upper-bound result in Lemma 22 and defer its proof to Appendix D.7. We note that Lemma 22 and its proof are analogous to those of Lemma 14.

**Lemma 22.** *Under $\bar{\mathcal{E}}$, it holds that*

$$\sum_{k \in \overline{\mathcal{T}}_K} \overline{U}_1^k(s_1^k) \leq \frac{352H^2 SA\ell_1\ell_{2,\overline{T}_K}}{\varepsilon} + 292HS^2 A\ell_{2,\overline{T}_K}\ell_{3,\overline{T}_K} + 3HSA\ell_1$$

*for all $K \in \mathbb{N}$.*

We require one more technical lemma, which is necessary to derive an upper bound of $\overline{T}_K$ from an inequality it satisfies.

**Lemma 23.** *One has*

$$\frac{5632H^2 SA\ell_1\ell_{2,K_0}}{\varepsilon^2} + \frac{4672HS^2 A\ell_{2,K_0}\ell_{3,K_0} + 48HSA\ell_1}{\varepsilon} < K_0\,.$$

The proof of this lemma is deferred to Appendix D.8

Now, we are ready to prove Proposition 3.

*Proof of Proposition 3.* By Lemma 21, we have that for all policies in $\Pi$ are $\varepsilon$-optimal, which proves the first part of the proposition.
Now, we prove the second part of the proposition, that there are at most $K_0$ episodes whose policies are not included in $\Pi$. By Lemma 20, $\hat{U}_1^k(s_1^k) > \varepsilon/8$ implies that $\overline{U}_1^k(s_1^k) > \varepsilon/16$. Hence, the

number of episodes where $\hat{U}_1^k(s_1^k) > \varepsilon/8$ holds during the first $K$ episodes is at most $\overline{T}_K$. Therefore, it is sufficient to show that $\overline{T}_K \leq K_0$ holds for all $K \in \mathbb{N}$.

Using Lemma 22, we obtain the following condition on $\overline{T}_K$:

$$
\frac{\varepsilon \overline{T}_K}{16} \leq \sum_{k \in \overline{\mathcal{T}}_K} \overline{U}_1^k(s_1^k)
$$

$$
\leq \frac{352 H^2 SA \ell_1 \ell_{2,\overline{T}_K}}{\varepsilon} + 292 HS^2 A \ell_{2,\overline{T}_K} \ell_{3,\overline{T}_K} + 3 HSA \ell_1 \,,
$$

where the first inequality holds since $\overline{U}_1^k(s_1^k)$ is greater than $\varepsilon/16$ when $k \in \overline{\mathcal{T}}_K$ by definition, and the second inequality is from Lemma 22. Rearranging the terms, we deduce that $\overline{T}_K$ satisfies the following inequality for any $K \in \mathbb{N}$:

$$
\overline{T}_K \leq \frac{5632 H^2 SA \ell_1 \ell_{2,\overline{T}_K}}{\varepsilon^2} + \frac{4672 HS^2 A \ell_{2,\overline{T}_K} \ell_{3,\overline{T}_K} + 48 HSA \ell_1}{\varepsilon} \,.
$$

This inequality, combined with Lemma 23, shows that one can not have $\overline{T}_K = K_0$ for any $K \in \mathbb{N}$. Since $\overline{T}_K$ starts at $\overline{T}_0 = 0$ and increases by at most 1 as $K$ increases, we conclude that $\overline{T}_K < K_0$ must hold for all $K \in \mathbb{N}$. $\qquad\square$

### D.5  PROOF OF LEMMA 19

*Proof of Lemma 19.*  We prove that the following stronger inequality holds by backward induction on $h$:

$$
U_h^k(s) \leq 2 \hat{U}_h^k(s) - \frac{1}{2H} (U_h^k)^2(s) \,.
$$

The inequality is trivial when $h = H + 1$. Suppose the inequality holds for $h + 1$. The inequality is trivial when $\hat{U}_h^k(s) = H$. Assume that $\hat{U}_h^k(s) < H$, so that $\hat{U}_h^k(s) = \hat{\beta}^k(s,a) + \hat{P}^k \hat{U}_{h+1}^k(s,a)$, where $a := \pi_h^k(s)$. We have that

$$
U_h^k(s) \leq \beta^k(s,a) + P U_{h+1}^k(s,a)
$$
$$
= \beta^k(s,a) + (P - \hat{P}^k) U_{h+1}^k(s,a) + \hat{P}^k U_{h+1}^k(s,a) \,. \tag{24}
$$

We bound the second term in inequality (24) by applying Lemma 29 with $\rho = 4$.

$$
(P - \hat{P}^k) U_{h+1}^k(s,a) \leq \frac{1}{4H} \mathrm{Var}(U_{h+1}^k)(s,a) + \frac{9 HS \ell_{3,k}(s,a)}{N^k(s,a)} \,. \tag{25}
$$

We bound the last term of inequality (24) using the induction hypothesis as follows:

$$
\hat{P}^k U_{h+1}^k(s,a) \leq \hat{P}^k \left( 2 \hat{U}_{h+1}^k - \frac{1}{2H} (U_{h+1}^k)^2 \right)(s,a)
$$
$$
= 2 \hat{P}^k \hat{U}_{h+1}^k(s,a) + \frac{1}{2H} (P - \hat{P}^k)(U_{h+1}^k)^2(s,a) - \frac{1}{2H} P(U_{h+1}^k)^2(s,a) \,. \tag{26}
$$

For $(P - \hat{P}^k)(U_{h+1}^k)^2(s,a)$, we apply Lemma 29 with $\rho = 8$ and obtain the following bound:

$$
(P - \hat{P}^k)(U_{h+1}^k)^2(s,a) \leq \frac{1}{8H^2} \mathrm{Var}((U_{h+1}^k)^2)(s,a) + \frac{17 H^2 S \ell_{3,k}(s,a)}{N^k(s,a)}
$$
$$
\leq \frac{1}{2} \mathrm{Var}(U_{h+1}^k)(s,a) + \frac{17 H^2 S \ell_{3,k}(s,a)}{N^k(s,a)} \,, \tag{27}
$$

where we use Lemma 35 for the last inequality. Plugging inequality (27) into inequality (26), we obtain that

$$
\hat{P}^k U_{h+1}^k(s,a) \leq 2 \hat{P}^k \hat{U}_{h+1}^k(s,a) + \frac{1}{4H} \mathrm{Var}(U_{h+1}^k)(s,a) + \frac{9 HS \ell_{3,k}(s,a)}{N^k(s,a)} - \frac{1}{2H} P(U_{h+1}^k)^2(s,a) \,. \tag{28}
$$

Plugging inequalities (25) and (28) into inequality (24), we obtain that

$$U_h^k(s) \le \beta^k(s,a) + \frac{18HS\ell_{3,k}(s,a)}{N^k(s,a)} + \frac{1}{2H}\left(\mathrm{Var}(U_{h+1}^k)(s,a) - P(U_{h+1}^k)^2(s,a)\right) + 2\hat{P}^k\hat{U}_{h+1}^k(s,a).$$

By Lemma 27, we have that

$$\mathrm{Var}(U_{h+1}^k)(s,a) - P(U_{h+1}^k)^2(s,a) \le -(U_h^k)^2(s) + 2H\max\{\Delta_h(U^k)(s,a), 0\}$$
$$\le -(U_h^k)^2(s) + 2H\beta^k(s,a),$$

where the last inequality uses that

$$\Delta_h(U^k)(s,a) = U_h^k(s) - PU_{h+1}^k(s,a) \le (\beta^k(s,a) + PU_{h+1}^k(s,a)) - PU_{h+1}^k(s,a) = \beta^k(s,a).$$

Therefore, we conclude that

$$U_h^k(s) \le 2\beta^k(s,a) + \frac{18HS\ell_{3,k}(s,a)}{N^k(s,a)} - \frac{1}{2H}(U_h^k)^2(s) + 2\hat{P}^k\hat{U}_{h+1}^k(s,a)$$

$$= 2\hat{\beta}^k(s,a) + 2\hat{P}^k\hat{U}_{h+1}^k(s,a) - \frac{1}{2H}(U_h^k)^2(s)$$

$$= 2\hat{U}_h^k(s) - \frac{1}{2H}(U_h^k)^2(s),$$

where the first equality comes from that $\hat{\beta}^k(s,a) = \beta^k(s,a) + \frac{9HS\ell_{3,k}(s,a)}{N^k(s,a)}$ by their definitions and the second by $\hat{U}_h^k(s) = \hat{\beta}^k(s,a) + \hat{P}^k\hat{U}_{h+1}^k(s,a)$. □

## D.6 PROOF OF LEMMA 20

*Proof of Lemma 20.* We prove the following stronger inequality by backward induction on $h$:

$$\hat{U}_h^k(s) \le 2\overline{U}_h^k(s) - \frac{1}{2}(\overline{U}_h^k)^2(s).$$

The inequality trivially holds when $h = H + 1$ or $\overline{U}_h^k(s) = H$. Suppose the inequality holds for $h + 1$ and $\overline{U}_h^k(s) < H$. Using the induction hypothesis, we derive that

$$\hat{U}_h^k(s) \le \hat{\beta}^k(s,a) + \hat{P}^k\hat{U}_{h+1}^k(s,a)$$

$$\le \hat{\beta}^k(s,a) + \hat{P}^k\left(2\overline{U}_{h+1}^k - \frac{1}{2H}(\overline{U}_{h+1}^k)^2\right)(s,a)$$

$$= \hat{\beta}^k(s,a) + (\hat{P}^k - P)\left(2\overline{U}_{h+1}^k - \frac{1}{2H}(\overline{U}_{h+1}^k)^2\right)(s,a) + P\left(2\overline{U}_{h+1}^k - \frac{1}{2H}(\overline{U}_{h+1}^k)^2\right)(s,a)$$

$$= \hat{\beta}^k(s,a) + 2(\hat{P}^k - P)\overline{U}_{h+1}^k(s,a) + \frac{1}{2H}(P - \hat{P}^k)(\overline{U}_{h+1}^k)^2(s,a)$$

$$\quad + P\left(2\overline{U}_{h+1}^k - \frac{1}{2H}(\overline{U}_{h+1}^k)^2\right)(s,a), \tag{29}$$

where $a := \pi_h^k(s)$. Using Lemma 29 with $\rho = 8$, we obtain that

$$(\hat{P}^k - P)\overline{U}_{h+1}^k(s,a) \le \frac{1}{8H}\mathrm{Var}(\overline{U}_{h+1}^k)(s,a) + \frac{17HS\ell_{3,k}(s,a)}{N^k(s,a)}$$

and

$$(P - \hat{P}^k)(\overline{U}_{h+1}^k)^2(s,a) \le \frac{1}{8H^2}\mathrm{Var}((\overline{U}_{h+1}^k)^2)(s,a) + \frac{17H^2S\ell_{3,k}(s,a)}{N^k(s,a)}$$

$$\le \frac{1}{2}\mathrm{Var}(\overline{U}_{h+1}^k)(s,a) + \frac{17H^2S\ell_{3,k}(s,a)}{N^k(s,a)},$$

where we use Lemma 35 for the last inequality. Plugging these bounds into inequality (29), we obtain that

$$\hat{U}_h^k(s) \le \hat{\beta}^k(s,a) + \frac{1}{2H}\mathrm{Var}(\overline{U}_{h+1}^k)(s,a) + \frac{43HS\ell_{3,k}(s,a)}{N^k(s,a)} + P\left(2\overline{U}_{h+1}^k - \frac{1}{2H}(\overline{U}_{h+1}^k)^2\right)(s,a)$$

$$= \overline{\beta}^k(s,a) + \frac{1}{2H}\left(\mathrm{Var}(\overline{U}_{h+1}^k)(s,a) - P(\overline{U}_{h+1}^k)^2(s,a)\right) + 2P\overline{U}_{h+1}^k(s,a),$$

where the last equality comes from that $\bar{\beta}^k(s,a) = \hat{\beta}^k(s,a) + \frac{43HS\ell_{3,k}(s,a)}{N^k(s,a)}$ by their definitions. Using Lemma 27, we have

$$\text{Var}(\bar{U}_{h+1}^k)(s,a) - P(\bar{U}_{h+1}^k)^2(s,a) \leq -(\bar{U}_h^k)^2(s) + 2H\max\{\Delta_h(\bar{U}^k)(s,a),0\}$$
$$\leq -(\bar{U}_h^k)^2(s) + 2H\bar{\beta}^k(s,a)\,,$$

where the last inequality uses that

$$\Delta_h(\bar{U}^k)(s,a) = \bar{U}_h^k(s) - P\bar{U}_{h+1}^k(s,a) \leq (\bar{\beta}^k(s,a) + P\bar{U}_{h+1}^k(s,a)) - P\bar{U}_{h+1}^k(s,a) = \bar{\beta}^k(s,a)\,.$$

Therefore, we conclude that

$$\hat{U}_h^k(s) \leq 2\bar{\beta}^k(s,a) + 2P\bar{U}_{h+1}^k(s,a) - \frac{1}{2H}(\bar{U}_h^k)^2(s)$$
$$= 2\bar{U}_h^k(s) - \frac{1}{2H}(\bar{U}_h^k)^2(s)\,,$$

completing the induction. $\qquad\square$

### D.7 PROOF OF LEMMA 22

Analogously to Lemma 14, Lemma 22 is proved in two steps: first, using the concentration results to bound $\sum_k \bar{U}^k$ with $\sum_{k,h} \bar{\beta}^k(s_h^k, a_h^k)$, and second, using that $\sum_{n=1}^N 1/n \leq 1 + \log N$ to bound $\sum_{k,h} \bar{\beta}^k(s_h^k, a_h^k)$. However, more meticulous care is required for the second step, as the bound must depend only on $\bar{T}_K$ and be independent of $K$.

**Lemma 24.** *Under $\bar{\mathcal{E}}$, it holds that*

$$\sum_{k\in\bar{\mathcal{T}}_K} \bar{U}_1^k(s_1^k) \leq 2 \sum_{k\in\bar{\mathcal{T}}_K} \sum_{h=1}^{\bar{\eta}^k-1} \bar{\beta}^k(s_h^k, a_h^k) + 3HSA\ell_{1,K}$$

*for all $K \in \mathbb{N}$.*

*Proof.* The proof is identical to the proof of Lemma 14, except that the use of Lemmas 9 and 10 is replaced with Lemmas 16 and 17. $\qquad\square$

**Lemma 25.** *Under $\bar{\mathcal{E}}$, it holds that*

$$\sum_{k\in\bar{\mathcal{T}}_K} \sum_{h=1}^{\bar{\eta}^k-1} \bar{\beta}^k(s_h^k, a_h^k) \leq 2SA\left(\frac{88H^2\ell_1}{\varepsilon} + 73HS\ell_{3,\bar{T}_K}\right)\ell_{2,\bar{T}_K}$$

*for all $K \in \mathbb{N}$.*

*Proof.* Recall that $\bar{N}^k(s,a)$ represents the number of times the state-action pair $(s,a) \in \mathcal{S} \times \mathcal{A}$ is visited in episodes that satisfy $\bar{U}_1^i(s_1^i) > \varepsilon/16$ up to the $(k-1)$-th episode. Clearly, $N^k(s,a) \geq \bar{N}^k(s,a)$. By Lemma 34 with $C_1 = 88H^2\ell_1/\varepsilon + 73HS\log(12SA/\delta)$ and $C_2 = 73HS$, we have that $\bar{\beta}(n) := (C_1 + C_2\log(1 + \log n))/n$ is non-increasing. Therefore, we know that $\bar{\beta}^k(s,a) = \bar{\beta}(N^k(s,a)) \leq \bar{\beta}(\bar{N}^k(s,a))$. Thus, we have that

$$\sum_{k\in\bar{\mathcal{T}}_K} \sum_{h=1}^{\bar{\eta}^k-1} \bar{\beta}^k(s_h^k, a_h^k) \leq \sum_{k\in\bar{\mathcal{T}}_K} \sum_{h=1}^{\bar{\eta}^k-1} \bar{\beta}(\bar{N}^k(s_h^k, a_h^k))\,.$$

Since $\bar{N}^{K+1}(s,a) \leq \bar{T}_K H$, we have $\bar{\beta}(\bar{N}^k(s,a)) \leq \bar{\gamma}/\bar{N}^k(s,a)$, where $\bar{\gamma} = 88H^2\ell_1/\varepsilon + 73HS\ell_{3,\bar{T}_K}$. By Lemma 31, we conclude that

$$\sum_{k\in\bar{\mathcal{T}}_K} \sum_{h=1}^{\bar{\eta}^k-1} \bar{\beta}(\bar{N}^k(s_h^k, a_h^k)) \leq \sum_{k\in\bar{\mathcal{T}}_K} \sum_{h=1}^{\bar{\eta}^k-1} \frac{\bar{\gamma}}{\bar{N}^k(s_h^k, a_h^k)}$$
$$\leq 2\bar{\gamma}SA\log\left(1 + \frac{\bar{T}_K H}{SA}\right)$$
$$= 2SA\left(\frac{88H^2\ell_1}{\varepsilon} + 73HS\ell_{3,\bar{T}_K}\right)\ell_{2,\bar{T}_K}\,.$$

To be more specific, we apply Lemma 31 to the episodes in $\overline{\mathcal{T}}_K$, meaning that $\tau^k$ in Lemma 31 should be the trajectory of the $k$-th episode that satisfies $\overline{U}_1^i(s_1^i) > \varepsilon/16$, and the sum is taken over $\overline{T}_K$ episodes. $\qquad\square$

*Proof of Lemma 22.* Combine the results of Lemmas 24 and 25. $\qquad\square$

### D.8 PROOF OF LEMMA 23

Before proving Lemma 23, a technical lemma regarding the logarithmic terms is required.

**Lemma 26.** *The following inequalities are true:*

$$\ell_{2,K_0} \leq 2\ell_{4,\varepsilon} \tag{30}$$

$$\ell_{3,K_0} \leq 2\ell_1 + \ell_{5,\varepsilon} \,. \tag{31}$$

*Proof.* We first provide a crude bound for $\log K_0 H$. Let $B := \frac{H^2\ell_1}{\varepsilon^2} + \frac{HS(2\ell_1+\ell_{5,\varepsilon})}{\varepsilon}$. By definition, we have $K_0 \leq 12000 BSA\ell_{4,\varepsilon}$ and $\ell_{4,\varepsilon} = \log(1 + 270BH))$. First, applying Lemma 32 on $\log(1 + 270BH)$ with $C_2 = 270$, we obtain that

$$\ell_{4,\varepsilon} \leq \frac{\log(1+270)}{270} \cdot (270BH) \leq 6BH \,.$$

Therefore, we have

$$K_0 H \leq 12000 BHSA\ell_{4,\varepsilon} \leq 72000 B^2 H^2 SA \,. \tag{32}$$

To prove inequality (30), we use inequality (32) and proceed as follows:

$$\begin{aligned}
\ell_{2,K_0} &= \log\left(1 + \frac{K_0 H}{SA}\right) \\
&= \log\left(1 + 72000 B^2 H^2\right) \\
&\leq 2\log\left(1 + \sqrt{72000}BH\right) \\
&\leq 2\log(1 + 270BH) \\
&= 2\ell_{4,\varepsilon} \,,
\end{aligned}$$

where the first inequality holds since $1 + x^2 \leq (1+x)^2$ for all $x \geq 0$.

To prove inequality (31), we need to further bound $B$. Since $\frac{24HSA}{\delta} \geq 48$, applying Lemma 32 with $C_1 = 48$ yields

$$\ell_1 \leq \frac{\log 48}{48} \cdot \frac{24HSA}{\delta} \leq \frac{2HSA}{\delta} \,. \tag{33}$$

Applying $\log x \leq x/e$, we obtain that $\ell_{5,\varepsilon} \leq 1 + \log(H/\varepsilon) \leq 1 + H/(e\varepsilon) \leq 2H/\varepsilon$. Then, it holds that

$$\begin{aligned}
2\ell_1 + \ell_{5,\varepsilon} &\leq \frac{4HSA}{\delta} + \frac{2H}{\varepsilon} \\
&\leq \frac{4HSA}{\delta} \cdot \frac{H}{\varepsilon} + \frac{HSA}{\delta} \cdot \frac{H}{\varepsilon} \\
&= \frac{5H^2 SA}{\delta\varepsilon} \,,
\end{aligned} \tag{34}$$

where the second inequality uses that $H/\varepsilon \geq 1$ and $HSA/\delta \geq 2$. Then, we bound $B$ as follows:

$$\begin{aligned}
B &= \frac{H^2\ell_1}{\varepsilon^2} + \frac{HS(2\ell_1+\ell_{5,\varepsilon})}{\varepsilon} \\
&\leq \frac{4H^3 SA}{\delta\varepsilon^2} + \frac{5H^3 S^2 A}{\delta\varepsilon^2} \\
&\leq \frac{9H^3 S^2 A}{\delta\varepsilon^2} \,,
\end{aligned} \tag{35}$$

where the first inequality applies inequalities (33) and (34) simultaneously. Utilizing these bounds, we derive an upper bound of $\log K_0 H$ as follows:

$$\log K_0 H \leq \log 72000 B^2 H^2 SA$$
$$\leq \log \frac{5832000 H^8 S^5 A^3}{\delta^2 \varepsilon^4}$$
$$\leq \log \frac{106817 H^4 S^5 A^3}{\delta^2} + \log \frac{e^4 H^4}{\varepsilon^4} ,$$

where the first inequality applies inequality (32), the second inequality comes from inequality (35), and the last inequality uses that $5832000/e^4 \leq 106871$. The first term can be further bounded as follows:

$$\log \frac{106817 H^4 S^5 A^3}{\delta^2} \leq \log \frac{26705 H^5 S^5 A^5}{\delta^5}$$
$$\leq 5 \log \frac{8 HSA}{\delta}$$
$$\leq \frac{7 HSA}{\delta} ,$$

where the first inequality uses that $H \geq 1, \delta \leq 1$, and $A \geq 2$, the second inequality holds since $26705 \leq 8^5 = 32768$, and the last inequality is due to Lemma 32 with $C_1 = 16$ and $5 \times 8 \times (\log 16)/16 \leq 7$. Using these results, we further bound $\log K_0 H$ as follows:

$$\log K_0 H \leq \frac{7 HSA}{\delta} + 4 \log \frac{eH}{\varepsilon}$$
$$\leq \frac{7 HSA}{\delta} \log \frac{eH}{\varepsilon} + \frac{2 HSA}{\delta} \log \frac{eH}{\varepsilon}$$
$$= \frac{9 HSA}{\delta} \log \frac{eH}{\varepsilon} , \tag{36}$$

where the second inequality uses $\log(eH/\varepsilon) \geq 1$ and $HSA/\delta \geq 2$. We conclude that inequality (31), the bound of $\ell_{3,K_0}$, is true by the following steps:

$$\ell_{3,K_0} = \log \frac{2SA \log K_0 H}{\delta}$$
$$\leq \log \frac{18 HS^2 A^2 \log \frac{eH}{\varepsilon}}{\delta^2}$$
$$\leq \log \frac{16 H^2 S^2 A^2}{\delta^2} + \log \frac{9}{8} + \log \log \frac{eH}{\varepsilon}$$
$$\leq 2 \log \frac{4 HSA}{\delta} + 1 + \log \log \frac{eH}{\varepsilon}$$
$$\leq 2\ell_1 + \ell_5 ,$$

where the first inequality holds by inequality (36), and the last inequality uses $\log(9/8) \leq 1$. $\qquad\square$

*Proof of Lemma 23.* Note that $K_0 \geq 12000 SA$, hence $\ell_{2,K_0} \geq 1$ holds. Then, we have $48 HSA\ell_1/\varepsilon \leq 48 HSA\ell_1\ell_{2,K_0}/\varepsilon \leq 48 H^2 SA\ell_1\ell_{2,K_0}/\varepsilon^2$, therefore it is sufficient to prove that

$$\frac{5680 H^2 SA\ell_1\ell_{2,K_0}}{\varepsilon^2} + \frac{4672 HS^2 A\ell_{2,K_0}\ell_{3,K_0}}{\varepsilon} \leq K_0 .$$

Applying Lemma 26, we conclude that

$$\frac{5680H^2SA\ell_1\ell_{2,K_0}}{\varepsilon^2} + \frac{4672HS^2A\ell_{2,K_0}\ell_{3,K_0}}{\varepsilon}$$

$$\leq \frac{11360H^2SA\ell_1\ell_{4,\varepsilon}}{\varepsilon^2} + \frac{4672HS^2A(2\ell_1 + \ell_{5,\varepsilon})\ell_{4,\varepsilon}}{\varepsilon}$$

$$\leq \frac{11362H^2SA\ell_1\ell_{4,\varepsilon}}{\varepsilon^2} + \frac{4672HS^2A(2\ell_1 + \ell_{5,\varepsilon})\ell_{4,\varepsilon}}{\varepsilon} - 2$$

$$\leq \frac{12000H^2SA\ell_1\ell_{4,\varepsilon}}{\varepsilon^2} + \frac{5000HS^2A(2\ell_1 + \ell_{5,\varepsilon})\ell_{4,\varepsilon}}{\varepsilon} - 2$$

$$\leq \left\lfloor \frac{12000H^2SA\ell_1\ell_{4,\varepsilon}}{\varepsilon^2} + \frac{5000HS^2A(2\ell_1 + \ell_{5,\varepsilon})\ell_{4,\varepsilon}}{\varepsilon} \right\rfloor - 1$$

$$= K_0 - 1 < K_0 \,,$$

where the first inequality applies the results of Lemma 26 simultaneously, and the second inequality uses that $H^2SA\ell_1\ell_{4,\varepsilon}/\varepsilon^2 \geq 1$. $\qquad\square$

## E  TECHNICAL LEMMAS

**Lemma 27.** *Let $C \geq 0$ be a constant. Let $\{V_h\}_{h=1}^{H+1}$ be a sequence of $H + 1$ functions such that $V_h : \mathcal{S} \to [0, C]$ for all $h \in [H + 1]$. For any $(s, a) \in \mathcal{S} \times \mathcal{A}$, the variance of $V_{h+1}$ under $P(\cdot|s, a)$ is bounded as follows:*

$$\mathrm{Var}(V_{h+1})(s, a) \leq -\Delta_h(V^2)(s, a) + 2C \max\{\Delta_h(V)(s, a), 0\} \,.$$

*Equivalently, the following inequality holds:*

$$\mathrm{Var}(V_{h+1})(s, a) - P(V_{h+1})^2(s, a) \leq -(V_h(s))^2 + 2C \max\{\Delta_h(V)(s, a), 0\} \,.$$

*Proof.* We add and subtract $(V_h(s))^2$ to $\mathrm{Var}(V_{h+1})(s, a)$ and obtain the following:

$$\mathrm{Var}(V_{h+1})(s, a) = P(V_{h+1})^2(s, a) - (PV_{h+1}(s, a))^2$$

$$= \underbrace{P(V_{h+1})^2(s, a) - (V_h(s))^2}_{I_1} + \underbrace{(V_h(s))^2 - (PV_{h+1}(s, a))^2}_{I_2} \,.$$

We have $I_1 = -\Delta_h(V^2)(s, a)$ by definition. We bound $I_2$ as follows:

$$I_2 = (V_h(s) + PV_{h+1}(s, a))(V_h(s) - PV_{h+1}(s, a))$$

$$\leq 2C \max\{\Delta_h(V)(s, a), 0\} \,,$$

where the inequality uses that $0 \leq V_h(s) + PV_{h+1}(s, a) \leq 2C$ and the definition of $\Delta_h(V)(s, a)$. Plugging in these bounds for $I_1$ and $I_2$ proves the first inequality of the lemma.

The second inequality is obtained by subtracting $P(V_{h+1})^2(s, a)$ from both sides of the first inequality and using that $-\Delta_h(V^2)(s, a) - P(V_{h+1})^2(s, a) = -(V_h)^2(s, a)$ by definition. $\qquad\square$

**Lemma 28.** *For any $(s, a) \in \mathcal{S} \times \mathcal{A}$ and $h \in [H]$, it holds that $\Delta_h(V^*)(s, a) \geq r(s, a) \geq 0$.*

*Proof.* This inequality is due to the Bellman optimality equation:

$$\Delta_h(V^*)(s, a) = V_h^*(s) - PV_{h+1}^*(s, a)$$

$$= \max_{a' \in \mathcal{A}}(r(s, a') + PV_{h+1}^*(s, a')) - PV_{h+1}^*(s, a)$$

$$\geq (r(s, a) + PV_{h+1}^*(s, a)) - PV_{h+1}^*(s, a)$$

$$= r(s, a) \geq 0 \,.$$

$\qquad\square$

**Lemma 29.** *Let $C > 0$ be a constant. Under the event of Lemma 7, the following inequality holds for all $(s, a) \in \mathcal{S} \times \mathcal{A}$, $k \in \mathbb{N}$, $\rho > 0$, and $V : \mathcal{S} \to [-C, C]$:*

$$\left|(\hat{P}^k - P)V(s, a)\right| \leq \frac{1}{C\rho}\mathrm{Var}(V)(s, a) + \frac{C(2\rho + 1)S\ell_{3,k}(s, a)}{N^k(s, a)} \,.$$

*Proof.* Without loss of generality, we assume that $PV(s, a) = 0$ and $C = 1$ since the inequality is invariant under constant translations and scalings of $V$. By Lemma 7, for any $(s, a, s') \in \mathcal{S} \times \mathcal{A} \times \mathcal{S}$, it holds that

$$\left| \hat{P}^k(s' \mid s, a) - P(s' \mid s, a) \right| \leq 2\sqrt{\frac{2P(s' \mid s, a)\ell_{3,k}(s, a)}{N^k(s, a)}} + \frac{2\ell_{3,k}(s, a)}{3N^k(s, a)}.$$

Multiplying both sides by $|V(s')|$ and using that $|V(s')| \leq C = 1$, we obtain that

$$\left| \left( \hat{P}^k(s'|s, a) - P(s'|s, a) \right) V(s') \right| \leq 2|V(s')|\sqrt{\frac{2P(s' \mid s, a)\ell_{3,k}(s, a)}{N^k(s, a)}} + \frac{2\ell_{3,k}(s, a)}{3N^k(s, a)}.$$

We apply the AM-GM inequality, $2ab \leq a^2/\rho + \rho b^2$ for any $a, b, \rho > 0$, on the first term of the right-hand side with $a = \sqrt{P(s'|s, a)}|V(s')|$ and $b = \sqrt{2\ell_{3,k}(s, a)/N^k(s, a)}$:

$$2|V(s')|\sqrt{\frac{2P(s' \mid s, a)\ell_{3,k}(s, a)}{N^k(s, a)}} + \frac{2\ell_{3,k}(s, a)}{3N^k(s, a)}$$

$$\leq \frac{1}{\rho}P(s' \mid s, a)(V(s'))^2 + \frac{2\rho\ell_{3,k}(s, a)}{N^k(s, a)} + \frac{2\ell_{3,k}(s, a)}{3N^k(s, a)}$$

$$\leq \frac{1}{\rho}P(s' \mid s, a)(V(s'))^2 + \frac{(2\rho + 1)\ell_{3,k}(s, a)}{N^k(s, a)},$$

which implies that

$$\left| \left( \hat{P}^k(s'|s, a) - P(s'|s, a) \right) V(s') \right| \leq \frac{1}{\rho}P(s' \mid s, a)(V(s'))^2 + \frac{(2\rho + 1)\ell_{3,k}(s, a)}{N^k(s, a)}. \tag{37}$$

Taking the sum over $s' \in \mathcal{S}$, we obtain that

$$\left| (\hat{P}^k - P)V(s, a) \right| = \left| \sum_{s' \in \mathcal{S}} \left( \hat{P}^k(s' \mid s, a) - P(s' \mid s, a) \right) V(s') \right|$$

$$\leq \sum_{s' \in \mathcal{S}} \left| \left( \hat{P}^k(s' \mid s, a) - P(s' \mid s, a) \right) V(s') \right|$$

$$\leq \sum_{s' \in \mathcal{S}} \left( \frac{1}{\rho}P(s' \mid s, a)(V(s'))^2 + \frac{(2\rho + 1)\ell_{3,k}(s, a)}{N^k(s, a)} \right)$$

$$= \frac{1}{\rho}\mathrm{Var}(V)(s, a) + \frac{(2\rho + 1)S\ell_{3,k}(s, a)}{N^k(s, a)},$$

where the first inequality is the triangle inequality, the second is inequality (37), and the last equality is by $PV(s, a) = 0$, which implies $\mathrm{Var}(V)(s, a) = P(V^2)(s, a)$. $\square$

**Lemma 30.** *For any sequence of $K$ trajectories, we have*

$$\sum_{k=1}^{K} \mathbb{1}\{\eta^k \neq H + 1\} \leq SA \log_2 2H$$

*and*

$$\sum_{k=1}^{K} \mathbb{1}\{\bar{\eta}^k \neq H + 1\} \leq SA \log_2 2H.$$

*Proof.* We only prove the first inequality, as the proof for the second inequality is identical. We focus on the state-action pair that triggers the stopping of $\eta^k$:

$$\sum_{k=1}^{K} \mathbb{1}\{\eta^k \neq H + 1\} = \sum_{k=1}^{K} \sum_{(s,a) \in \mathcal{S} \times \mathcal{A}} \mathbb{1}\{\eta^k \neq H + 1, (s_{\eta^k}^k, a_{\eta^k}^k) = (s, a)\}$$

$$= \sum_{(s,a) \in \mathcal{S} \times \mathcal{A}} \sum_{k=1}^{K} \mathbb{1}\{\eta^k \neq H + 1, (s_{\eta^k}^k, a_{\eta^k}^k) = (s, a)\}.$$

If $\eta^k \neq H + 1$, then by definition, it implies that $n_{\eta^k}^k(s_{\eta^k}^k, a_{\eta^k}^k) = 2N^k(s_{\eta^k}^k, a_{\eta^k}^k) + 1$, which in turn implies that $N^{k+1}(s_{\eta^k}^k, a_{\eta^k}^h) \geq 2N^k(s_{\eta^k}^k, a_{\eta^k}^k) + 1$. For any $(s, a) \in \mathcal{S} \times \mathcal{A}$ and $K \in \mathbb{N}$, let $M_K(s, a)$ be the number of $k \in [K]$ such that $N^{k+1}(s, a) \geq 2N^k(s, a) + 1$. Then, we infer that

$$\sum_{k=1}^{K} \mathbb{1}\{\eta^k \neq H + 1, (s_{\eta^k}^k, a_{\eta^k}^k) = (s, a)\} \leq M_K(s, a).$$

Now, it is sufficient to prove that $M_K(s, a) \leq \log_2 2H$ for all $(s, a) \in \mathcal{S} \times \mathcal{A}$. Since $N^{k+1}(s, a) \leq N^k(s, a) + H$, we infer that $N^{k+1}(s, a) \geq 2N^k(s, a) + 1$ occurs only if $N^k(s, a) < H$. On the other hand, using induction on $k$, one can prove that $N^{k+1}(s, a) \geq 2^{M_k(s,a)} - 1$ holds for all $k \in \mathbb{N}$. Hence, once $M_k(s, a)$ attains the value $\lfloor \log_2 H \rfloor + 1$ for some $k$, we have $N^{k+1}(s, a) \geq H$. Therefore, $M_k(s, a)$ does not increase after it reaches $\lfloor \log_2 H \rfloor + 1$, implying that $M_K(s, a) \leq \lfloor \log_2 H \rfloor + 1 \leq \log_2 2H$ for all $K \in \mathbb{N}$. $\qquad\square$

**Lemma 31.** *Let $\{\tau^k\}_{k=1}^{\infty}$ be any sequence of trajectories with $\tau^k = (s_1^k, a_1^k, R_1^k, \ldots, s_{H+1}^k)$. Let $\{\gamma_k\}_{k=1}^{\infty}$ be a sequence of increasing positive real numbers. Then, it holds that for any $K \in \mathbb{N}$,*

$$\sum_{k=1}^{K} \sum_{h=1}^{\eta^k - 1} \frac{\gamma_k}{N^k(s_h^k, a_h^k)} \leq 2\gamma_k SA \log\left(1 + \frac{KH}{SA}\right).$$

*Proof.* By the stopping rule of $\eta^k$, we have $N^k(s_h^k, a_h^k) \geq \frac{1}{2} n_h^k(s_h^k, a_h^k)$ when $h < \eta^k$. It also implies that when $h < \eta^k$, $n^k(s_h^k, a_h^k) \geq 2$ must hold. Hence, we have that

$$\begin{aligned}
\sum_{k=1}^{K} \sum_{h=1}^{\eta^k - 1} \frac{\gamma_k}{N^k(s_h^k, a_h^k)} &\leq \sum_{k=1}^{K} \sum_{h=1}^{\eta^k - 1} \frac{2\gamma_k}{n_h^k(s_h^k, a_h^k)} \\
&\leq \sum_{(s,a) \in \mathcal{S} \times \mathcal{A}} \sum_{n=2}^{N^{K+1}(s,a)} \frac{2\gamma_k}{n} \\
&\leq 2\gamma_K \sum_{(s,a) \in \mathcal{S} \times \mathcal{A}} \mathbb{1}\{N^{K+1}(s, a) \geq 2\} \log N^{K+1}(s, a) \\
&\leq 2\gamma_K \sum_{(s,a) \in \mathcal{S} \times \mathcal{A}} \log(1 + N^{K+1}(s, a)).
\end{aligned}$$

Since $\log(1 + x)$ is concave, applying Jensen's inequality implies that

$$\sum_{(s,a) \in \mathcal{S} \times \mathcal{A}} \log(1 + N^{K+1}(s, a)) \leq SA \log\left(\frac{\sum_{(s,a) \in \mathcal{S} \times \mathcal{A}}(1 + N^{K+1}(s, a))}{SA}\right)$$

$$= SA \log\left(1 + \frac{KH}{SA}\right).$$

$\qquad\square$

**Lemma 32.** *For any constant $C_1 \geq e$, if $x \geq C_1$, then $\log x \leq \frac{\log C_1}{C_1} x$. Also, for any constant $C_2 > 0$, we have $\log(1+x) \leq \frac{\log(1+C_2)}{C_2} x$ for $x \geq C_2$, and $\log(1+x) \geq \frac{\log(1+C_2)}{C_2} x$ for $0 < x \leq C_2$.*

*Proof.* By elementary calculus, one can check that $(\log x)/x$ decreases on $[e, \infty)$. Then, $x \geq C_1 \geq e$ implies $(\log x)/x \leq (\log C_1)/C_1$, which proves the first inequality. For the second inequality, note that $\log(1 + x)$ is concave, hence $g(x) := \frac{\log(1+C_2)}{C_2} x - \log(1 + x)$ is convex. Note that $g(0) = g(C_2) = 0$, therefore by its convexity, we have that $g(x) \geq 0$ whenever $x \geq C_2$ and $g(x) \leq 0$ when $0 < x \leq C_2$. $\qquad\square$

**Lemma 33.** *For $m \in \mathbb{N} \cup \{0\}$ and a constant $C \geq 3$, we define the following function:*

$$f(m) := \min\left\{1, \frac{25SA(C + 2\log(1 + m))\log\left(1 + \frac{2^m H}{SA}\right)}{2^m}\right\}.$$

*Then, $f$ is non-increasing.*

*Proof.* We directly show $f(m) \geq f(m+1)$ for any $m$. We write $z := 2^m/(SA)$, so $f(m) = \min\{1, 25(C + 2\log(1+m))(\log(1+Hz))/z\}$. Let $m_0 := \max\{m \in \mathbb{N} \cup \{0\} : 2^m \leq 25SA\}$. We deal with two cases, $m \leq m_0$ and $m > m_0$, separately.

**Case 1** $m \leq m_0$ : We show that $f(m) = 1$ for $m \leq m_0$, which implies $f(m) \geq f(m+1)$. First, we have that $25(C + 2\log(1+m)) \geq 75$ by $C \geq 3$ and $\log(1+m) \geq 0$. Thus, we must show that $(75\log(1+Hz))/z \geq 1$. Note that $H \geq 1$ and $z \leq 25$ when $m \leq m_0$, therefore it is sufficient to prove that $\frac{75}{z}\log(1+z) \geq 1$ for $z \leq 25$. By Lemma 32 with $C_2 = 25$, we have that $\log(1+x) \geq \frac{\log(1+25)}{25}x$ for $x \leq 25$, hence we have $\frac{75}{z}\log(1+z) \geq \frac{75\log 26}{25} \geq 1$.

**Case 2** $m > m_0$: We prove that $f(m) \geq f(m+1)$ by showing that the second argument of the minimum in the definition of $f$ is decreasing when $m > m_0$. Specifically, we show that

$$\frac{(C + 2\log(1+m))\log(1 + \frac{2^m H}{SA})}{2^m} \geq \frac{(C + 2\log(2+m))\log(1 + \frac{2^{m+1}H}{SA})}{2^{m+1}} .$$

Rearranging the terms and plugging in $z = 2^m/(SA)$, one can see that it is sufficient to prove

$$(C + 2\log(2+m))\log(1 + 2Hz) \leq 2(C + 2\log(1+m))\log(1 + Hz) . \tag{38}$$

First, we bound $C + 2\log(2+m)$ as follows:

$$\begin{aligned} C + 2\log(2+m) &= C + 2(\log(1+m/2) + \log 2) \\ &\leq C + 2\log(1+m) + 1.5 \\ &\leq \frac{3}{2}C + 3\log(1+m) \\ &= \frac{3}{2}(C + 2\log(1+m)) , \end{aligned}$$

where the second inequality uses that $C \geq 3$ and $\log(1+m) \geq 0$. We bound $\log(1 + 2Hz)$ as follows:

$$\begin{aligned} \log(1 + 2Hz) &\leq \log 2 + \log(\frac{1}{2} + Hz) \\ &\leq 1 + \log(1 + Hz) \\ &\leq \frac{4}{3}\log(1 + Hz) , \end{aligned}$$

where the last inequality uses that $\log(1 + Hz) \geq 3$ when $z \geq 25$, which holds since $m > m_0$. As we have derived $C + 2\log(2+m) \leq \frac{3}{2}(C + 2\log(1+m))$ and $\log(1 + 2Hz) \leq \frac{4}{3}\log(1 + Hz)$, by multiplying the two inequalities we conclude that inequality (38) holds. □

**Lemma 34.** *Let $C_1 \geq C_2 > 0$ be constants. Let $f(x) = \frac{1}{x}(C_1 + C_2\log(1 + \log x)))$ for $x > 0$. Then, $f$ is non-increasing on $x \geq 1$.*

*Proof.* Taking the derivative of $f$, we obtain that

$$f'(x) = \frac{\frac{C_2}{1+\log x} - C_1 - C_2\log(1 + \log x)}{x^2} .$$

Note that the numerator is decreasing in $x$, and when plugging in $x = 1$, the numerator becomes $C_2 - C_1 \leq 0$. Therefore, we have that $f'(x) \leq 0$ for all $x \geq 1$. □

**Lemma 35** (Lemma 30 in Chen et al. (2021)). *Let $C \geq 0$ be a constant and $X$ be a random variable such that $|X| \leq C$ almost surely. Then, $\mathrm{Var}(X^2) \leq 4C^2 \mathrm{Var}(X)$.*

## F  CONCENTRATION INEQUALITIES

All the concentration inequalities used in the analysis are based on the following proposition, which is derived by following the proof of Theorem (1.6) in Freedman (1975).

**Proposition 4.** *Let $\{X_t\}_{t=1}^{\infty}$ be a martingale difference sequence with respect to a filtration $\{\mathcal{F}_t\}_{t=0}^{\infty}$. Suppose $X_t \leq 1$ holds almost surely for all $t \geq 1$. Let $V_t := \mathbb{E}[X_t^2|\mathcal{F}_{t-1}]$ for $t \geq 1$ and take $\lambda > 0$ and $\delta \in (0,1]$ arbitrarily. Then, the following inequality holds for all $n \in \mathbb{N}$ with probability at least $1 - \delta$:*

$$\sum_{t=1}^{n} X_t \leq \frac{e^\lambda - 1 - \lambda}{\lambda} \sum_{t=1}^{n} V_t + \frac{1}{\lambda} \log \frac{1}{\delta}. \tag{39}$$

*Proof.* Let $M_n = \exp(\sum_{t=1}^{n}(X_t - ((e^\lambda - 1 - \lambda)/\lambda)V_t))$ for all $n \in \mathbb{N}$, where $M_0 = 1$. Corollary 1.4 (a) in Freedman (1975) states that $\{M_n\}_{n=0}^{\infty}$ is a supermartingale with respect to $\{\mathcal{F}_n\}_{n=0}^{\infty}$. By Ville's maximal inequality, we infer that $\mathbb{P}(\sup_{n \geq 0} M_n \geq 1/\delta) \leq \delta$, which implies that $\mathbb{P}(\forall n, M_n \leq 1/\delta) \geq 1 - \delta$. Taking the logarithm on both sides and rearranging the terms, we check that $M_n \leq 1/\delta$ is equivalent to inequality (39), completing the proof. $\square$

We mainly use the following two corollaries of Proposition 4. The first one has appeared in the literature several times (Beygelzimer et al., 2011; Agarwal et al., 2014; Xu & Zeevi, 2020; Foster & Rakhlin, 2023).

**Lemma 36.** *Let $C > 0$ be a constant and $\{X_t\}_{t=1}^{\infty}$ be a martingale difference sequence with respect to a filtration $\{\mathcal{F}_t\}_{t=0}^{\infty}$ with $X_t \leq C$ almost surely for all $t \in \mathbb{N}$. Then, for any $\lambda \in (0,1]$ and $\delta \in (0,1]$, the following inequality holds for all $n \in \mathbb{N}$ with probability at least $1 - \delta$:*

$$\sum_{t=1}^{n} X_t \leq \frac{3\lambda}{4C} \sum_{t=1}^{n} \mathbb{E}[X_t^2|\mathcal{F}_{t-1}] + \frac{C}{\lambda} \log \frac{1}{\delta}.$$

*Proof.* For $\lambda \in (0,1]$, it holds that $e^\lambda \leq 1 + \lambda + (e - 2)\lambda^2$, hence, $\frac{e^\lambda - 1 - \lambda}{\lambda} \leq (e - 2)\lambda$. Let $X_t' = X_t/C$. Applying Proposition 4 on $\{X_t'\}_{t=1}^{\infty}$, we obtain that

$$\sum_{t=1}^{n} X_t' \leq (e - 2)\lambda \sum_{t=1}^{n} \mathbb{E}[(X_t')^2|\mathcal{F}_{t-1}] + \frac{1}{\lambda} \log \frac{1}{\delta}$$

holds for all $n \in \mathbb{N}$ with probability at least $1 - \delta$. Bounding $e - 2 \leq 3/4$ and multiplying both sides by $C$ completes the proof since $CX_t' = X_t$ and $C\mathbb{E}[(X_t')^2|\mathcal{F}_{t-1}] = \mathbb{E}[X_t^2|\mathcal{F}_{t-1}]/C$. $\square$

The second corollary is a time-uniform version of Bernstein's inequality for martingales that incorporates a $\log \log n$ factor instead of $\log n$.

**Lemma 37.** *Let $\{X_t\}_{t=1}^{\infty}$ be a martingale difference sequence with respect to a filtration $\{\mathcal{F}_t\}_{t=0}^{\infty}$. Assume that $X_t \leq 1$ holds almost surely for all $t \geq 1$. Let $T_n = \sum_{t=1}^{n} \mathbb{E}[X_t^2 \mid \mathcal{F}_{t-1}]$ be the sum of conditional variances. Then, for any $c > 0$ and $\delta \in (0,1]$, the following inequality holds for all $n \in \mathbb{N}$ with probability at least $1 - \delta$:*

$$\sum_{t=1}^{n} X_t \leq 2\sqrt{\max\{T_n, c\} \log \frac{2(1 + \log^+(T_n/c))^2}{\delta}} + \frac{1}{3} \log \frac{2(1 + \log^+(T_n/c))^2}{\delta},$$

*where $\log^+ x := \log(\max\{1, x\})$.*

*Proof.* Let $\{\lambda_j\}_{j=1}^{\infty}$ be a sequence that satisfies $0 < \lambda_j < 3$ for all $j \in \mathbb{N}$. We apply Proposition 4 with $\lambda \leftarrow \lambda_j$ and $\delta \leftarrow \delta/(2j^2)$ for $j \in \mathbb{N}$. For fixed $j$, we have

$$\mathbb{P}\left(\exists n \in \mathbb{N} : \sum_{t=1}^{n} X_t > \frac{e^{\lambda_j} - 1 - \lambda_j}{\lambda_j} T_n + \frac{1}{\lambda_j} \log \frac{2j^2}{\delta}\right) \leq \frac{\delta}{2j^2}.$$

Using the Taylor series expansion, we verify that the following holds for $0 < \lambda_j < 3$:

$$\frac{e^{\lambda_j} - 1 - \lambda_j}{\lambda_j} = \sum_{k=2}^{\infty} \frac{\lambda_j^{k-1}}{k!} \leq \sum_{k=2}^{\infty} \frac{\lambda_j^{k-1}}{2 \cdot 3^{k-2}} = \frac{\lambda_j}{2(1 - \frac{\lambda_j}{3})}.$$

Therefore, we have that with probability at least $1 - \delta/(2j^2)$, the following inequality holds for all $n \in \mathbb{N}$:

$$\sum_{t=1}^{n} X_t \leq \frac{\lambda_j}{2(1 - \frac{\lambda_j}{3})} T_n + \frac{1}{\lambda_j} \log \frac{2j^2}{\delta} . \tag{40}$$

We take

$$\lambda_j = \frac{\sqrt{\log \frac{2j^2}{\delta}}}{\sqrt{c} e^{\frac{j-1}{2}} + \frac{1}{3}\sqrt{\log \frac{2j^2}{\delta}}} .$$

One can check that $0 < \lambda_j < 3$. We have

$$\frac{\lambda_j}{2\left(1 - \frac{\lambda_j}{3}\right)} = \frac{\sqrt{\log \frac{2j^2}{\delta}}}{2\sqrt{c} e^{\frac{j-1}{2}}} \qquad \text{and} \qquad \frac{1}{\lambda_j} = \frac{1}{3} + \frac{\sqrt{c} e^{\frac{j-1}{2}}}{\sqrt{\log \frac{2j^2}{\delta}}} .$$

Plugging in these values to inequality (40), we obtain the following inequality:

$$\sum_{t=1}^{n} X_t \leq \left( \frac{T_n}{2\sqrt{c} e^{\frac{j-1}{2}}} + \sqrt{c} e^{\frac{j-1}{2}} \right) \sqrt{\log \frac{2j^2}{\delta}} + \frac{1}{3} \log \frac{2j^2}{\delta} . \tag{41}$$

Taking the union bound over $j \in \mathbb{N}$ and noting that $\sum_{j=1}^{\infty} \frac{\delta}{2j^2} \leq \delta$, we obtain that inequality (41) holds for all $j$ and for all $n$ simultaneously with probability at least $1 - \delta$, which can be written in the following equivalent form:

$$\mathbb{P}\left( \exists n \in \mathbb{N} : \sum_{t=1}^{n} X_t \geq \min_{j \in \mathbb{N}} \left\{ \left( \frac{T_n}{2\sqrt{c} e^{\frac{j-1}{2}}} + \sqrt{c} e^{\frac{j-1}{2}} \right) \sqrt{\log \frac{2j^2}{\delta}} + \frac{1}{3} \log \frac{2j^2}{\delta} \right\} \right) \leq \delta .$$

Now, we choose $j = j_n$ for each $n \in \mathbb{N}$ and provide a bound for the minimum. We define $T'_n := \max\{T_n, c\}$ and take $j_n = 1 + \lfloor \log(T'_n/c) \rfloor$. Then, we have $e^{\frac{j_n-1}{2}} \leq \sqrt{T'_n/c}$ and $e^{-\frac{j_n-1}{2}} \leq \sqrt{ec/T'_n}$, implying that

$$\min_{j \in \mathbb{N}} \left\{ \left( \frac{T_n}{2\sqrt{c} e^{\frac{j-1}{2}}} + \sqrt{c} e^{\frac{j-1}{2}} \right) \sqrt{\log \frac{2j^2}{\delta}} + \frac{1}{3} \log \frac{2j^2}{\delta} \right\}$$

$$\leq \left( \frac{T_n}{2\sqrt{c} e^{\frac{j_n-1}{2}}} + \sqrt{c} e^{\frac{j_n-1}{2}} \right) \sqrt{\log \frac{2j_n^2}{\delta}} + \frac{1}{3} \log \frac{2j_n^2}{\delta}$$

$$\leq \left( \frac{T_n}{2} \sqrt{\frac{e}{T'_n}} + \sqrt{T'_n} \right) \sqrt{\log \frac{2j_n^2}{\delta}} + \frac{1}{3} \log \frac{2j_n^2}{\delta}$$

$$\leq 2\sqrt{T'_n \log \frac{2j_n^2}{\delta}} + \frac{1}{3} \log \frac{2j_n^2}{\delta} ,$$

where in the last inequality we apply $T_n \leq T'_n$ and $\sqrt{\frac{e}{2}} + 1 \leq 2$. Finally, noting that $j_n \leq 1 + \log(T'_n/c) = 1 + \log^+(T_n/c)$, we obtain that the following inequality holds for all $n \in \mathbb{N}$, with probability at least $1 - \delta$:

$$\sum_{t=1}^{n} X_t \leq 2\sqrt{\max\{T_n, c\} \log \frac{2(1 + \log^+(T_n/c))^2}{\delta}} + \frac{1}{3} \log \frac{2(1 + \log^+(T_n/c))^2}{\delta} .$$

$\square$

## G  EXPERIMENT DETAILS

In this section, we provide additional details for the experiments described in Section 5. Specific transitions and reward functions of the RiverSwim environment are depicted in Figure 2. All parameters are set according to the algorithms' theoretical values as described in their respective papers.

For `EQO`, the parameters are set as described in Theorem 2, where the algorithm is unaware of the number of episodes. The algorithm of `UCRL2` is modified to adapt to the episodic finite-horizon setting. We report the average cumulative regret and standard deviation over 10 runs of 100,000 episodes in Figure 3, with the average execution time per run summarized in Table 4.

We observe the superior performance of `EQO`. When $S$ and $H$ are small, only `ORLC` shows competitive performance against `EQO`, but our algorithm outperforms `ORLC` by increasing margins as $S$ and $H$ grow. Especially in the case where $S = 40$ and $H = 160$, only our algorithm learns the MDP within the given number of episodes and achieves sub-linear cumulative regret. We also note that our algorithm takes less execution time.

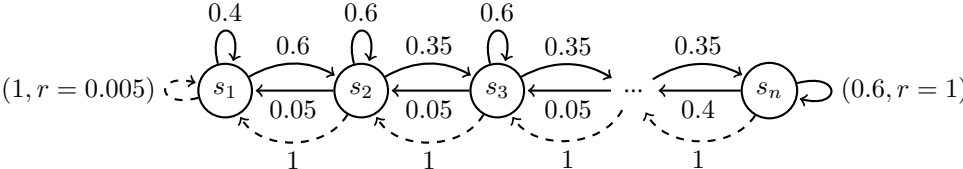

Figure 2: The RiverSwim MDP with $n$ states. The solid arrows and dashed arrows describe the state transitions of the "right" and "left"actions respectively with their probabilities labeled. $r = X$ shows the reward of the state-action pair, where $r = 0$ if not shown.

Table 4: Average execution time of each algorithm in seconds.

| Algorithm | $S = 10$ $H = 40$ | $S = 20$ $H = 80$ | $S = 30$ $H = 120$ | $S = 40$ $H = 160$ |
|---|---|---|---|---|
| UCRL2 (Jaksch et al., 2010) | 1899.5 | 7298.9 | 17541.9 | 22594.3 |
| UCBVI-BF (Azar et al., 2017) | 699.0 | 2171.4 | 4439.3 | 6785.6 |
| EULER (Zanette & Brunskill, 2019) | 991.0 | 2847.3 | 5643.7 | 8353.7 |
| ORLC (Dann et al., 2019) | 1219.4 | 3871.1 | 7408.7 | 11655.0 |
| MVP (Zhang et al., 2021a) | 523.4 | 2155.4 | 4106.5 | 6687.3 |
| EQO (Ours) | 535.2 | 1904.0 | 3847.1 | 6713.1 |

We report additional experiment results conducted under a MDP described in Figure 2 of Dann et al. (2021). It is a deterministic MDP where the reward is given only for the last action. If the agent has followed the optimal policy until the penultimate time step, it faces a state such that one action has an expected reward of 0.5 and the other has 0. If the agent's actions deviate from the optimal policy, then it receives an expected reward of either 0 or $\varepsilon = 0.02$, depending on the final action. Refer to Appendix C in Dann et al. (2021) for more details about the MDP. We report the average cumulative regret and standard deviation over 10 runs of 500,000 episodes in Figure 4. UCRL is excluded due to its high computational cost under large state space. We also add a tuned version of `EQO`, highlighting its strength when the parameter is set appropriately. `EQO` with theoretical parameters outperforms all other algorithms except `ORLC`. When appropriately tuned, `EQO` incurs the smallest regret by significant margins. While it may be unfair to compare the results of the algorithms with and without the tuning of the parameters, we draw the reader's attention to the complicated structure of the bonus terms of `ORLC`. As presented in Algorithm 3 in Dann et al. (2019), the bonus terms of the algorithm utilize at least twenty terms to estimate both upper and lower bounds of the optimal values, making it almost intractable to effectively tune the algorithm. For the other algorithms, their bonus terms also consist of multiple terms, being subject to the same problem. Only `EQO` offers comprehensive control over the algorithm through a single parameter. What we highlight here is not only the empirical performance of `EQO` but also the simplicity of the algorithm that makes it extremely convenient to tune.

We conduct additional experiments in two more complex environments: Atari 'freeway_10_fs30' (Bellemare et al., 2013) and Minigrid 'MiniGrid-KeyCorridorS3R1-v0' (Chevalier-Boisvert et al., 2023). We have obtained their tabularized versions from the BRIDGE dataset (Laidlaw et al., 2023). Most Atari and MiniGrid environments are either too large in terms of the number

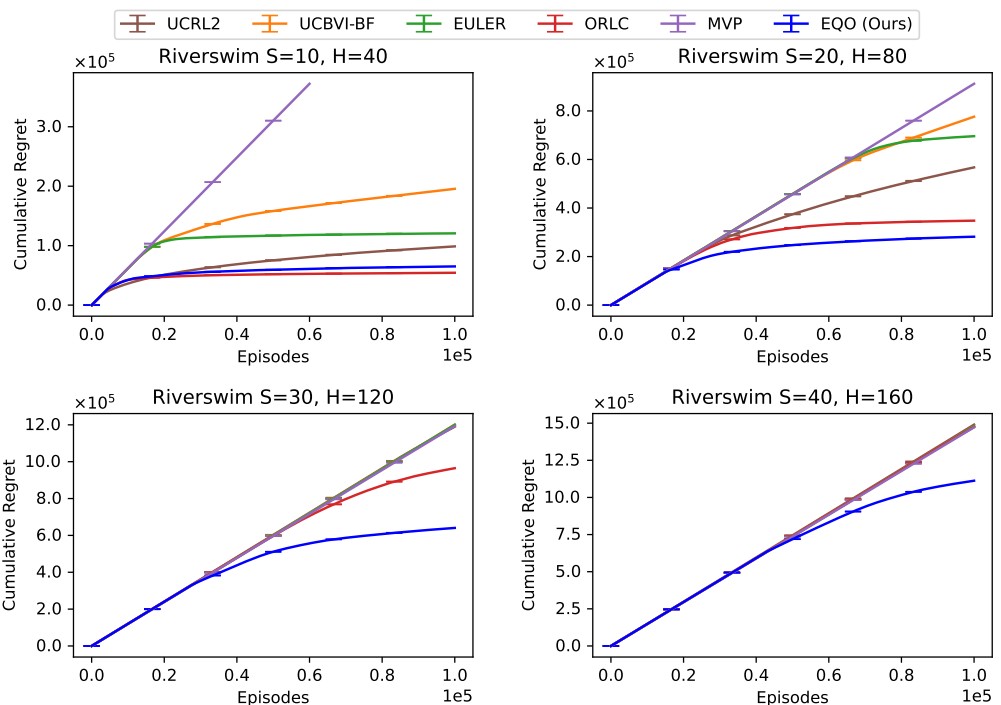

Figure 3: Cumulative regret under RiverSwim MDP with varying $S$ and $H$.

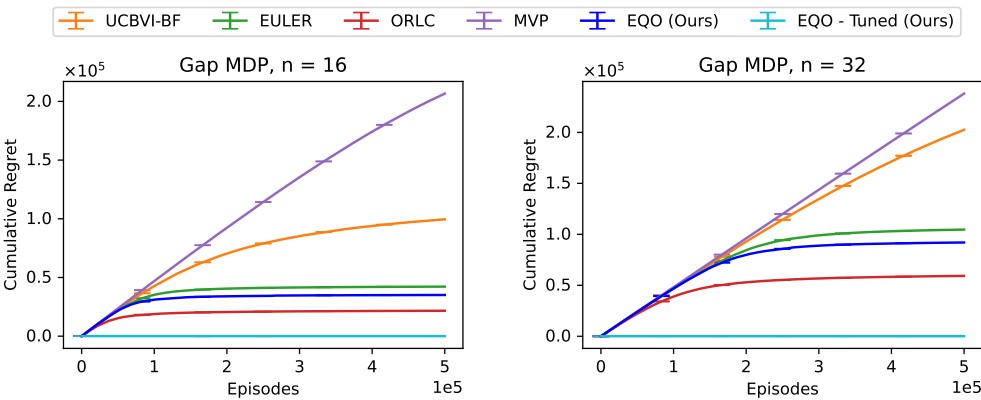

Figure 4: Cumulative regret under MDP described in Figure 2 of Dann et al. (2021) with varying $n$.

of states to perform tabular learning or too simple, where a greedy policy performs best, diminishing the purpose of comparing the efficiency of exploration strategies. These two specific environments are selected from each group for their moderate sizes and complexities. Both MDPs have over 150 states and more than two actions, making them much more complex than the RiverSwim MDP. We note that instead of the conventional 25% chance of sticky actions (Machado et al., 2018), we employ a 25% chance of random actions to preserve the Markov property.

We include PSRL (Osband et al., 2013), a randomized algorithm for tabular reinforcement learning, for a more diverse comparison, while UCRL2 is excluded due to its high computational cost in large state spaces. We report the average cumulative regret and standard deviation over 10 runs of 5,000 episodes in Figure 5. Considering these environments as more real-world-like settings, we tune each algorithm to achieve their best performance for each instance. For both settings, EQO consistently demonstrates competitive performance.

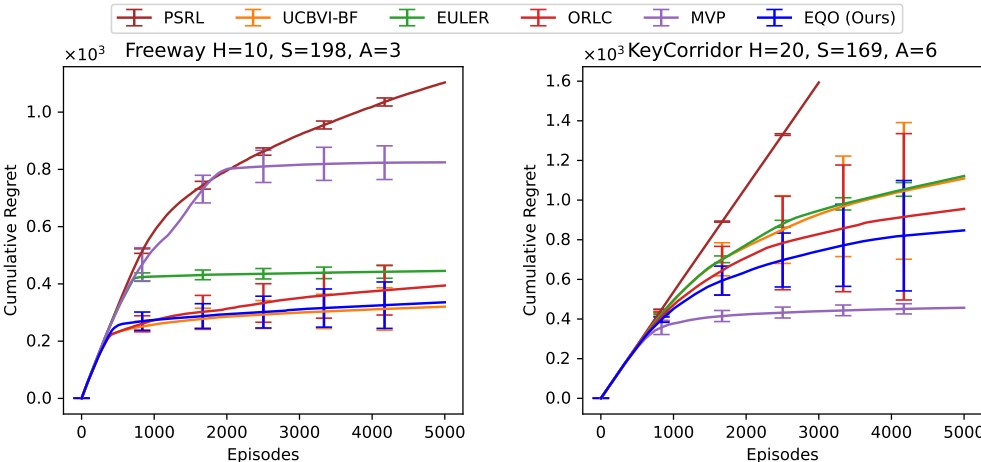

Figure 5: Cumulative regret under environments with larger state spaces.

**Comparing with Bayesian algorithms.** PSRL achieves a Bayesian regret guarantee for a given prior distribution over the MDPs (Osband et al., 2013); however, the prior is not available under the current experimental setting. While it is possible to construct an artificial prior, the performance of these algorithms depends on the prior; that is, they gain an advantage if the prior is informative. This makes it potentially unfair to compare them with algorithms that have frequentist regret guarantees, as the latter cannot use any prior information and must be more conservative. For example, RLSVI, another randomized algorithm, requires constant-scale perturbations for a Bayesian guarantee (Osband et al., 2016). However, the perturbations must be inflated by a factor of $HSA$ to ensure a frequentist regret bound (Zanette et al., 2020). Typically, the constant-scaled version empirically performs much better for most reasonable MDPs.

One way to make the comparison viable would be to tune the frequentist algorithms. As the purpose of the experiment with the RiverSwim MDP is to compare the performance of algorithms with theoretical guarantees, we set the parameters according to their theoretical values, and hence we do not tune the parameters and cannot include PSRL for comparison (since it does not have theoretical values for its parameters in this setting). In these two additional experiments, we consider their settings to be closer to real-world environments, where tuning the parameters becomes highly necessary. Therefore, we tune the algorithms and include PSRL. For EQO, tuning the algorithm is straightforward: treat $c_k$ in Algorithm 1 as a $k$-independent parameter as in Theorems 1, 3 and 4 and tune its value. However, as mentioned earlier, the other algorithms have multiple terms with complicated structures as bonuses, and how they should be tuned is not clear. For the results of Figure 5, a multiplicative factor for the whole bonus term is set as a tuning parameter, therefore maintaining the same complexity as EQO.

## H  EXPLOITING UNIFORM-REWARD SETTING

Although our Assumption 1 generalizes the uniform-reward setting, algorithms may perform better when prior information that the reward at each time step is at most 1 is available. Algorithm 3 describes how EQO may adapt to the uniform-reward setting. We show that the theoretical guarantees enjoyed by Algorithm 1 remain valid for Algorithm 3 under the uniform-reward setting, and provide additional experimental results that compare the performance of the algorithms when all of them exploit the uniform reward structure. Our algorithm continues to exhibit its superiority over the other algorithms.

### H.1  THEORETICAL GUARANTEES FOR UNIFORM-REWARD SETTING

In this subsection, we rigorously demonstrate that under the uniform-reward setting, Algorithm 3 enjoys the same theoretical guarantees as Theorems 1 to 4. Set $\lambda_k$ as defined in each of the theorems and let $c'_k = 7\ell_{1,k}/\lambda_k$, so that $c'_k H = c_k$. We note that under this choice of parameters, the bonus

---

**Algorithm 3:** EQO for Uniform-Reward Setting

**Input:** $\{c'_k\}_{k=1}^{\infty}$

1 **for** $k = 1, 2, \ldots, K$ **do**

2      Set $N^k(s, a)$, $\hat{r}^k(s, a)$, and $\hat{P}^k(s'|s, a)$ as in Algorithm 1;

3      **for** $h = H, H - 1, \ldots, 1$ **do**

4          **foreach** $(s, a) \in \mathcal{S} \times \mathcal{A}$ **do**

5              $b^k(s, a) \leftarrow c'_k(H - h + 1)/N^k(s, a)$;

6              $Q_h^k(s, a) \leftarrow \begin{cases} \min\left\{\hat{r}^k(s, a) + b^k(s, a) + \hat{P}^k V_{h+1}^k(s, a), H - h + 1\right\} & \text{if } N^k(s, a) > 0 \\ H - h + 1 & \text{if } N^k(s, a) = 0 \end{cases}$;

7          $V_h^k(s) \leftarrow \max_{a \in \mathcal{A}} Q_h^k(s, a)$ for all $s \in \mathcal{S}$;

8          $\pi_h^k(s) \leftarrow \operatorname{argmax}_{a \in \mathcal{A}} Q_h^k(s, a)$ for all $s \in \mathcal{S}$;

9      Execute $\pi^k$ and obtain $\tau^k = (s_1^k, a_1^k, R_h^k, \ldots, s_H^k, a_H^k, R_H^k, s_{H+1}^k)$;

---

term of Algorithm 3 is less than or equal to the bonus term of Algorithm 1. Therefore, the parts where we derive upper bounds for $b^k(s, a)$ in the analysis remain valid, and the only part where we need lower bounds for $b^k(s, a)$ is in the proof of Lemma 2, where we show the quasi-optimism. We show that quasi-optimism holds under the different choice of bonus terms when the reward structure is uniform, which implies that Algorithm 3 enjoys the same theoretical guarantees.

First, the high-probability events of Lemmas 5, 6 and 8 should be adjusted to the new bounds of $V_{h+1}^*$ and $R_h^k$. $V_h^*(s)$ is at most $H - h + 1$ for all $s \in \mathcal{S}$ and $R_h^k$ is at most 1. One can easily derive from the proofs that the inequalities of each lemma can be replaced with the following inequalities respectively:

$$\left|(\hat{P}^k - P)V_{h+1}^*(s, a)\right| \leq \frac{\lambda_k \mathbb{1}\{h \neq H\}}{4(H - h)} \operatorname{Var}(V_{h+1}^*)(s, a) + \frac{3(H - h)\ell_{1,k}}{\lambda_k N^k(s, a)},$$

$$(P - \hat{P}^k)(V_{h+1}^*)^2(s, a) \leq \frac{1}{2} \operatorname{Var}(V_{h+1}^*)(s, a) + \frac{6(H - h)^2 \ell_{1,k}}{N^k(s, a)},$$

$$\left|\hat{r}^k(s, a) - r(s, a)\right| \leq \lambda_k r(s, a) + \frac{\ell_{1,k}}{\lambda_k N^k(s, a)},$$

where we define $\mathbb{1}\{h \neq H\}/(H - h)$ to be 0 when $h = H$. Let $\mathcal{E}'$ be the counterpart of $\mathcal{E}$ such that the events of Lemmas 5, 6 and 8 are replaced by the events above.

Now, we show that Algorithm 3 enjoys quasi-optimism.

**Lemma 38** (Quasi-optimism for Algorithm 3). *Under $\mathcal{E}'$, it holds that for all $S \in \mathcal{S}$, $h \in [H]$, and $k \in \mathbb{N}$,*

$$V_h^k(s) + \frac{3}{2}\lambda_k(H - h + 1) \geq V_h^*(s).$$

*Proof.* We prove the following inequality by backward induction on $h \in [H + 1]$:

$$V_h^*(s) - V_h^k(s) \leq \lambda_k \left(2V_h^*(s) - \frac{\mathbb{1}\{h \neq H + 1\}}{2(H - h + 1)}(V_h^*)^2(s)\right).$$

It is easy to show that the inequality holds when $h = H + 1$ or $V_h^k(s) = H - h + 1$. Suppose $h \leq H$ and $V_h^k(s) < H - h + 1$, and that the inequality holds for $h + 1$. Following the proof of Lemma 2 with the refined concentration inequalities, we arrive at the following inequality:

$$V_h^*(s) - V_h^k(s) \leq -b^k(s, a^*) + \frac{(6(H - h) + 1)\ell_{1,k}}{\lambda_k N^k(s, a^*)} + \lambda_k r(s, a^*) + 2\lambda_k P V_{h+1}^*(s, a^*)$$

$$+ \frac{\lambda_k \mathbb{1}\{h \neq H\}}{2(H - h)} \left(\operatorname{Var}(V_{h+1}^*)(s, a^*) - P(V_{h+1}^*)^2(s, a^*)\right). \tag{42}$$

We first note that $b^k(s, a^*) = \frac{7(H-h+1)\ell_{1,k}}{\lambda_k N^k(s,a^*)} \geq \frac{(6(H-h)+1)\ell_{1,k}}{\lambda_k N^k(s,a^*)}$, therefore the sum of the first two terms is not greater than 0. Now, we bound the last term. Note that $\text{Var}(V_{h+1}^*)(s, a^*) - P(V_{h+1}^*)(s, a^*) = -(PV_{h+1}^*(s, a^*))^2$ is non-positive, therefore we have that

$$\frac{\lambda_k \mathbb{1}\{h \neq H\}}{2(H-h)} \left( \text{Var}(V_{h+1}^*)(s, a^*) - P(V_{h+1}^*)^2(s, a^*) \right)$$

$$\leq \frac{\lambda_k}{2(H-h+1)} \left( \text{Var}(V_{h+1}^*)(s, a^*) - P(V_{h+1}^*)^2(s, a^*) \right), \tag{43}$$

where the inequality also holds for $h = H$ as both sides are 0 in that case. Applying Lemma 27, we obtain that

$$\text{Var}(V_{h+1}^*)(s, a^*) - P(V_{h+1}^*)^2(s, a^*) \leq -(V_h^*)^2(s) + 2(H-h+1)r(s, a^*). \tag{44}$$

Combining inequalities (42),(43), and (44), we conclude that

$$V_h^*(s) - V_h^k(s) \leq \lambda_k r(s, a^*) + 2\lambda_k PV_{h+1}^*(s, a^*) - \frac{\lambda_k}{2(H-h+1)}(V_h^*)^2(s) + \lambda_k r(s, a^*)$$

$$= \lambda_k \left( 2V_h^*(s) - \frac{1}{2(H-h+1)}(V_h^*)^2(s) \right),$$

completing the induction argument. □

## H.2 Additional Experiments

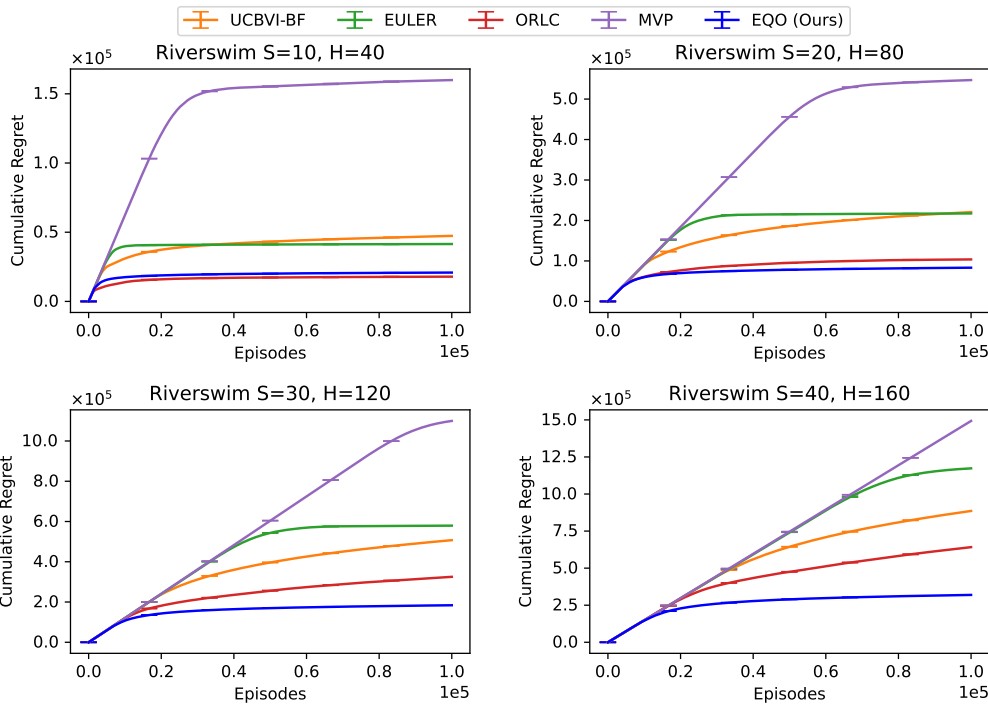

Figure 6: Cumulative regret of algorithms under the RiverSwim MDP with varying $S$ and $H$. All algorithms are aware of the uniform reward structure of the MDP.

We provide additional experimental results showing the performance of the algorithms when the uniform reward structure is exploited. Note that the RiverSwim MDP satisfies the uniform-reward assumption. To fairly compare algorithms that are analyzed under more general assumptions, we adjust them according to the following criteria:

(a) If the algorithms clip the estimated values by $H$, we adjust it to $H - h + 1$, which is the maximum possible value for any state at time step $h$.

    (b) If the algorithms have any terms proportional to $H$ in their bonus, we change them to $H-h$, as it is intuitive that the optimistic bonus term should not account for how many time steps have passed before the current step.

We note that in the experiment presented in Section 5, the algorithms are not allowed to exploit the uniform reward structure. Therefore, algorithms that originally exploit it underwent the opposite conversion for fair comparisons. We do not claim that these simple conversions ensure the validity of the analyses under the opposite assumptions. However, we believe that they are sufficient for numerical comparisons.

We display the results in Figure 6. We exclude UCRL2 as it is originally designed for MDPs without horizons, and we could not find any straightforward conversions for it. With the additional information, all the algorithms exhibit significant improvements in their performance when compared with Figure 3. We emphasize that our algorithm continues to demonstrate its superior performance over the other algorithms.

