# OpenReview forum: "Minimax Optimal Reinforcement Learning with Quasi-Optimism"
_ICLR.cc/2025/Conference — ICLR 2025 Poster_

### Official Review · Reviewer_ud3P · 2024-10-25

**Soundness:** 3
**Presentation:** 3
**Contribution:** 2
**Rating:** 6
**Confidence:** 4

**Summary:**

This paper studies the tabular MDP setting. They propose a new type of bonus for the UCRL algorithm, and prove that the regret of the algorithm matches the lower bound. They also show an upper bound to the PAC sample complexity, which matches to the lower bound as well.

**Strengths:**

1. The paper is well written. The proofs are correct.

2. The algorithms have regret which match the lower bound, and the algorithm is much simpler to those minimax optimal algorithms in previous literatures (e.g. [Azar et al,. 2017]) and do not need to estimate the variance explicitly.

**Weaknesses:**

1. In terms of the minimax optimality, this algorithm only matches the lower bound in the dominate terms. The low order terms does not match, while the regret of the algorithm in [1] matches the lower bound even for low order terms (the setting in [1] is time-inhomogenous, but the algorithm seems to work for time-homogenous setting as well)

2. The bonus terms added in Algorithm 1 and Algorithm 2 seem to be similar to the bonus term in Eq. (4) in [2] and Theorem 2 and 3 in [3]. There is no comparison to these related works in the paper.

3. The numerical experiments in this paper is only for the toy example. There is no experiments for real world applications.

[1] Zhang, Zihan, et al. "Settling the sample complexity of online reinforcement learning." The Thirty Seventh Annual Conference on Learning Theory. PMLR, 2024.

[2] Simchi-Levi, David, Zeyu Zheng, and Feng Zhu. "A Simple and Optimal Policy Design with Safety against Heavy-tailed Risk for Stochastic Bandits." arXiv preprint arXiv:2206.02969 (2022).

[3] Simchi-Levi, David, Zeyu Zheng, and Feng Zhu. "Regret distribution in stochastic bandits: Optimal trade-off between expectation and tail risk." arXiv preprint arXiv:2304.04341 (2023).

**Questions:**

1. Is there any way we can tighten the low order term from $S^2A$ into $SA$ to match the lower bound?

2. Can you obtain first order regret bound (regret bound in terms of $V^*$) and second order regret bound (regret bound in terms of $\mathrm{Var}(V^*)$)?

---

> ### Author Response · Authors · 2024-11-19
>
> We sincerely appreciate the reviewer’s time and effort in reviewing our paper and providing valuable feedback.
>
> Before we delve into our responses to each of the comments and questions, with all due respect, we would like to kindly note our perspective on the review's evaluation, as we feel it may not fully recognize the significance of our contributions. Our work achieves **the sharpest-ever minimax regret bound** to date and  **the tightest PAC bound** to date, even under **the weakest assumptions** —results that are both theoretically substantial and novel.
> This achievement is made possible by the development of a **new type of algorithm** and a **new analytical framework**, which are distinctly different from existing techniques. These contributions, in themselves, are noteworthy and merit recognition.
>
> What makes our work even more remarkable is that we achieve these results with a highly practical and easy-to-use algorithm. This combination of the state-of-the-art theoretical guarantees and practical usability is exceptionally rare in the minimax regime, further underscoring the value and impact of our contributions.
>
> ### **[W1/Q1] Time-homogeneous setting - Comparisons with Zhang et al. (2024)**
>
> We are glad to have the opportunity to clarify this specific point.
> The analysis of Zhang et al. (2024) does **not** hold for the time-homogeneous setting.
> The reasons are explained with details in Section 7 of their paper, which we will explain shortly.
> Tightening the low-order term from $S^2A$ to $SA$ for the time-homogeneous setting is still an open problem, and we strongly believe that the not solving this problem should not be considered our weakness.
> On the contrary, by improving the logarithmic factors, we achieve the tightest bounds for the time-homogeneous setting, so, if anything, it deserves credit.
> As mentioned in the footnote on page 3, the time-inhomogeneous setting is regarded as a special case of the time-homogeneous setting.
> Specifically, when encountering a time-inhomogeneous MDP with a state space $\mathcal{S}$ of cardinality $S$, it can be transformed into a time-homogeneous one with a state space $\mathcal{S}' := \mathcal{S} \times [H]$, where $(s, h) \in \mathcal{S}'$ corresponds to state $s$ visited at time step $h$ in the original MDP.
> The transformed state space has cardinality $S':= HS$.
> By running our algorithm on the time-homogeneous variant, it achieves a $\tilde{\mathcal{O}}(H \sqrt{S'AK}) = \tilde{\mathcal{O}}(H^{3/2} \sqrt{SAK})$ regret bound, which is minimax optimal for the time-inhomogeneous setting.
> However, algorithms designed specifically for the time-inhomogeneous setting cannot (at least trivially) achieve $\tilde{\mathcal{O}}(H\sqrt{SAK})$ regret for time-homogeneous instances since they cannot exploit time-homogeneity, that is, they cannot effectively use data of the same state-action pair collected from different time steps.
> Additionally, some analyses, including the one by Zhang et al. (2024), rely on the acyclic structure of the time-inhomogeneous MDP, being unable to fully adapt to time-homogeneous MDPs with cycles.
> Therefore, considering the time-homogeneous setting is, in fact, a generalization, but not vice-versa.

---

> > ### Comment · Reviewer_ud3P · 2024-11-24
> >
> > Thanks to the authors for their responses. I have increased the score accordingly. However, I am still concerned on the technical contribution of this paper. As mentioned by the authors, Simchi-Levi et al. (2022, 2023) focused on bandit setting while this paper focused on tabular RL, which has one more step of estimating the variance. This technique of estimation on variance also appears in many previous literatures, e.g. [1]. Additionally, the author mentioned that Simchi-Levi et al. (2022, 2023) does not use Freedman inequality, but in their analysis it achieves similar effects of the Freedman inequality. I recommend the authors to carry more detailed comparison to previous literatures.
> >
> > [1] Mohammad Gheshlaghi Azar, Ian Osband, and Remi Munos, "Minimax Regret Bounds for Reinforcement Learning".

---

> > > ### Author Response · Authors · 2024-11-25
> > >
> > > We sincerely thank the reviewer for increasing the score and recognizing our work!
> > >
> > > First, we would like to note that we included a discussion about Simchi-Levi et al. (2022, 2023) in the Related Work section (Section 1.1) of the last revision.
> > > We would like to kindly emphasize that one of our core contributions is that **our algorithm does not estimate the variance**, unlike Azar et al. (2017) and all subsequent related works, as highlighted in the introduction.
> > > Instead, we derive a novel method for controlling the amount of underestimation that may occur due to the unaddressed variance. We sincerely want to make sure that this key aspect is recognized.
> > >
> > > The bonus terms of Simchi-Levi et al. (2022, 2023) are designed to guarantee exponential decay of the failure probability of Hoeffding's inequality and the resulting light-tailedness of regret, and do not control underestimation.
> > > In addition, we highlight that the failure probabilities of the relevant concentration inequalities differ, demonstrating that the two bonus terms are fundamentally distinct.
> > >
> > > Our bonus term has the form $\frac{1}{n} \sqrt{\log \frac{1}{\delta}}$, originating from a concentration inequality that occurs with probability at least $1 - \delta$.
> > > In Simchi-Levi et al. (2022, 2023), their $\frac{1}{n} \sqrt{\log T}$-style bonus term leads to a concentration inequality whose failure probability is at most $O(\exp ( - \frac{T \log T}{n}))$.
> > > Plugging in this value into $\delta$ of our bonus term yields a completely different form.
> > >
> > > We confidently assure the reviewer that our analysis technique is significantly distinct from Simchi-Levi et al. (2022, 2023), as well as from previous works in RL. Thank you once again for your increased score. We hope we have addressed any remaining concerns and would be happy to respond to any further questions.

---

> ### Author Response · Authors · 2024-11-19
>
> ### **[W2] Comparisons with Simchi-Levi et al. (2022, 2023)**
>
> Thank you for bringing Simchi-Levi et al. (2022, 2023) to our attention. We are happy to include discussions of these papers in the revision. However, we would like to emphasize that the bonus term in our Algorithm 2 is fundamentally distinct from those proposed by Simchi-Levi et al. (2022, 2023). Based on our inspection, we believe the reviewer's comment is primarily relevant to Algorithm 1 with specific parameters chosen according to Theorems 1 and 2.
>
> To clarify the differences, while the resulting forms of the bonus terms may appear similar, the underlying motivations and derivations differ significantly. As the reviewer may be aware, Simchi-Levi et al. (2022, 2023) focus on multi-armed bandits with the primary objective of controlling the tail probability of the regret distribution—that is, minimizing the probability of observing large regret. Their specific bonus term arises from satisfying the probabilistic requirements needed for application of Hoeffding's inequality.
>
> In contrast, our work is aimed at developing a novel and simple algorithm for tabular reinforcement learning. The bonus term in our algorithm stems from a distinct context—decoupling the variance factors and visit counts that naturally arise in reinforcement learning settings when applying Freedman’s inequality. The use of this variant of Freedman’s inequality naturally leads to the form of the bonus term we employ. Importantly, Simchi-Levi et al. (2022, 2023) do not appear to leverage Freedman's inequality, either directly or indirectly, in their derivations.
>
> For these reasons, we believe the similarities between our work and that of Simchi-Levi et al. (2022, 2023) are largely superficial. While we are happy to include a discussion of these papers in the context of theorems for completeness, we do not view the absence of direct comparisons as a significant weakness of our paper.
>
> That said, we were intrigued to find that a similar form of bonus term has emerged from a different context. The work itself is intriguing, and verifying whether our algorithm also enjoys light-tailed regret would be an interesting direction for future research. We thank the reviewer again for this feedback. In the updated version, we have included the comparison with the references you provided.

---

> ### Author Response · Authors · 2024-11-19
>
> ### **[W3] Real-world application experiment?**
>
> We would like to clarify the scope and methodology of our experiments. If by "real-world application," the reviewer refers to experiments using real-world *offline* datasets, we respectfully explain the inherent challenges associated with such evaluations in reinforcement learning (RL) settings.
>
> Conducting experiments in RL with offline datasets is difficult due to the sequential decision-making nature of RL and its reliance on partial feedback. RL algorithms depend on feedback tied to specific sequences of actions and resulting state transitions, whereas offline datasets typically lack counterfactual information—i.e., the transitions and rewards for actions not taken during data collection. This limitation makes direct evaluation of RL algorithms on fixed real-world datasets impractical without significant adjustments, such as converting them into semi-synthetic datasets, which are essentially synthetic.
>
> For these reasons, we adopted the approach commonly used in the RL literature, focusing on theoretical results validated with synthetic environments. This methodology provides controlled experimentation and ensures that the theoretical properties of the algorithm are demonstrated under well-defined conditions. Regarding the RiverSwim environment, we emphasize that it is not merely a toy example but a well-established benchmark in RL literature, known for being both challenging and effective for evaluating algorithms' efficiency.
>
> To further address the reviewer's concerns, we have conducted additional experiments including two more complex environments: Atari 'freeway\_10\_fs30' [A] and Minigrid 'MiniGrid-KeyCorridorS3R1-v0' [B].
> These environments were tabularized using data from the BRIDGE dataset [C]. We observe that our algorithm consistently demonstrates competitive performance in these settings, further substantiating its practical value. Specific details and results of these experiments can be found in Appendix G of the uploaded revision (see Table 4 and Figures 1, 3, 4, 5, and 6).
>
> Additionally, we note that most works in RL theory primarily focus on theoretical contributions and often do not include experiments at all. Even when experiments are conducted, they are typically of comparable complexity (or much less) to those presented in our work. If the reviewer believes additional verification in more realistic settings is necessary, we are open to suggestions, including pointers to specific simulation environments, and we welcome further discussion to identify effective ways to demonstrate the advantages of our algorithm.
> So far, we believe that the updated version of our paper provides very compelling evidence of the effectiveness of our proposed method, including more thorough experiments than the vast majority of RL theory literature, much of which does not include any experimental validation.
>
> [A] M. G. Bellemare, Y. Naddaf, J. Veness, and M. Bowling. The arcade learning environment: An
> evaluation platform for general agents. Journal of Artificial Intelligence Research, 47:253–279,
> jun 2013.
> [B] Maxime Chevalier-Boisvert, Bolun Dai, Mark Towers, Rodrigo de Lazcano, Lucas Willems, Salem Lahlou, Suman Pal, Pablo Samuel Castro, and Jordan Terry. Minigrid \& miniworld: Modular \& customizable reinforcement learning environments for goal-oriented tasks. CoRR,
> abs/2306.13831, 2023.
> [C] Cassidy Laidlaw, Stuart Russell, and Anca Dragan. Bridging rl theory and practice with the effective
> horizon. In NeurIPS, 2023.
>
>
> ### **[Q2] Regret bound in terms of $V^ * $ and $\text{Var}(V^ * )$**
>
> To better answer your questions, we respectfully request additional clarification on what $V^*$ and $\text{Var}(V^*)$ represent, as these terms do not appear to align with the notations in our manuscript and could be interpreted in multiple ways.
> Specific definitions of first-order and second-order regret would also be appreciated, so that we can better understand and address your question.

---

### Official Review · Reviewer_HgJ4 · 2024-10-29

**Soundness:** 4
**Presentation:** 4
**Contribution:** 3
**Rating:** 8
**Confidence:** 4

**Summary:**

This paper studies regret minimization in tabular RL. They propose an algorithm based on the principle of “quasi-optimism” that, rather than satisfying standard optimism, satisfies a relaxed version of optimism (their estimates are not truly optimistic, but deviate from optimism by a bounded amount). They show that this relaxation of optimism allows them to obtain the tightest known bounds on regret minimization in tabular RL (in the time homogeneous setting), while also simplifying the algorithm substantially as compared to existing optimal algorithms for tabular RL. Furthermore, they show that their algorithm leads to better empirical performance than all existing algorithms for tabular RL with theoretical guarantees.

**Strengths:**

1. This paper introduces novel algorithmic and proof techniques for tabular RL, which allow it to obtain the tightest known bounds. As the majority of minimax optimal algorithms for tabular RL rely essentially on standard optimism, it was interesting to see these new techniques. In particular, the fact that you can achieve optimal regret without variance-dependent bonuses is surprising. I believe the insights and techniques presented here are a valuable and interesting contribution to the tabular RL literature.

2. The simplicity of the algorithm is also very nice. The algorithms with the best known guarantees tend to be quite complex and it is not obvious that one can achieve optimality with a very simple algorithm. This paper demonstrates this is possible (and indeed, achieves bounds tighter than the previously best known bounds), which is an exciting conclusion.

**Weaknesses:**

1. The motivation for this paper could be cleaned up somewhat. It is motivated by saying that we want algorithms for tabular RL which are both theoretically optimal but also perform well practically, and that we don’t currently have this. To my knowledge, there are essentially no real-world applications of tabular RL, however; it is more of a theoretical exercise. I do not think this is an issue, but it would be better to not overstate the importance of having a practically useful algorithm (or better justification for why this is important should be given).

2. Furthermore, I do not think the experimental results sufficiently validate the proposed algorithm’s practicallity or superior empirical performance over existing works. The algorithm is only validated on one environment (though the size and horizon are varied). If the authors want to convincingly make the argument that their approach yields better empirical performance, I would suggest benchmarking on additional environments (for example, those from the BRIDGE dataset of [1]). I do not think this is necessary—I think the theoretical results are sufficient on their own—I would just suggest toning down the claims on empirical performance without further experiments.

3. It would also be nice to see an empirical comparison against a posterior sampling-style approach (e.g. the algorithm of [2]).

4. I found the proof sketch in Section 4.4 to be fairly mechanical—some more intuitive explanation here would help elicit the key takeaways (for example it seems like the version of Freedman’s inequality here is playing a large role and giving an improvement over more traditional Bernstein-style bounds—this could be made more explicit).

Minor typos:
- Line 252: I believe there should be brackets around $c_k$.
- Line 523: “that the our algorithm” -> “that our algorithm”.

[1] Laidlaw, Cassidy, Stuart J. Russell, and Anca Dragan. "Bridging rl theory and practice with the effective horizon." Advances in Neural Information Processing Systems 36 (2023): 58953-59007.

[2] Osband, Ian, Daniel Russo, and Benjamin Van Roy. "(More) efficient reinforcement learning via posterior sampling." Advances in Neural Information Processing Systems 26 (2013).

**Questions:**

None.

---

> ### Author Response · Authors · 2024-11-19
>
> We sincerely appreciate the reviewer’s time and effort in reviewing our paper, recognizing our contributions, and providing valuable feedback.
>
> ### **[W1] Need for a practical tabular RL method**
>
> We appreciate the reviewer's feedback regarding the motivation for our work. While many real-world applications involve complex environments, there are many instances where tabular reinforcement learning (RL) methods are both applicable and beneficial.
> Let us provide some examples based on first-hand experience:
>
> Navigation Tasks: In structured environments, such as warehouse robotics or grid-based pathfinding, the state and action spaces can be discretized effectively. For example, a robot navigating a warehouse can have its environment represented as a grid, with each cell corresponding to a unique state and discrete movements as actions. In such settings, tabular RL methods can be employed to learn optimal navigation policies, enabling the robot to efficiently reach target locations while avoiding obstacles.
>
> Recommender Systems with Discrete User States: Certain recommender systems categorize users into discrete states based on their behaviors or preferences. Each state represents a distinct user status, and the system selects recommendations (actions) accordingly. By modeling the recommendation process as a Markov Decision Process (MDP) with discrete states and actions, tabular RL techniques can be applied to make recommendations more adaptive and efficient.
>
> It is important to note that in many sequential decision-making problems, whether function approximation can be used instead of a tabular MDP is not a choice available to the algorithm designer. This is because feature information for states or actions may not be provided or available at the outset. In such cases, function approximation is not a viable option, leaving tabular MDPs as the only formulation available for solving the problem.
>
> Furthermore, many real-world sequential decision-making problems are still formulated as tabular MDPs, particularly in domains where the problem space is inherently discrete or where limited feature information constrains the ability to use more complex representations. These examples illustrate that tabular RL is not merely a theoretical construct but remains a practical and necessary tool in numerous applications.
>
> Developing efficient tabular RL algorithms that are both theoretically sound and practically effective is crucial for optimizing performance in such domains. Our work aims to bridge this gap by providing algorithms that offer strong theoretical guarantees while being applicable to real-world problems where tabular methods are suitable

---

> ### Author Response · Authors · 2024-11-19
>
> ### **[W2] More experiments required (e.g. environments of BRIDGE)**
> ### **[W3] Experiment of PSRL**
>
> **[W2 \& W3]**
> We sincerely thank you for recognizing our theoretical contribution.
> Since the BRIDGE dataset is a dataset for deep RL, we find most of the environments in it not suitable for tabular RL experiments: either their state spaces are too large or they are too simple that a greedy policy performs the best, diminishing the purpose of comparing the efficiency of exploration strategies.
> We found two environments, 'freeway\_10\_fs30' from the Atari benchmark and 'MiniGrid-KeyCorridorS3R1-v0' from Minigrid, whose sizes and complexities fit our purposes.
> Although the reviewer mentioned that it is not necessary, we report the performance of each algorithm in these environments, where the results for PSRL are also included.
> We observe that PSRL is ineffective in these environments, whereas ours learns the MDPs very quickly.
>
> We would like to make a few notes regarding comparing our algorithm with PSRL.
>
> First, we would like to explain some challenges in directly comparing with algorithms with Bayesian guarantees, such as PSRL and RLSVI.
> These algorithms achieve Bayesian regret guarantees for given prior distributions over the MDPs; however, the prior is not available under the current experimental setting.
> While it is possible to construct an artificial prior, the performance of these algorithms depends on the prior; that is, they gain an advantage if the prior is informative.
> This makes it potentially unfair to compare them with algorithms that have frequentist regret guarantees, as the latter cannot use any prior information and must be more conservative.
> For example, RLSVI, another randomized algorithm, requires constant-scale perturbations for a Bayesian guarantee [A].
> However, the perturbations must be inflated by a factor of $HSA$ to ensure a frequentist regret bound [B].
> Typically, the constant-scaled version empirically performs much better for most reasonable MDPs.
> One way to make the comparison viable would be tuning the frequentist algorithms.
> While our EQO suffers no problem with this, how to tune the existing algorithms is unclear.
> A notable advantage of our algorithm is its simplicity in practice. While baseline algorithms often involve multiple parameters with complex dependencies, our approach consolidates these into a single parameter, $c_k$, making tuning much more straightforward. Theoretical results in Theorems 1, 3, and 4 justify setting $c_k$ as a $k$-independent constant, offering both theoretical and practical convenience.
> It is worth noting that the disadvantages of existing methods in parameter tuning are not limited to the number of parameters but also include the requirement that these tunable parameters often need to be specified as a rate (e.g., decreasing or increasing) rather than as a constant. This makes tuning significantly more complicated in practice. In contrast, our method requires only a single constant, making it much easier to deploy in real-world scenarios.
>
> As the purpose of the experiment with the RiverSwim MDP is to compare the performance of algorithms with theoretical guarantees, we set the parameters according to their theoretical values, and hence we do not tune the parameters and cannot include PSRL for comparison (since it does not have theoretical values for its parameters in this setting).
> For the additional experiments, we consider these settings are more like the real-world environments, where tuning the parameters becomes highly necessary.
> Therefore, we tune the algorithms and include PSRL.
> For each frequentist algorithm, a multiplicative factor for the whole bonus term is set as a tuning parameter, therefore maintaining the same complexity as EQO.
>
> [A] Ian Osband, Benjamin Van Roy, and Zheng Wen. Generalization and exploration via randomized value functions. In International Conference on Machine Learning, pp. 2377–2386. PMLR, 2016.
> [B] Andrea Zanette, David Brandfonbrener, Emma Brunskill, Matteo Pirotta, and Alessandro Lazaric. Frequentist regret bounds for randomized least-squares value iteration. In International Conference on Artificial Intelligence and Statistics, pp. 1954–1964. PMLR, 2020.

---

> ### Author Response · Authors · 2024-11-19
>
> ### **[W4] More intuitive explanation for the proof**
>
> We are more than happy to add more details in Section 4.4, especially regarding the proof of Lemma 1.
> We will certainly add most major points and intuitive explanations.
> The variant of Freedman's inequality we use is indeed crucial.
> While we have included the details in the uploaded revision, we briefly summarize some relevant points.
>
> The proof of quasi-optimism begins with a basic decomposition of $V_h^*- V_h^k$ as follows:
> $$
> \begin{aligned}
>     V_h^*(s) - V_h^k(s) & \le PV_{h+1}^*(s, a^*) - b^k(s, a^*) - \hat{P}^k V_h^k(s, a^*)
>     \\\\
>     & = - b^k(s, a^*) + \underbrace{( P - \hat{P}^k) V_{h+1}^*(s, a^*) }_ {I_1}+ \underbrace{\hat{P}^k (V_{h+1}^*- V_{h+1}^k) (s, a^*)}_ {I_2} . \qquad (1)
> \end{aligned}
> $$
> With the previous method of guaranteeing full optimism, one assumes $I_2 \le 0$ using mathematical induction, then faces the challenging task of fully bounding $I_1$ by $b^k(s, a^*)$.
> In the proof of Lemma 2 (previously Lemma 1), we set a slightly relaxed induction hypothesis.
> As a result, $I_2$ may be greater than zero, while $b^k(s, a^*)$ no longer needs to fully bound $I_1$.
> The key to quasi-optimism is to *allow underestimation of $I_1$, while controlling the resulting increase in $I_2$.*
>
> When bounding $I_1$, we use a variant of Freedman's inequality (Lemma 36), which implies (Lemma 5)
> $$
>     \left| (\hat{P}^k - P) V_{h+1}^*(s, a^*) \right| \le \frac{\lambda_k}{4 H}  \text{Var}(V_{h+1}^*)(s, a^*) + \frac{3 H \ell}{\lambda_k N^k(s, a^*)}
>      .
> $$
> $\lambda_k$ is some parameter and $\ell$ is some logarithmic factor.
> One advantage of this form is that the variance term and the $1/n$ term are isolated, whereas the previous Bernstein-type bound includes a term of the form $\sqrt{\text{Var} / n}$.
> While the sum of the variances achieves a tight bound within the expectation, the $1 / n$ terms must be summed according to actual visit counts.
> This discrepancy necessitates the use of multiple concentration inequalities, alternating between the expected and sampled trajectories.
> However, this lemma allows us to address the two factors independently.
> Especially for the proof of quasi-optimism, Lemma 2 (previously Lemma 1), we compensate the $1 / N$ term with the bonus $b$, while leaving the variance term unattended, which will be summed up across the time steps due to $I_2$ in Eq. (1) and attain a tight bound by itself.
> This approach is impossible with $\sqrt{\text{Var} / N}$ terms.

---

> > ### Comment · Reviewer_HgJ4 · 2024-11-22
> > **Response**
> >
> > Thanks to the authors for their response. I would like to keep my score as is. I would encourage the authors to provide explicit citations for the above claims on the applicability of tabular RL in practice, and include these in the final version of the paper.

---

> > > ### Author Response · Authors · 2024-11-23
> > >
> > > Thank you very much for your continued support!
> > > We will include examples and citations of applications of tabular RL in the revision.

---

### Official Review · Reviewer_pA4T · 2024-11-03

**Soundness:** 3
**Presentation:** 2
**Contribution:** 4
**Rating:** 8
**Confidence:** 3

**Summary:**

The authors proposed a novel algorithm for reinforcement learning in finite episodic MDPs based on the principle of Quasi-Optimism. In particular, they showed that simple exploration bonuses without any variance information can achieve the minimax optimal regret bound. Additionally, the method is easy to implement and shows good empirical performance.

**Strengths:**

- To my knowledge, it is the first result on a bonus-based RL algorithm that does not use variance information to achieve a minimax optimal regret bound. It is a major improvement over the prior work since, before, only posterior-sampling-based algorithms could achieve minimax optimal bound without the direct usage of the variance estimates (see, e.g., Tiapkin et al. 2022).



Tiapkin, D., Belomestny, D., Calandriello, D., Moulines, É., Munos, R., Naumov, A., ... & Ménard, P. (2022). Optimistic posterior sampling for reinforcement learning with few samples and tight guarantees. Advances in Neural Information Processing Systems, 35, 10737-10751.

**Weaknesses:**

- The algorithm cannot adapt to deterministic environments. In particular, the usage of variance information allows the Bernstein bonuses to become of order almost $H/N$ in the case of deterministic environments since the variance, in this case, will be learned to be zero. In contrast, the bonuses of the presented algorithm always have order $H\sqrt{K}/N$, which is much larger than adaptive variance-dependent bonuses.
- The main technical novelty presentation should be improved. In particular, I would prefer to see the full proof of quasi-optimism in the main text since it is the most insightful part of the paper that shows why the scaling of the additional term is $\lambda_k H$ and not $\lambda_k H^2$, how it may follow from a naive analysis that uses precisely the statement of Lemma 1 for the induction. In particular, very subtle work with variance and second moments looks to be a crucial part of the paper that allows the errors not to accumulate through the horizon, and this part should be acknowledged in the paper much better. I consider raising my score if the presentation of the proof in the main text shows this critical aspect.

**Questions:**

- Why the time-homogeneous setting was selected? What limits the application of these techniques to a time-inhomogeneous setting?
- The experiments do not compare randomized exploration methods such as PSRL and RLSVI. Could you add these additional comparisons?
- How does the regret bound change under the sparse regret setting $R^k_h \leq 1$?

---

> ### Author Response · Authors · 2024-11-19
>
> We sincerely appreciate the reviewer’s time and effort in reviewing our paper and providing valuable feedback.
>
> Before we delve into our responses to each of the comments and questions, we would like to make sure that the significance of our contributions is adequately recognized. Our work achieves **the sharpest-ever minimax regret bound** to date and  **the tightest PAC bound** to date, even under **the weakest assumptions** —results that are both theoretically substantial and novel.
> This achievement is made possible by the development of a **new type of algorithm** and a **new analytical framework**, which are distinctly different from existing techniques. These contributions, in themselves, are noteworthy and merit recognition.
>
> What makes our work even more remarkable is that we achieve these results with a highly practical and easy-to-use algorithm. This combination of the state-of-the-art theoretical guarantees and practical usability is exceptionally rare in the minimax regime, further underscoring the value and impact of our contributions.
>
>
> ### **[W1] Deterministic case**
>
> Thank you for raising this point. We are happy to discuss and clarify that dealing with a deterministic environment is not an issue.
>
> While plainly setting $O(\sqrt{K})$ for $c_k$ in our algorithm introduces an additional $\sqrt{K}$-dependence in the bonus term compared to algorithms that utilize empirical variances in deterministic environments, such a parameter choice for $c_k$ is particularly designed to provide minimax guarantees for regret performance. Clearly, our algorithm achieves minimax regret bounds across a wide range of environments, achieving both robustness and practicality. However, note that the constant $c_k$ can be specified suitably for any particular environment to achieve better empirical performance.
>
> For PAC tasks, as shown in Line 4 of Algorithm 2, the bonus terms in our algorithm are independent of $k$ or $K$. This independence highlights the flexibility of our approach and its adaptability to various problem settings, including deterministic settings.
>
> To address concerns about deterministic environments, one could extend our algorithm by incorporating empirical variance alongside our existing mechanism and adopting a hybrid approach. For instance, the algorithm could selectively switch to variance-based bonuses for specific state-action pairs (or the entire environment) when they are observed to be deterministic or sufficiently close to deterministic. This modification would allow the algorithm to leverage the advantages of empirical variance where applicable while retaining the efficiency of our current method.
>
> We also emphasize the practicality of our algorithm as a key distinction. While baseline algorithms often involve multiple parameters with complex dependencies, our approach consolidates these into a single parameter, $c_k$, significantly simplifying tuning. Theoretical results in Theorems 1, 3, and 4 justify setting $c_k$ as a $k$-independent constant, providing both theoretical rigor and practical convenience. Moreover, existing methods often require tunable parameters specified as rates (e.g., decreasing or increasing), which can make tuning substantially more challenging in practice. In contrast, our method's single-constant parameterization makes it easier to deploy in real-world scenarios, maintaining both efficiency and simplicity.
>
> In summary, while our algorithm is not specifically tailored for deterministic environments, it provides a general, robust solution with minimax optimality and practical usability.

---

> ### Author Response · Authors · 2024-11-19
>
> ### **[W2] Presentation of the main technical novelty**
>
> We are more than happy to add more details in Section 4.4, especially regarding the proof of Lemma 1.
> Although we find it challenging to include its full proof due to space limitation, we will certainly add most major points and intuitive explanations.
> Dealing with variance and second moments is indeed crucial, and it is the core part that derives an addition $\lambda_k H$ term instead of $\lambda_k H^2$.
> While we have included the details in the uploaded revision, we briefly summarize some relevant points.
>
> We begin with a basic decomposition of $V_h^*- V_h^k$ as follows:
>
> $$
> \begin{aligned}
>     V_h^*(s) - V_h^k(s) & \le PV_{h+1}^*(s, a^*) - b^k(s, a^*) - \hat{P}^k V_h^k(s, a^*)\\\\
>     & = - b^k(s, a^*) + \underbrace{( P - \hat{P}^k) V_{h+1}^*(s, a^*) }_ {I_1} + \underbrace{\hat{P}^k (V_{h+1}^*- V_{h+1}^k) (s, a^*)}_ {I_2}  . \qquad (1)
> \end{aligned}
> $$
>
> With the previous method of guaranteeing full optimism, one assumes $I_2 \le 0$ using mathematical induction, then faces the challenging task of fully bounding $I_1$ by $b^k(s, a^*)$.
> In the proof of Lemma 2 (previously Lemma 1), we set a slightly relaxed induction hypothesis.
> As a result, $I_2$ may be greater than zero, while $b^k(s, a^*)$ no longer needs to fully bound $I_1$.
> The key to quasi-optimism is to *allow underestimation of $I_1$, while controlling the resulting increase in $I_2$.*
>
> Using a variant of Freedman's inequality (Lemma 36) to bound $I_1$, we obtain that
> \begin{equation*}
>     V_h^*(s) - V_h^k(s) \le -b^k(s, a^*) + \frac{\lambda_k}{4 H}  \text{Var}(V_{h+1}^*)(s, a^*) + \frac{3 H \ell}{\lambda_k N^k(s, a^*)} + \hat{P}^k (V_{h+1}^k- V_{h+1}^*) (s, a^*)  .
> \end{equation*}
> We set $b^k(s, a^*) = \frac{3 H \ell}{\lambda_k N^k(s, a^*)}$ to compensate the $1 / N$ term, but leave the variance term.
> Then, we obtain a recurrence relation of
> \begin{equation*}
>     V_h^*(s) - V_h^k(s) \le \frac{\lambda_k}{4H} \text{Var}(V_{h+1}^*)(s, a^*) + \hat{P}^k(V_{h+1}^* - V_{h+1}^k)(s, a^*)
>      .
> \end{equation*}
> Indeed, a naive bound for $V_h^*(s) - V_h^k(s)$ derived from this inequality is $\lambda_k H^2$, as the variance term is at most $H^2$ and is summed over at most $H$ time steps.
> Meanwhile, the expected sum of variances along a trajectory has a non-trivial bound of $H^2$ instead of $H^3$, and this fact has been frequently exploited to achieve better $H$-dependency in the regret bound.
> However, it has not been used for showing optimism, as assuming $I_2 \le 0$ and fully bounding $I_1$ with $b^k(s, a^*)$ in Eq. (1) does not allow any interaction between time steps.
> Furthermore, the existing proofs of the fact rely on the boundedness of returns (see, for example, Eq. (26) in Azar et al. (2017)).
> We derive a novel method that does not require such a condition and is applicable to showing quasi-optimism.
> The main challenges include dealing with the fact that the sum of variances are tightly bounded only when summed along a true trajectory based on $P$, whereas the recurrence term $I_2$ is based on its estimation $\hat{P}^k$.
> We overcome this difficulty by proposing a technical induction argument that covers the additional errors incurred by the difference.

---

> ### Author Response · Authors · 2024-11-19
>
> ### **[Q1] Time-homogeneous setting**
>
> We are glad to have the opportunity to clarify this specific point.
> In short, there is no limit of this technique to be applied to the time-inhomogeneous setting.
> In fact, *the minimax regret bound for the time-inhomogeneous setting, $\tilde{\mathcal{O}}(H^{3/2}\sqrt{SAK})$, can be directly derived from our results without any modification in the proof.*
> As mentioned in the footnote on page 3, the time-inhomogeneous setting is regarded as a special case of the time-homogeneous setting.
> Specifically, when encountering a time-inhomogeneous MDP with a state space $\mathcal{S}$ of cardinality $S$, it can be transformed into a time-homogeneous one with a state space $\mathcal{S}' := \mathcal{S} \times [H]$, where $(s, h) \in \mathcal{S}'$ corresponds to state $s$ visited at time step $h$ in the original MDP.
> The transformed state space has cardinality $S':= HS$.
> By running our algorithm on the time-homogeneous variant, it achieves a $\tilde{\mathcal{O}}(H \sqrt{S'AK}) = \tilde{\mathcal{O}}(H^{3/2} \sqrt{SAK})$ regret bound, which is minimax optimal for the time-inhomogeneous setting.
> However, algorithms designed specifically for the time-inhomogeneous setting cannot (at least trivially) achieve $\tilde{\mathcal{O}}(H\sqrt{SAK})$ regret for time-homogeneous instances since they cannot exploit time-homogeneity, that is, they cannot effectively use data of the same state-action pair collected from different time steps.
> Additionally, some analyses rely on the acyclic structure of the time-inhomogeneous MDP, being unable to fully adapt to time-homogeneous MDPs with cycles.
> Therefore, considering the time-homogeneous setting is, in fact, a generalization, which naturally implies the results for the time-inhomogeneous setting.

---

> ### Author Response · Authors · 2024-11-19
>
> ### **[Q2] Experiment with PSRL, RLSVI**
>
> Thank you for your suggestions.
> We conducted additional experiments in more complex environments as other reviewers suggested, and the result of PSRL is included.
> We observe that PSRL is ineffective in these environments, whereas ours learns the MDPs very quickly.
>
> We would like to make a few notes regarding comparing our algorithm with PSRL.
>
> First, we would like to explain some challenges in directly comparing with algorithms with Bayesian guarantees, such as PSRL and RLSVI.
> These algorithms achieve Bayesian regret guarantees for given prior distributions over the MDPs; however, the prior is not available under the current experimental setting.
> While it is possible to construct an artificial prior, the performance of these algorithms depends on the prior; that is, they gain an advantage if the prior is informative.
> This makes it potentially unfair to compare them with algorithms that have frequentist regret guarantees, as the latter cannot use any prior information and must be more conservative.
> For example, RLSVI requires constant-scale perturbations for a Bayesian guarantee [A].
> However, the perturbations must be inflated by a factor of $HSA$ to ensure a frequentist regret bound [B].
> Typically, the constant-scaled version empirically performs much better for most reasonable MDPs.
> One way to make the comparison viable would be tuning the frequentist algorithms.
> While our EQO suffers no problem with this, how to tune the existing algorithms is unclear.
> A notable advantage of our algorithm is its simplicity in practice. While baseline algorithms often involve multiple parameters with complex dependencies, our approach consolidates these into a single parameter, $c_k$, making tuning much more straightforward. Theoretical results in Theorems 1, 3, and 4 justify setting $c_k$ as a $k$-independent constant, offering both theoretical and practical convenience.
> It is worth noting that the disadvantages of existing methods in parameter tuning are not limited to the number of parameters but also include the requirement that these tunable parameters often need to be specified as a rate (e.g., decreasing or increasing) rather than as a constant. This makes tuning significantly more complicated in practice. In contrast, our method requires only a single constant, making it much easier to deploy in real-world scenarios.
>
> As the purpose of the experiment with the RiverSwim MDP is to compare the performance of algorithms with theoretical guarantees, we set the parameters according to their theoretical values, and hence we do not tune the parameters and cannot include PSRL for comparison (since it does not have theoretical values for its parameters in this setting).
> For the additional experiments, we consider these settings are more like the real-world environments, where tuning the parameters becomes highly necessary.
> Therefore, we tune the algorithms and include PSRL.
> For each frequentist algorithm, a multiplicative factor for the whole bonus term is set as a tuning parameter, therefore maintaining the same complexity as EQO.
>
> [A] Ian Osband, Benjamin Van Roy, and Zheng Wen. Generalization and exploration via randomized value functions. In International Conference on Machine Learning, pp. 2377–2386. PMLR, 2016.
> [B] Andrea Zanette, David Brandfonbrener, Emma Brunskill, Matteo Pirotta, and Alessandro Lazaric. Frequentist regret bounds for randomized least-squares value iteration. In International Conference on Artificial Intelligence and Statistics, pp. 1954–1964. PMLR, 2020.
>
> ### **[Q3] Sparse regret?**
>
> We believe the reviewer is referring to the sparse reward setting, where the return, $\sum_{h=1}^H R_h^k$, is bounded by 1.
> We would like to emphasize that our setting encompasses the sparse reward setting with only a scaling difference of $H$, where this scaling is chosen solely for simple comparisons with existing works.
> By scaling our algorithm and the proofs by the factor $1 / H$, we achieve a $\tilde{\mathcal{O}}(\sqrt{SAK})$ regret bound for the sparse setting.
> In fact, our assumptions are more general, as we only require the value function to be bounded by 1 after scaling, whereas the sparse reward setting bounds all possible outcomes of the return.
> As the value function is the expectation of the return, bounded returns imply bounded value function.
> We refer to Section 4.1 for details about the relationship between the two.

---

> > ### Comment · Reviewer_pA4T · 2024-11-24
> >
> > I would like to thank the authors for their response. With a more extended sketch of the main proof, I am happy to increase my score.

---

> > > ### Author Response · Authors · 2024-11-27
> > >
> > > Thank you very much for your continued support and for recognizing the significance of our contributions!

---

### Official Review · Reviewer_8Sjp · 2024-11-04

**Soundness:** 3
**Presentation:** 3
**Contribution:** 3
**Rating:** 6
**Confidence:** 4

**Summary:**

The paper studies reinforcement learning in tabular finite-horizon MDPs. The authors introduce EQO (Exploration via Quasi-Optimism), a novel algorithm which is basically a variant of UCBVI with much simpler bonuses of the form b_k(s,a) = c / N_k(s,a) (ie not requiring any variance term and decaying at 1/k rate instead of 1/sqrt(k)). EQO is shown to be minimax optimal for both regret minimization and PAC identification of an epsilon-optimal policy. The main novelty in the analysis is a notion of "quasi optimism", which allows EQO to achieve strong theoretical guarantees despite not being exactly optimistic at every episode. On a simple numerical experiment, EQO outperforms existing baselines in terms of cumulative regret.

**Strengths:**

1. Making provably-efficient RL algorithms, even for the simplest setting of tabular MDPs, more practical is a relevant problem, and the paper makes a good contribution in this direction
2. EQO achieves strong results despite its simplicity, which I think is a big plus
3. I found the idea of "quasi optimism" novel and interesting. Most existing works do need "strong" optimism (ie Q functions are optimistic at any s,a,h,k) to derive theoretical guarantees, and it is good to see that this can be relaxed
4. The paper is overall well written: the context is well motivated from the very beginning, sufficient intuition is given behind all results, and proofs/notation are clear (though I have to say that I only skimmed through the proofs quickly)
5. Relaxing existing assumptions on reward/return/value distributions (Assumption 1 and 2) is also a relevant contribution (though I have never found them as limiting factors for existing approaches, but rather just tricks to simplify proofs/notation)

**Weaknesses:**

My main concern is about the way these novel bonuses are built: it seems that the price to pay for having b^k(s,a) decay as 1/N_k(s,a) is an inflation of the multiplicative constant (c_k) by a factor sqrt(K) or sqrt(k), whereas existing bonuses only have logarithmic dependences in this quantity. This may have a some important negative implications:
1. If I am not mistaken, this will prevent deriving any sort of logarithmic (in K) / gap-dependent bound for the same algorithm, since a factor sqrt(K) in the regret may be unavoidable if forced into the confidence bonuses. If this is true, I think it is a quite big limitation, as one of the main directions to make these approaches "practical" is to show that they can adapt to the complexity of the specific MDP instance they face (mostly though logarithmic / gap-dependent bounds), instead of paying the cost of the hardest instance (which may not even be of practical relevance).
2. There is a long line of works on instance-dependent results for regret minimization [1] and PAC RL [2,3,4,5] mostly studying algorithms that are quite similar to the baselines considered here. My concern is that the current algorithm cannot achieve results comparable to them, which means that comparing to existing baselines only in terms of worst-case results may not give a complete view of EQO's pros and cons. Also, in light of this, I feel that discussing literature on instance-dependent results is quite important, but at the moment all these papers are missing from the related literature
3. This also break a bit the story about existing algorithms focusing only on minimax optimality. Eg the introduction states that "Although provably efficient RL algorithms offer regret bounds that are nearly optimal (up to logarithmic or constant factors), they are often designed to handle worst-case scenarios. This focus on worst-case outcomes leads to overly conservative behavior". But it is known that these algorithms can get guarantees which go beyond the minimax one, while (again, if the conjecture above is true) EQO may not. This would make EQO even more "minimax-focused" than existing literature.
4. I was surprised to see EQO outperforming all existing baselines despite the sqrt(K) term in the confidence bounds. I wonder if the experiments focus too much on the "low-K regime", i.e., where the algorithms are still far from converging to a (near-) optimal policy and have to pay a sqrt(K) regret. I wonder what happens in the "large-K regime", where existing algorithms should essentially transition to logarithmic regret while the EQO should (probably) still suffer sqrt(K). Also, existing papers on instance-dependent RL show examples of MDPs where good adaptive algorithms would achieve sample-complexity/regret much smaller than minimax (eg when you have a very large sub-optimality gap in some state that allows the agent to quickly "eliminate" an entire branch of the MDP, see eg Figure 1 in [1]). I wonder how EQO compares to existing baselines on such "favorable" instances

Other less important limitations:

5. About the bound for best-policy identification (Theorem 4): one important thing to note is that the algorithm has no adaptive stopping criterion, which means that in practice we have to run it until K_0 to get the guarantee on the returned policy. That can be very conservative, as an algorithm with adaptive stopping may empirically take much less than the derived worst-case bound


### REFERENCES


[1] Dann, C., Marinov, T. V., Mohri, M., & Zimmert, J. (2021). Beyond value-function gaps: Improved instance-dependent regret bounds for episodic reinforcement learning. Advances in Neural Information Processing Systems, 34, 1-12.

[2] Wagenmaker, A. J., Simchowitz, M., & Jamieson, K. (2022, June). Beyond no regret: Instance-dependent pac reinforcement learning. In Conference on Learning Theory (pp. 358-418). PMLR.

[3] Tirinzoni, A., Al Marjani, A., & Kaufmann, E. (2022). Near instance-optimal pac reinforcement learning for deterministic mdps. Advances in neural information processing systems, 35, 8785-8798.

[4] Wagenmaker, A., & Jamieson, K. G. (2022). Instance-dependent near-optimal policy identification in linear mdps via online experiment design. Advances in Neural Information Processing Systems, 35, 5968-5981.

[5] Al-Marjani, A., Tirinzoni, A., & Kaufmann, E. (2023, July). Active coverage for pac reinforcement learning. In The Thirty Sixth Annual Conference on Learning Theory (pp. 5044-5109). PMLR.

**Questions:**

1. What's the authors opinion about the impossibility of proving log(K) regret with EQO? Again, I am not 100% sure about it so I'd be happy to be proven wrong
2. Is the assumption of time-homogeneous transition kernel important for the techniques developed here or could the proofs go through even with time-inhomogeneous dynamics?

---

> ### Author Response · Authors · 2024-11-19
>
> We sincerely appreciate the reviewer’s time and effort in reviewing our paper and providing valuable feedback.
>
> Before we delve into our responses to each of the comments and questions, with all due respect, we would like to kindly note our perspective on the review's evaluation, as we feel it may not fully recognize the significance of our contributions. Our work achieves **the sharpest-ever minimax regret bound** to date and  **the tightest PAC bound** to date, even under **the weakest assumptions** —results that are both theoretically substantial and novel.
> This achievement is made possible by the development of a **new type of algorithm** and a **new analytical framework**, which are distinctly different from existing techniques. These contributions, in themselves, are noteworthy and merit recognition.
>
> What makes our work even more remarkable is that we achieve these results with a highly practical and easy-to-use algorithm. This combination of the state-of-the-art theoretical guarantees and practical usability is exceptionally rare in the minimax regime, further underscoring the value and impact of our contributions.

---

> ### Author Response · Authors · 2024-11-19
>
> ### **[W1-4, Q1] Instance-dependent bound?**
>
> [W1-3, Q1]
> We respectfully address the reviewer's comments and highlight the strengths of our work. While we acknowledge that gap-dependent bounds and the related literature represent an interesting direction, we emphasize that our primary focus is on achieving a minimax regret bound with a simple and robust algorithmic design. Importantly, the algorithms considered in our comparisons as well as vast majority of instance-independent regret analysis works do not explicitly claim or achieve instance-dependent bounds, nor do their respective papers discuss this possibility
> (Jaksch et al., 2010; Agrawal \&  Jia, 2017; Azar et al., 2017; Jin et al., 2018; Russo, 2019; Zhang et al., 2020; 2021a; Li et al. 2021; M\'enard et al. 2021b; Agarwal et al., 2023; He et al., 2023).
> Therefore, we believe it is appropriate to evaluate our work based on (instance-independent) minimax performance, which is the problem setting our work explicitly targets, as clearly stated in the paper.
>
> While we conjecture that plainly setting $O(\sqrt{K})$ for $c_k$ may be prohibitive for attaining $\log K$-type regret in the instance-dependent setting, we emphasize that the $O(\sqrt{K})$ choice for $c_k$ is proven only for the minimax setting in our work and has not been shown to be necessary for the instance-dependent setting. A different order for $c_k$ may indeed be required to achieve instance-dependent guarantees, but addressing this is beyond the scope of our current work.
>
> The parameter choice in our algorithm is designed to provide strong theoretical guarantees while maintaining simplicity. For PAC tasks, our bonus terms are independent of $k$ or $K$, as shown in Line 4 of Algorithm 2. This demonstrates the flexibility of our approach and its capability to adapt to different settings. Exploring parameter adaptivity to derive instance-dependent bounds is an intriguing direction, but it lies outside the scope of this work. Our focus is on presenting a simple yet effective parameter setting that achieves minimax regret, which is a significant contribution on its own.
>
> On the other hand, it is important to recognize a notable advantage of our algorithm is its simplicity in practice. While baseline algorithms often involve multiple parameters with complex dependencies, our approach consolidates these into a single parameter, $c_k$, making tuning much more straightforward. Theoretical results in Theorems 1, 3, and 4 justify setting $c_k$ as a $k$-independent constant, offering both theoretical and practical convenience.
> It is worth noting that the disadvantages of existing methods in parameter tuning are not limited to the number of parameters but also include the requirement that these tunable parameters often need to be specified as a rate (e.g., decreasing or increasing) rather than as a constant. This makes tuning significantly more complicated in practice. In contrast, our method requires only a single constant, making it much easier to deploy in real-world scenarios.
>
> [W4] Regarding the experiments, our experiments in the original submission with $K = 100,000$ episodes, hence up to $T = 16,000,000$ total timesteps (Figure 2 of the original submission), do not represent a "low-$K$" regime in any practical sense. This is particularly true when compared to other works in RL theory, most of which do not even include experiments.
> Nonetheless, we gladly conducted additional experiments including even larger $K = 500,000$ episodes on an MDP with large gaps, as described in Dann et al. (2021) provided by the reviewer as well as other additional experiments (see Figures 3, 4, 5, and 6 in the updated version). The results demonstrate that our algorithm still remains competitive, consistently outperforming most baselines.
>
> In conclusion, while gap-dependent bounds represent an interesting area of research, our work provides strong theoretical guarantees and demonstrates robust empirical performance under the minimax framework. We are grateful for the reviewer's thoughtful comments and welcome further discussion to refine and expand upon our contributions. We are more than happy to include this discussion in the revision and perform more experiments if needed. At the same time, we kindly request that our work be evaluated primarily within the context of the minimax regime, which aligns with our central focus and contributions.

---

> ### Author Response · Authors · 2024-11-19
>
> ### **[W5] Adaptive stopping criterion for BPI**
>
> We respectfully clarify that there may be a possible misunderstanding about our Algorithm 2. When solving the Best Policy Identification (BPI) task, the algorithm does include an adaptive stopping criterion. Specifically, the algorithm can terminate as soon as it identifies an $\varepsilon$-optimal policy, which occurs when the "if" statement on Line 11 is satisfied.
> Theorem 4 guarantees that this condition will be met *within* $K_0 + 1$ episodes but does not imply that the algorithm must run for exactly $K_0 + 1$ episodes. In practice, the stopping time is determined adaptively based on the algorithm's progress.
>
> The current version of Algorithm 2 does not explicitly highlight this stopping rule because it is designed to accommodate mistake-style PAC tasks, where stopping or returning a policy is not required. However, we appreciate the reviewer's suggestion and will gladly add details to Algorithm 2 to explicitly specify that the algorithm can terminate early once a good policy is found when solving the BPI task. This adjustment will help clarify the algorithm’s adaptability for different task settings.
>
> ### **[Q2] Time-homogeneous setting**
>
> We are glad to have the opportunity to clarify this specific point.
> In short, *the minimax regret bound for the time-inhomogeneous setting, $\tilde{\mathcal{O}}(H^{3/2}\sqrt{SAK})$, can be directly derived from our results without any modification in the proof.*
> As mentioned in the footnote on page 3, the time-inhomogeneous setting is regarded as a special case of the time-homogeneous setting.
> Specifically, when encountering a time-inhomogeneous MDP with a state space $\mathcal{S}$ of cardinality $S$, it can be transformed into a time-homogeneous one with a state space $\mathcal{S}' := \mathcal{S} \times [H]$, where $(s, h) \in \mathcal{S}'$ corresponds to state $s$ visited at time step $h$ in the original MDP.
> The transformed state space has cardinality $S':= HS$.
> By running our algorithm on the time-homogeneous variant, it achieves a $\tilde{\mathcal{O}}(H \sqrt{S'AK}) = \tilde{\mathcal{O}}(H^{3/2} \sqrt{SAK})$ regret bound, which is minimax optimal for the time-inhomogeneous setting.
> However, algorithms designed specifically for the time-inhomogeneous setting cannot (at least trivially) achieve $\tilde{\mathcal{O}}(H\sqrt{SAK})$ regret for time-homogeneous instances since they cannot exploit time-homogeneity, that is, they cannot effectively use data of the same state-action pair collected from different time steps.
> Additionally, some analyses rely on the acyclic structure of the time-inhomogeneous MDP, being unable to fully adapt to time-homogeneous MDPs with cycles.
> Therefore, considering the time-homogeneous setting is, in fact, a generalization, which naturally implies the results for the time-inhomogeneous setting.

---

> > ### Comment · Reviewer_8Sjp · 2024-11-27
> >
> > I thank the authors for the very detailed response and for running additional experiments and revising the paper in such a short time. It clarified all my doubts, especially the impossibility of proving log(K) bounds, the empirical performance in the long run, the misunderstanding about the BPI stopping condition (I had actually referred to Alg. 1 instead of 2, apologies for that), and the possibility to adapt the proofs to the time-inhomogeneous setting. I have thus increased my score and confidence.
> >
> > I don't consider increasing my score further as I don't find the idea of inflating the confidence bonus extremely exciting. It is definitely interesting and worth publication, but it comes with a key limitation that might prevent improving the theoretical results for this algorithm further, while one may argue that there is still lots of room for improvement for existing methods (eg through gap-dependent logarithmic bounds).
> >
> >
> > Let me also mention a few points on which I respectfully disagree with the authors (they did not influence my decision).
> >
> > > we kindly request that our work be evaluated primarily within the context of the minimax regime
> >
> > I don't believe a paper can be evaluated only in the minimax regime while ignoring the instance-dependent literature, as if the two were fundamentally different. Both are solving the same problem, regret minimization / PAC RL, and we should focus on designing the "best" (statistically and computationally) algorithms supported by the strongest theoretical guarantees.
> >
> > > For PAC tasks, our bonus terms are independent of k
> >
> > They do have an undesirable $1/\epsilon$ though. Since the sample complexity of PAC RL is of the order $K \simeq 1/\epsilon^2$, that's the equivalent of having a $\sqrt{K}$ dependence in the number of learning steps
> >
> > > Theoretical results in Theorems 1, 3, and 4 justify setting $c_k$ as a k-independent constant
> >
> > I don't see why. If $c_k$ does not depend on k or K or $1/\epsilon$, all theoretical results break and we enter the real of "heuristic algorithms", where everything can in principle be done.

---

> > > ### Author Response · Authors · 2024-12-04
> > >
> > > We sincerely thank the reviewer for increasing the score and recognizing the contribution of our work.
> > > We are more than happy to to briefly clarify that:
> > >
> > > > Theoretical results in Theorems 1, 3, and 4 justify setting  as a $k$-independent constant
> > >
> > > refers to small $k$ only, not large $K$ or $\varepsilon$, in the context of tuning the constant factors of the parameters to adapt to some unknown characteristics of the environment, such as the reward noise level.
> > > We have justification that our algorithm works well for some constant input $c_k \equiv c$, where it might perform better with another tuned constant.
> > >
> > > We again thank the reviewer for carefully reviewing our paper and providing valuable feedback.

---

### Meta-Review · Area_Chair_JWjQ · 2024-12-20

**Metareview:**

This paper studies a novel algorithm for tabular reinforcement learning based on the principle of Quasi-Optimism. This papers shows that strong optimism, which was required by prior work, could be relaxed and a much simpler bonus term is sufficient. The authors further show that the new algorithm leads to better empirical performance compared to existing ones.

Overall, this paper provides a nice contribution to the RL theory community. The reviewers also voted unanimously for acceptance.

**Additional Comments On Reviewer Discussion:**

The reviewers raised concerns regarding the motivation of the paper, the technical contribution of the paper, and whether the experiments are sufficient for validating the proposed algorithm’s practicallity. The authors provided detailed responses which successfully addressed those concerns and resulted in improved scores.

---

### Decision · Program_Chairs · 2025-01-22

Accept (Poster)